# Robust Integrated Learning and Pauli Noise Mitigation for Parametrized Quantum Circuits

**Md Mobasshir Arshed Naved**
Department of Computer Science
Purdue University
West Lafayette, USA
naved@purdue.edu

**Wenbo Xie**
Department of Computer Science
Purdue University
West Lafayette, USA
xie401@purdue.edu

**Wojciech Szpankowski**
Department of Computer Science
Purdue University
West Lafayette, USA
szpan@purdue.edu

**Ananth Grama**
Department of Computer Science
Purdue University
West Lafayette, USA
ayg@cs.purdue.edu

## Abstract

We propose a novel gradient-based framework for learning parameterized quantum circuits (PQCs) in the presence of Pauli noise in gate operation. The key innovation in our framework is the simultaneous optimization of model parameters and learning of an inverse noise channel, specifically designed to mitigate Pauli noise. Our parametrized inverse noise model utilizes the Pauli-Lindblad equation and relies on the principle underlying the Probabilistic Error Cancellation (PEC) protocol to learn an effective and scalable mechanism for noise mitigation. In contrast to conventional approaches that apply predetermined inverse noise models during execution, our method systematically mitigates Pauli noise by dynamically updating the inverse noise parameters in conjunction with the model parameters, facilitating task-specific noise adaptation throughout the learning process. We employ proximal stochastic gradient descent (proximal SGD) to ensure that updates are bounded within a feasible range to ensure stability. This approach allows the model to converge efficiently to a stationary point, balancing the trade-off between noise mitigation and computational overhead, resulting in a highly adaptable quantum model that performs robustly in noisy quantum environments.

## 1 Introduction

Quantum systems offer the potential to solve computationally hard problems in simulation, optimization, and machine learning. However, they pose significant challenges on NISQ devices [22]. The presence of quantum noise, which leads to errors in state preparation, gate operation, and measurement [12, 20], severely constrains the performance of variational quantum algorithms (VQAs) and quantum machine learning (QML) tasks. A major contributor to quantum noise in NISQ devices is gate noise, commonly modeled as Pauli noise [29]. Pauli noise occurs when unintended Pauli operators ($X, Y, Z$) are applied to quantum states during gate operations. These disruptions degrade the fidelity of quantum circuits and, in turn, the performance of parameterized quantum circuits (PQCs) commonly used in optimization and machine learning. Effective noise mitigation techniques are necessary to realize the potential of quantum computations on NISQ devices.

Pauli noise mitigation strategies, such as zero noise extrapolation (ZNE) and probabilistic error cancellation (PEC), have several practical drawbacks. ZNE mitigates Pauli noise by purposely

39th Conference on Neural Information Processing Systems (NeurIPS 2025).

amplifying it, either by stretching gate durations or by adding noisy operations [11]. It then estimates zero-noise results through carefully designed cancellation strategies. This approach relies on precise calibration based on the hardware's noise characteristics. To find the zero-noise result, ZNE runs circuits repeatedly at different noise levels, which introduces overhead for complex circuits or highly stochastic noise profiles. PEC utilizes detailed noise characterization, often by full or partial tomography to construct inverse noise operations [26]. For this reason, it is hard to implement for large systems [5] and highly variable noise profiles.

Van Den Berg's Sparse Pauli Lindblad Method [28] provides an efficient approach to model and mitigate Pauli noise by utilizing sparse representations of Lindblad operators. It puts forth a principled framework for modeling quantum noise by integrating the dynamics of open quantum systems. This approach eliminates the need for resource-intensive full-noise tomography but relies on the careful selection of dominant noise terms, which requires domain expertise. It also presumes stable noise profiles and has limited adaptability to rapidly changing noise environments. This limits its applicability for NISQ hardware, where noise is unpredictable and dynamic. All of these methods aim to mitigate noise while being oblivious to the specific quantum task/ operator.

In this paper, we propose a novel gradient-based framework to mitigate Pauli noise in PQCs addressing the challenges of adaptivity, efficiency, and efficacy. Unlike conventional approaches, *our method tightly integrates the process of learning parameterized inverse noise models, constructed using a sparse Lindblad-based formulation [28], with the optimization of PQCs in the training process.* This simultaneous learning approach allows our framework to adapt dynamically to task-specific objectives and noise characteristics. We present detailed derivations and theoretical foundations of our method, establishing its superiority, and demonstrate its performance in the context of real-world quantum machine learning scenarios. Our findings demonstrate that adaptive noise mitigation improves accuracy compared to state of the art baselines. Our integrated learning approach improves the reliability of quantum machine learning on NISQ hardware.

**Main Contributions:** We propose a joint gradient-based framework that optimizes parameterized quantum circuits (PQCs) while mitigating Pauli noise arising during unitary gate operations, thereby enabling noise-aware training. As with all gradient-based approaches, efficient gradient computation is essential to our method. While gradients w.r.t. PQC parameters can be efficiently estimated using the parameter shift rule [17], estimating gradients for inverse noise parameters is more challenging due to exponential scaling with the qubit count. To address this, we introduce a novel and efficient gradient estimation method tailored to inverse noise parameters enabling scalable and robust optimization of PQCs.

The remainder of the paper is organized as follows: Section 2 introduces the necessary background. Section 2.3 presents our proposed integrated framework for PQC optimization. Section 3.2.1 focuses on gradient derivation while Section 3.2.2 describes our quantum algorithm to estimate gradients of all model parameters, including the inverse noise parameters, a key component of our efficient joint optimization approach. Sample complexity and convergence results are discussed in Section 3.2.3 and Section 3.3, respectively. Numerical results are presented in Section 4, with related work and concluding remarks in Section 5 and Section 6.

## 2 Preliminaries

### 2.1 Parameterized quantum Circuit (PQC)

A PQC is a crucial element in the design of Variational Quantum Algorithms (VQAs). It is represented as a parameterized unitary operator $U_{\overrightarrow{b}}(\overrightarrow{\theta})$ defined by:

$$U_{\overrightarrow{b}}(\overrightarrow{\theta}) = \prod_{l=1}^{L} U_{l,b_l}, \quad U_{l,b_l} = \begin{cases} U_l(\theta_l) & b_l = 1 \\ V_l & o.w. \end{cases} \tag{1}$$

where $U_l(\theta_l) = \exp\{i\theta_l G_l\}$ is a parameterized quantum gate generated by a Pauli string $G_l \in \{I, X, Y, Z\}^{\otimes n}$, $\theta_l$ are learnable parameters, $V_l$ are non-Pauli constant unitaries, and $\overrightarrow{b} \in \{0,1\}^L$ is a constant binary vector.

## 2.2 Noise Model

Pauli noise is a form of quantum noise frequently observed in quantum computing and quantum information processing. It originates from stochastic errors in quantum gate operations, where each qubit in a quantum system may experience a noise process characterized by one of the Pauli operators. This type of noise can be described as a probabilistic application of Paulis— $X$, $Y$ or $Z$ to a qubit, each occurring with a certain probability. Since any Markovian noise can be approximated using Pauli noise via Pauli twirling [8], the dominant noise affecting gate operations can be effectively modeled as Pauli noise. Based on this observation, we adopt the Pauli-Lindblad noise model introduced in [28] to define the noise model over a noise-free quantum state $\rho$ as:

$$\Lambda(\bullet)(\rho) = \left( \bigcirc_{k \in \mathcal{K}} \left( \omega_k I \bullet I + (1 - \omega_k) P_k \bullet P_k^\dagger \right) \right)(\rho)$$
$$\text{where } \omega_k = (1 + \exp\{-2\lambda_k\})/2, \lambda_k \geq 0. \tag{2}$$

Here, $\bullet$ serves as a placeholder indicating that the map is applied to $\rho$, i.e., $\Lambda(\bullet)(\rho) = \Lambda(\rho)$. The notation $\bigcirc_{k \in \mathcal{K}}$ represents the composition of maps where each map corresponds to a noise model term in $\mathcal{K}$ associated with a Pauli string $P_k \in \{I, X, Y, Z\}^{\otimes n}$, and a model coefficient $\lambda_k$. These model terms reflect various noise interactions in the quantum system.

By definition, the non-physical inverse noise model to mitigate the aforementioned noise can be written as:

$$\Lambda^{-1}(\bullet)(\rho) = \left( \gamma \bigcirc_{k \in \mathcal{K}} \left( q_k I \bullet I - (1 - q_k) P_k \bullet P_k^\dagger \right) \right)(\rho)$$
$$\text{where, } q_k = (1 + \exp\{-2\sigma_k\})/2, \gamma = \exp\left\{ 2 \sum_{k \in \mathcal{K}} \sigma_k \right\}, \sigma_k \geq 0. \tag{3}$$

Here, values of $\sigma_k$ correspond to the coefficients of inverse noise model w.r.t. Paulis $P_k$.

*Remark* 2.1. Throughout this paper, the notation for nested map composition $\bigcirc_{j=1}^L f_j(\bullet)$ is defined as

$$f_L \circ f_{L-1} \circ \cdots \circ f_2 \circ f_1(\bullet) = f_L(f_{L-1}(\cdots f_2(f_1(\bullet))))$$

for any collection of maps $f_1, \cdots, f_L$.

## 2.3 Noise-Mitigated PQCs

A noise-mitigated parameterized quantum circuit (nmPQC) is an augmentation of the PQC, as described in Equation (1). It is carefully designed to minimize the adverse impact of noise. In contrast to the conventional design where each layer contains only unitary transformations, each layer of the nmPQC consists of the gate $U_{l,b_l}$, the noise model $\Lambda_l$ and its corresponding inverse noise model $\Lambda_l^{-1}$, as illustrated in Figure 1. With this design, the noise $\Lambda_l$ can be mitigated by the inverse noise operator $\Lambda_l^{-1}$, thereby improving the performance of the PQC.

Formally, an $L$-layer nmPQC, denoted by $\mathcal{U}_R$ is defined as follows:

$$\mathcal{U}_R(\bullet) = \bigcirc_{l=1}^L \Lambda_l^{-1} \circ \Lambda_l \circ \text{Ad}_{U_{l,b_l}}(\bullet) \tag{4}$$

where $\text{Ad}_{U_{l,b_l}}(\bullet) = U_{l,b_l} \bullet U_{l,b_l}^\dagger$, $U_{l,b_l}$ is the $l$-th gate unitary. Additionally, $\Lambda_l$ and $\Lambda_l^{-1}$ represent the noise and inverse noise models at gate $l$, as defined in Equation (2) and Equation (3), respectively. The noise model $\Lambda_l$ is parameterized by unknown noise parameters $\lambda_{l,k}$, while the inverse noise model $\Lambda_l^{-1}$ is parameterized by learnable inverse noise parameters $\sigma_{l,k}$ w.r.t. Paulis $P_k^{(l)}$, associated with each unitary $U_{l,b_l}$.

## 3 Integrated Learning Framework for nmPQC

In this section, we propose a gradient-based framework to train a parameterized quantum circuit on noisy quantum hardware, where the noise is modeled by a Pauli–Lindblad model. We begin by formalizing the learning objective and outlining the assumptions underlying our analysis.

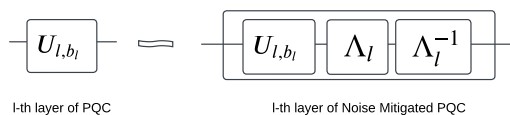

I-th layer of PQC       I-th layer of Noise Mitigated PQC

Figure 1: The diagram above illustrates a single layer of the nmPQC. The complete nmPQC is constructed by concatenating $L$ such blocks of this structure.

## 3.1 Problem Statement and Assumptions

Let $(\rho_t, y_t)$ be elements of the data set $\mathcal{D} = \left\{(\rho_t, y_t) \in \mathcal{B}(\mathbb{C}^N) \times \{-1, 1\}\right\}_{t=1}^{|\mathcal{D}|}$, where $N = 2^n$. We assume that $(\rho_t, y_t)$ are independent and identically distributed (i.i.d.) samples drawn from a known distribution. In order to train the model $\mathcal{U}_R$ defined in Equation (4), we formulate the following optimization problem:

$$\min_{\overrightarrow{\sigma}, \overrightarrow{\theta} \in \mathbb{R}^{\left|(\overrightarrow{\sigma}, \overrightarrow{\theta})\right|}} \mathcal{L}(\overrightarrow{\sigma}, \overrightarrow{\theta}) + \mathcal{G}(\overrightarrow{\sigma}, \overrightarrow{\theta}) \tag{5}$$

where the loss function $\mathcal{L} : \mathbb{R}^{\left|(\overrightarrow{\sigma}, \overrightarrow{\theta})\right|} \to \mathbb{R}$ is defined as :

$$\mathcal{L}(\overrightarrow{\sigma}, \overrightarrow{\theta}) = \frac{1}{|\mathcal{D}|} \sum_{t=1}^{|\mathcal{D}|} \mathcal{L}_t(\overrightarrow{\sigma}, \overrightarrow{\theta}) = \frac{1}{|\mathcal{D}|} \sum_{t=1}^{|\mathcal{D}|} (y_t - \text{tr}(\mathcal{M}\mathcal{U}_R(\rho_t)))^2 \tag{6}$$

which follows a squared loss formulation and $\text{tr}(\mathcal{M}\mathcal{U}_R(\rho_t))$ denotes the predicted measurement outcome corresponding to the observable $\mathcal{M}$. The function $\mathcal{G} : \mathbb{R}^{\left|(\overrightarrow{\sigma}, \overrightarrow{\theta})\right|} \to \mathbb{R} \cup \{+\infty\}$ enforces constraints on the learnable parameters by restricting $\overrightarrow{\sigma}$ and $\overrightarrow{\theta}$ to predefined domains. Specifically, $\mathcal{X}_{sigma}$ constrains the inverse noise parameters $\overrightarrow{\sigma}$ for each gate, while $\mathcal{X}_{theta}$ bounds the variational PQC parameters $\overrightarrow{\theta}$, ensuring they lie within a specified range and preserve parameter feasibility during optimization. The constraint sets are defined by $\mathcal{X}_{sigma} = \prod_{l=1}^{L} \prod_{k \in \mathcal{K}_l} [0, \mathfrak{B}^{(l,k)}]$, $\mathcal{X}_{theta} = \prod_{\substack{l=1 \\ b_l=1}}^{L} [\mathfrak{B}_0^{(l)}, \mathfrak{B}_1^{(l)}]$, $\mathfrak{B}^{(l,k)} \in \mathbb{R}_{\geq 0}$, and $\mathfrak{B}_0^{(l)}, \mathfrak{B}_1^{(l)}$ satisfy $-\infty < \mathfrak{B}_0^{(l)} \leq \mathfrak{B}_1^{(l)} < \infty$.

Using these definitions, $\mathcal{G}(\overrightarrow{\sigma}, \overrightarrow{\theta})$ can be written as:

$$\mathcal{G}(\overrightarrow{\sigma}, \overrightarrow{\theta}) = \begin{cases} 0 & \overrightarrow{\sigma}, \overrightarrow{\theta} \in \mathcal{X}_{sigma} \times \mathcal{X}_{theta} \\ +\infty & o.w. \end{cases} \tag{7}$$

In this framework, the learning task is comprised of: (i) optimizing the standard PQC parameters that define the unitary transformations; and (ii) learning the inverse noise parameters $\sigma_{l,k}$, which mitigate the unknown noise effect of $\Lambda_l$. The PQCs are restricted to $c$-local Pauli operators, ensuring that each unitary $U_{l,b_l}$ acts on at most $c$ qubits. As a result, the number of inverse noise parameters $\sigma_{l,k}$ per unitary is constrained to at most $4^c$, reflecting the overall number of Pauli strings in a $c$-qubit system.

To ensure feasibility and facilitate problem formulation, we make the following assumptions:

**Assumption 3.1.** The noise parameters $\lambda_{l,k}$ are constrained within the domain $\mathcal{X}_{sigma}$, which ensures that each $\lambda_{l,k}$ lies within the interval $[0, \mathfrak{B}^{(l,k)}] \subseteq \mathbb{R}_{\geq 0}$.

**Assumption 3.2.** The single-qubit Paulis, $I$, $X$, $Y$, and $Z$, are considered noiseless.

**Assumption 3.3.** The State Preparation and Measurement (SPAM) noise is absent.

**Assumption 3.4.** $\mathcal{M}$ is Hermitian with eigenvalues 1 and -1.

## 3.2 Gradient and its Estimation

As discussed in the previous section, obtaining the gradient of the objective function is essential for all gradient-based methods. In this subsection, we derive the gradient of the objective function defined in Equation (6) and present a quantum algorithm for its estimation.

### 3.2.1 Derivation of Gradient

First of all, we derive gradients of the per-sample loss function, $\mathcal{L}_t(\overrightarrow{\sigma}, \overrightarrow{\theta}) = (y_t - \operatorname{tr}(\mathcal{M}\mathcal{U}_R(\rho_t)))^2$. Using the parameter shift rule as described in [17], we compute the partial derivative of $\mathcal{L}_t(\overrightarrow{\sigma}, \overrightarrow{\theta})$ w.r.t. the PQC parameter $\theta_j$ (from Appendix A.2.2) as:

$$\frac{\partial \mathcal{L}_t(\overrightarrow{\sigma}, \overrightarrow{\theta})}{\partial \theta_j} = -2\left(y_t - \operatorname{tr}(\mathcal{M}\mathcal{U}_R(\rho_t))\right) \cdot \operatorname{tr}\left(\mathcal{M}\mathcal{U}_{t,>j}\left(\Lambda_j^{-1} \circ \Lambda_j \circ \mathcal{U}_{j,\frac{\pi}{2}}\left(\rho_t^{(j-1)}\right)\right)\right). \quad (8)$$

Here, we denote $\mathcal{U}_{j,a} = \frac{1}{2}\operatorname{Ad}_{U_j(\theta_j+a)} - \frac{1}{2}\operatorname{Ad}_{U_j(\theta_j-a)}$ for $a \in \mathbb{R}$, and introduce the intermediate result $\rho_t^{(j-1)}$ defined by: $\rho_t^{(j-1)} = \left(\bigcirc_{l=1}^{j-1}\Lambda_l^{-1} \circ \Lambda_l \circ \operatorname{Ad}_{U_{l,b_l}}\right)(\rho_t)$.

Next, differentiating $\mathcal{L}_t(\overrightarrow{\sigma}, \overrightarrow{\theta})$ w.r.t. an inverse noise parameter $\sigma_{j,q}$ yields (see Appendix A.2.1):

$$\frac{\partial \mathcal{L}_t(\overrightarrow{\sigma}, \overrightarrow{\theta})}{\partial \sigma_{j,q}} = -4\left(y_t - \operatorname{tr}(\mathcal{M}\mathcal{U}_R(\rho_t))\right) \cdot \operatorname{tr}\left(\mathcal{M}\mathcal{U}_{t,>j}\left(\Lambda_j^{-1}\left(\sum_{g \in \mathbf{G}_q^{(j)}} \alpha_{j,g}g\right)\right)\right). \quad (9)$$

where $\mathbf{G}_q^{(j)} = \left\{g \in \{I, X, Y, Z\}^{\otimes n} \middle| \left\{P_q^{(j)}, g\right\} = 0\right\}$ is the subset of Pauli strings that anti-commute with $P_q^{(j)}$, and the terms within the innermost parentheses arise from the Pauli decomposition of the Hermitian matrix $\rho_{t,j} = \Lambda_j \circ \operatorname{Ad}_{U_{j,b_j}} \circ \left(\bigcirc_{l=1}^{j-1}\Lambda_l^{-1} \circ \Lambda_l \circ \operatorname{Ad}_{U_{l,b_l}}\right)(\rho_t)$. Plugging in the identity stated in Proposition A.14, we simplify further to arrive at:

$$\frac{\partial \mathcal{L}_t(\overrightarrow{\sigma}, \overrightarrow{\theta})}{\partial \sigma_{j,q}} = -4\left(y_t - \operatorname{tr}(\mathcal{M}\mathcal{U}_R(\rho_t))\right) \cdot \operatorname{tr}\left(\mathcal{M}\mathcal{U}_{t,>j}\left(\Lambda_j^{-1} \circ \mathcal{U}_{P_q^{(j)}}(\rho_{t,j})\right)\right), \quad (10)$$

where $\mathcal{U}_{P_q^{(j)}}(\bullet) = \frac{1}{2}\left(I \bullet I - P_q^{(j)} \bullet (P_q^{(j)})^\dagger\right)$.

### 3.2.2 Universal Estimation Algorithm

By examining equations Equation (8) and Equation (10), it becomes clear that the primary computational challenge in evaluating these partial derivatives arises from estimating the trace terms. These trace terms can be categorized into three distinct types: $\operatorname{tr}(\mathcal{M}\mathcal{U}_R(\rho_t))$, $\operatorname{tr}\left(\mathcal{M}\mathcal{U}_{t,>j}\left(\Lambda_j^{-1} \circ \operatorname{Ad}_{P_q^{(j)}}(\rho_{t,j})\right)\right)$, and $\operatorname{tr}\left(\mathcal{M}\mathcal{U}_{t,>j}\left(\operatorname{Ad}_{U(\theta_j+a)}(\rho_{t,j})\right)\right)$. To facilitate the estimation of the relevant trace terms, we reduce each one to a standard form and present the derivations in Equation (28), Equation (29), and Equation (30). Upon analysis of these expressions, we observe a common structural form across them:

$$\operatorname{tr}\left(\mathcal{M}\bigcirc_{l=1}^{L}\Lambda_l^{-1} \circ \operatorname{Ad}_{W_l} \circ \Lambda_l \circ \operatorname{Ad}_{U_l}(\rho)\right), \quad (11)$$

where $W_l$ is a Pauli string and $U_l$ is an unitary. Building on this observation, we introduce the universal estimation algorithm, described in Algorithm 2, designed to unbiasedly approximate the expression in Equation (11) (see Lemma A.16 for details). The corresponding quantum circuit implementation is illustrated in Figure 2.

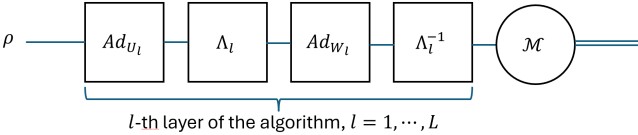

$l$-th layer of the algorithm, $l = 1, \cdots, L$

Figure 2: (1) The quantum circuit implementing the universal estimation algorithm: operator $W_l$ corresponds to a fixed Pauli string, while $U_l$ are constant or parameterized (variational) $c$-local non-Pauli quantum operations. (2) Note that the inverse channels $\Lambda_l^{-1}$ are non-physical and cannot be directly realized on quantum hardware. Consequently, the implementation of the circuit requires classical preprocessing. Further details are provided in Appendix A.3.1.

Importantly, this quantum-classical hybrid algorithm enables unbiased estimation of these trace terms on noisy quantum hardware without requiring explicit acquisition of the structure of noise channels $\Lambda_l$, which is typically computationally expensive.

### 3.2.3 Sample Complexity

The universal estimation algorithm serves as the fundamental building block for the partial derivative estimation procedures defined in Algorithm 3 and Algorithm 4, corresponding to the partial derivatives of $\mathcal{L}_t$ given in Equation (8) and Equation (10). In essence, the universal estimation algorithm is executed repeatedly to obtain unbiased approximations of the quantities defined in Equation (8) and Equation (10), up to a prescribed accuracy (see Proposition A.17 for details). Ultimately, by iteratively applying Algorithm 3 and Algorithm 4, we obtain an approximation of the gradient of $\mathcal{L}_t$ at a given point $(\overrightarrow{\theta}, \overrightarrow{\sigma})$.

A natural question arises regarding sample complexity: how many measurements are required to obtain a reliable estimate of the gradient at $(\overrightarrow{\sigma}, \overrightarrow{\theta})$? Referring to Lemma A.22, we observe that, with probability at least $1 - \delta$, the number of measurements (or number of runs) required to estimate a single partial derivative within an additive error $\varepsilon$ is no less than $\frac{32^2}{2\left((\Gamma_{\overrightarrow{\sigma}}^{-1})^2 \cdot \varepsilon\right)^2} \ln\left(\frac{2\left|(\overrightarrow{\sigma}, \overrightarrow{\theta})\right|}{\delta}\right)$, where $\Gamma_{\overrightarrow{\sigma}} = \exp\left\{2 \sum_{l=1}^{L} \sum_{k \in \mathcal{K}_l} \sigma_{l,k}\right\}$. This expression shows that the sample complexity (or the measurement complexity) can grow rapidly if $\varepsilon$ or $\delta$ is too small or if the number of parameters $\left|(\overrightarrow{\sigma}, \overrightarrow{\theta})\right|$ becomes large, making scalability a significant challenge.

To address the scalability challenge, we incorporate two key techniques. First, in each update round, we randomly select a subset of coordinate directions to update, where each direction is independently chosen with probability $1/p$ for some integer $p \geq 1$ (see Algorithm 5). This sampling strategy reduces the number of estimations required per iteration. The resulting sample complexity is summarized in the following theorem:

**Theorem 3.1** (Informal Version of Theorem A.23). *Under Assumption 3.2 - Assumption 3.4, The expected sample complexity (measurement complexity) required for Algorithm 5 to obtain a estimation of the gradient with additive error $\varepsilon > 0$ with probability at least $1 - \delta$, where $\delta \in (0, 1)$ is at least*

$$\frac{\left|(\overrightarrow{\sigma}, \overrightarrow{\theta})\right|}{p} \cdot \frac{32^2}{2\left((\Gamma_{\overrightarrow{\sigma}}^{-1})^2 \cdot \varepsilon\right)^2} \ln\left(\frac{2\left|(\overrightarrow{\sigma}, \overrightarrow{\theta})\right|}{\delta p}\right)$$

Second, when the current parameters $(\overrightarrow{\sigma}, \overrightarrow{\theta})$ are far from a stationary point, we relax the estimation accuracy to prioritize correct gradient directions over high precision. Concretely, we set the error tolerance to $\varepsilon = \Gamma_{\overrightarrow{\sigma}}^2 \kappa$ for some $\kappa \in (0, 1)$. In particular, by applying the bound established in Theorem 3.1, the expected total number of measurements per update round is at least $\frac{\left|(\overrightarrow{\sigma}, \overrightarrow{\theta})\right|}{p} \cdot \frac{32^2}{2\kappa^2} \ln\left(\frac{2\left|(\overrightarrow{\sigma}, \overrightarrow{\theta})\right|}{\delta p}\right)$. This adaptive strategy ensures that early-stage updates remain computationally efficient while preserving convergence behavior.

### 3.3 Optimization and Convergence

To train the model introduced in Equation (4), we consider the optimization problem defined in Equation (5). Specifically, we optimize the objective function in Equation (5) using the proximal stochastic gradient descent (proximal SGD) method, as described in Algorithm 1. Unlike standard SGD, proximal SGD updates the parameters according to the rule

$$(\overrightarrow{\sigma}^{(t+1)}, \overrightarrow{\theta}^{(t+1)}) \leftarrow \text{prox}_{\mathcal{G}}\left((\overrightarrow{\sigma}^{(t)}, \overrightarrow{\theta}^{(t)}) - \eta^{(t)} \widetilde{\nabla}_{\overrightarrow{\sigma}, \overrightarrow{\theta}}^{(t)} \mathcal{L}_{i_t}\right), \tag{12}$$

where $\widetilde{\nabla}_{\overrightarrow{\sigma}, \overrightarrow{\theta}}^{(t)} \mathcal{L}_{i_t}$ denotes the estimated gradient of $\mathcal{L}_{i_t}$, generated by Algorithm 5 at the point $(\overrightarrow{\sigma}, \overrightarrow{\theta})$ for a randomly selected index $i_t \in [|\mathcal{D}|]$, and $\text{prox}_{\mathcal{G}}$ denotes the proximal operator associated with $\mathcal{G}$.

The proximal operator is defined as

$$\text{prox}_{\mathcal{G}}(\sigma^{(t)}, \theta^{(t)}) = \underset{(\sigma^{(t')}, \theta^{(t')}) \in \mathcal{X}}{\text{argmin}} \left( \mathcal{G}(\sigma^{(t')}, \theta^{(t')}) + \frac{1}{2} \|(\sigma^{(t')}, \theta^{(t')}) - (\sigma^{(t)}, \theta^{(t)})\|^2 \right).$$

where $\mathcal{X} \equiv \mathcal{X}_{sigma} \times \mathcal{X}_{theta}$ is the search space. This construction ensures that every element of the sequence $\left\{ (\overrightarrow{\sigma}^{(t)}, \overrightarrow{\theta}^{(t)}) \right\}_{t=1}^{T}$ generated by the update rule in Equation (12) remains within $\mathcal{X}$, guaranteeing feasibility throughout the optimization process.

---

**Algorithm 1** Learning Algorithm

---

**Dataset:** $\mathcal{D} = \{(\rho_t, y_t)\}$

**Input:** nmPQC (number of layers $L$, $\overrightarrow{b} \in \mathbb{R}^L$), Measurement $\mathcal{M}$, Total iterations T, Learning rate schedule $\{\eta^{(t)}\}_{t=1}^{T}$, Number of shots $\widetilde{N} \in (2\mathbb{Z})_{\geq 0}$, and positive integer $p$.

{The non-increasing learning rate schedule $\eta^{(t)}$ is such that $\eta^{(t)} \in (0, 1)$, and $\sum_{t=0}^{T} \eta^{(t)}$ is finite}

**Initialization:** $\forall l \in \left[\left\| \overrightarrow{b} \right\|_0\right]$, $\theta_l \sim \text{Uniform}(\mathfrak{B}_0^{(l)}, \mathfrak{B}_1^{(l)})$; $\forall l \in [L]$, $k \in \mathcal{K}_l$, $\sigma_{l,k} \leftarrow 0$.

**for** $t = 1, \dots, T$ **do**

    $i_t \sim \text{uniform}(1, \dots, |\mathcal{D}|)$ {Sample a random index}

    $\overrightarrow{p}_t \sim (\text{Bernoulli}(1/p))^{\left|(\overrightarrow{\sigma}, \overrightarrow{\theta})\right|}$

    $\widetilde{\nabla}_{\overrightarrow{\sigma}, \overrightarrow{\theta}}^{(t)} \mathcal{L}_{i_t} \leftarrow \texttt{gradient\_estimator}\left(i_t, M, \overrightarrow{\sigma}^{(t)}, \overrightarrow{\theta}^{(t)}, y_{i_t}, \widetilde{N}, \overrightarrow{p}_t\right)$     {Algorithm 5}

    $(\overrightarrow{\sigma}^{(t+1)}, \overrightarrow{\theta}^{(t+1)}) \leftarrow \text{prox}_{\mathcal{G}}\left((\overrightarrow{\sigma}^{(t)}, \overrightarrow{\theta}^{(t)}) - \eta^{(t)} \widetilde{\nabla}_{\overrightarrow{\sigma}, \overrightarrow{\theta}}^{(t)} \mathcal{L}_{i_t}\right)$

**end for**

---

We now analyze the convergence of the learning algorithm defined in Algorithm 1. To begin, we present several key properties of the objective function and the estimated gradient. First, in Lemma A.28, we show that the function $\mathcal{L}$ is locally Lipschitz continuous over the compact set $\mathcal{X}$, a key property for ensuring convergence. Next, in Lemma A.29, we demonstrate that $\mathcal{L} + \mathcal{G}$ is lower bounded, thereby ensuring the existence of an optimizer. Finally, in Proposition A.30, we prove that the estimated gradient produced by Algorithm 5 satisfies the bounded variance condition, which is a necessary requirement for bounded convergence behavior. Building on these properties, we summarize the convergence rate of the learning algorithm in the following theorem:

**Theorem 3.2** (Informal Version of Theorem A.31). *Under Assumption 3.1, for the non-convex objective function $\mathcal{L} + \mathcal{G}$ defined in Equation* (6)*, the convergence rate of the Algorithm 1 satisfies*

$$\left( \frac{1}{\sum_{t=1}^{T} \eta_t} \sum_{t=1}^{T} \eta_t \mathbb{E}\left[\mathcal{L}(w_t)\right] \right) - \mathcal{L}(\overline{w}) \leq \frac{p \cdot \|w_0 - \overline{w}\|_2^2}{2\sum_{t=1}^{T} \eta_t} + \frac{3L_{\mathcal{X}} \sum_{t=1}^{T} \eta_t \mathfrak{B}_{\max}}{2\sum_{t=1}^{T} \eta_t} + \frac{p \sum_{t=1}^{T} \eta_t^2 \mathcal{V}}{2\sum_{t=1}^{T} \eta_t},$$

*where $w_0 = \left(\overrightarrow{\sigma}^{(0)}, \overrightarrow{\theta}^{(0)}\right)$ is the initial point, $\overline{w} = \overline{(\overrightarrow{\sigma}, \overrightarrow{\theta})}$ denotes a stationary point, and $w_t = (\overrightarrow{\sigma}^{(t)}, \overrightarrow{\theta}^{(t)})$ are the iterates generated by proximal SGD. The step size $\eta^{(t)} \in (0, 1)$ forms a monotonically decreasing sequence such that $\sum_{t=0}^{\infty} \eta_t < \infty$. The constants $\mathcal{V}$ and $\mathfrak{B}_{\max}$ are defined in Theorem A.31.*

*Remark* 3.3. The factor $p$ in the bound above arises from the random selection strategy employed in Algorithm 5. A larger value of $p$ leads to a sparser estimated gradient $\widetilde{\nabla}_{\overrightarrow{\sigma}, \overrightarrow{\theta}}^{(t)} \mathcal{L}_{i_t}$, which in turn results in a slower convergence rate.

## 4 Numerical Results

### 4.1 Experimental Setup and Evaluation

We evaluate our learning algorithm (Algorithm 1) on a binary classification task on the standard MNIST dataset as in prior related efforts [3, 30], focusing on digits 3 and 6. The data is preprocessed via principal component analysis (PCA) to reduce dimensionality, then normalized and encoded into

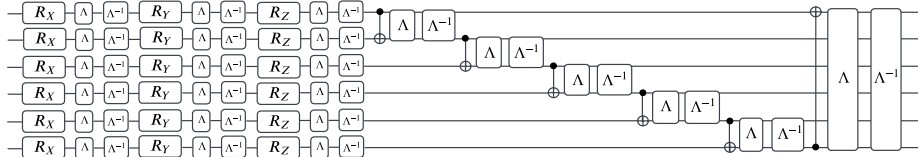

Figure 3: A sample HEA layer features a 6-qubit $R_X + R_Y + R_Z + \text{CNOT}$ design. Here, $\Lambda$ and $\Lambda^{-1}$ are gate-specific; one-qubit noise and its inverse are applied to $R_X$, $R_Y$ and $R_Z$, while two-qubit noise models are used for the non-parameterized CNOT gates.

quantum states using amplitude encoding, with binary labels: $-1$ for digit 3 and 1 for digit 6. A total of 5000 input-label pairs are generated, which are randomly shuffled and split into 80% for training and 20% for testing. To isolate training-time noise and mitigation, we directly initialize the simulator to the amplitude-encoded PCA state (i.e., no explicit state-preparation circuit). For the nmPQC, we employ a noisy 6-qubit hardware-efficient ansatz (HEA), as illustrated in Figure 3, consisting of two layers of parameterized single-qubit rotation gates $R_X$, $R_Y$, and $R_Z$, combined with circular entangling CNOT gates as the base PQC. At the end of the circuit, all qubits are measured in $Z$-basis. We determine the initial learning rate, $\eta^{(1)}$, via a coarse-to-fine search: a wide log-spaced grid sweep to localize a promising interval, followed by a binary search within that interval to refine the value. During training, we maintain a monotonically decreasing learning-rate schedule. The framework is simulated on a multi-core CPU computing cluster, and the simulation is implemented using Qiskit [10].

To assess the effectiveness of the proposed mitigation framework which augments the PQC with nmPQC (as in Figure 3), we compare it with the following configurations: (1) A noiseless PQC, serving as a best-case performance benchmark. (2) A noisy PQC without mitigation, serving as a baseline. (3) A mitigated PQC using a probabilistic quantum noise model, adapted from Van Den Berg et al. [28]. We pre-estimate noise parameters using their procedure and fix them as inverse noise parameters during training. While their method is not originally intended for variational optimization, we adapt it as a benchmark, referring to it as "Van Den Berg et al.'s mitigation approach". For training we use proximal SGD to optimize the model parameters along with the inverse noise parameter(for our method). Model performance is evaluated by accuracy, while noise mitigation efficacy is assessed using mean squared error (MSE) over training epochs.

## 4.2 Experimental Results

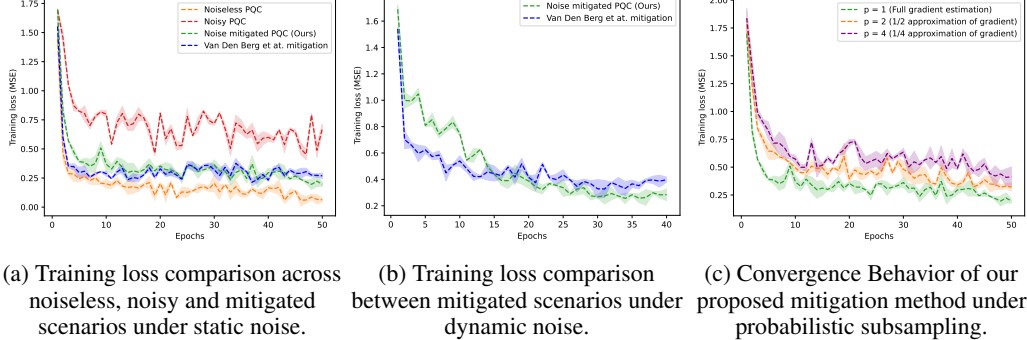

(a) Training loss comparison across noiseless, noisy and mitigated scenarios under static noise.

(b) Training loss comparison between mitigated scenarios under dynamic noise.

(c) Convergence Behavior of our proposed mitigation method under probabilistic subsampling.

Figure 4: Training performance of the binary classification using MNIST dataset in the quantum noise context. Each epoch summarizes 50 iterations of the training process.

We evaluate the effectiveness of our proposed method by comparing its training loss and testing accuracy against alternative configurations in the trivial scenario (i.e., full gradient estimation). Each experiment is repeated three times per setting, and the standard deviation is used to report variability. Figure 4a shows the convergence behavior of different methods over epochs under a static noise setting, where all CNOT gates in the noisy and mitigated models are assigned fixed error rates. The

noiseless model serves as a best-case reference. As expected, the noiseless PQC achieves the lowest training loss. The noisy PQC without any mitigation mechanism exhibits significantly higher loss, highlighting the effects of quantum noise. Both mitigation strategies– our proposed method and that of Van Den Berg et al.[28] achieve an improved reduction in training loss compared to the noisy baseline. The Van Den Berg et al. approach pre-estimates noise models using the random benchmarking technique [9] and applies their inverses using the learned noise parameters during training, yielding an initial advantage. In contrast, our method learns the inverse noise parameters during training, eliminating the need for noise pre-characterization and calibration, and enabling streamlined, task-specific adaptive mitigation. We observe that Van Den Berg et al.'s approach achieves lower training loss initially, but as training continues, our method surpasses it, ultimately achieving slightly better overall performance. Table 1 summarizes the experimental accuracies of different approaches, showing that our method outperforms the baseline noisy PQC and Van Den Berg et al.'s.

To evaluate adaptability, we simulate dynamic noise by slightly increasing error rates of three CNOT gates shortly after training begins for mitigated methods only. While Van Den Berg et al.'s approach continues using its static pre-estimated noise model, our method dynamically updates the inverse noise parameters during training. As shown in Figure 4b, our method effectively learns the inverse noise parameters in response to dynamic changes and ultimately achieves lower MSE than the method of Van Den Berg et al.

Table 1: Classification accuracies across Different methods

|  | ACCURACY |
| --- | --- |
| NOISELESS PQC | $94.13 \pm 0.31$ |
| NOISY PQC (BASELINE) | $79.69 \pm 3.70$ |
| MITIGATED PQC (VAN DEN BERG ET.AL) | $85.37 \pm 3.44$ |
| **OURS** | $86.43 \pm 3.14$ |

In addition, we investigate the trade-off between gradient estimation cost and optimization performance by evaluating the convergence behavior of our learning algorithm (Algorithm 1) under probabilistic subsampling (see Appendix A.3.3). This experiment uses the same static noise profile as in Figure 4a. In this setting, for a fixed parameter $p$, each coordinate direction (i.e., parameter) is selected independently with probability $\frac{1}{p}$ during each gradient update step. The case $p = 1$ corresponds to full gradient estimation, while larger $p$ values introduce sparsity in the gradient estimates.

Figure 4c depicts the convergence behavior for various $p$ values over 50 epochs. For $p = 1$, we observe faster and smoother convergence, with a steady decline in loss and minimal fluctuations, demonstrating the benefit of full gradient estimation. As $p$ increases, convergence slows, and the loss curve becomes noisier with higher overall loss levels, reflecting the impact of sparser gradient updates.

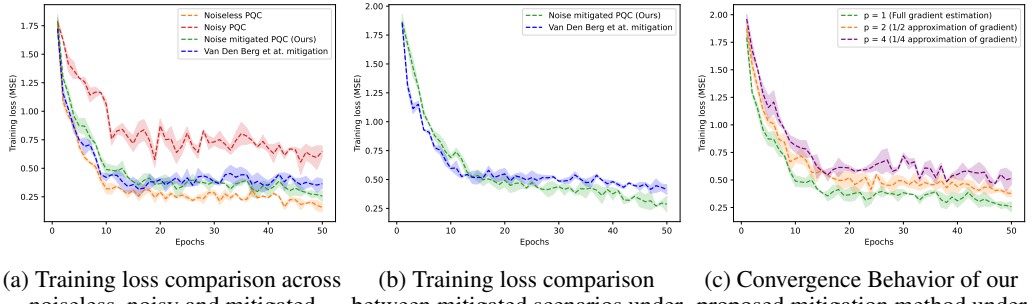

(a) Training loss comparison across noiseless, noisy and mitigated scenarios under static noise.

(b) Training loss comparison between mitigated scenarios under dynamic noise.

(c) Convergence Behavior of our proposed mitigation method under probabilistic subsampling.

Figure 5: Training performance of the binary classification using Fashion-MNIST dataset (Pullover vs Shirt) in the quantum noise context. Each epoch summarizes 50 iterations of the training process.

**Additional results.** We repeat the same protocol on the Fashion-MNIST dataset (Pullover vs. Shirt) [32]; trends mirror MNIST under static (Figure 5a) and dynamic (Figure 5b) noise, and probabilistic subsampling behaves similarly (Figure 5c). Additional experimental details and the accuracy table appear in Appendix A.6.1. Furthermore, under the same dynamic-noise protocol across both datasets, we compare PEC and ZNE against our method (see Appendix A.6.2); in our experiments, our framework adapts online and attains the best final metric (lower loss/MSE).

## 5 Related Work

PQCs are central to many quantum machine learning models, particularly in VQAs [2], originally introduced for ground state preparation [21]. VQAs adopt a hybrid quantum-classical loop, where a quantum circuit evaluates an objective and a classical optimizer adjusts parameters. This framework supports applications such as quantum chemistry [15], combinatorial optimization [6], and quantum neural networks (QNNs) [24, 23, 1].

Despite their flexibility, PQCs are susceptible to quantum noise, including decoherence, gate noise, and SPAM errors [22]. Quantum Error Correction (QEC) offers robust protection [27], but its overhead remains prohibitive for NISQ devices. Instead, Quantum Error Mitigation (QEM) methods such as ZNE [14, 26] aim to reduce noise without heavy resource demands.

Pauli noise, a common model of gate errors, can exponentially suppress gradients in VQAs, impeding training [8, 31]. Several approaches have emerged to address this. PEC techniques model noise as an invertible stochastic process, correcting it via probabilistic sampling [26]. Learning-based strategies such as those by Strikis et al. [25] and Czarnik et al. [4] train on auxiliary circuits to estimate mitigated expectations. Van den Berg et al. [28] propose incorporating Lindblad-modeled noise directly into the optimization. A recent diffusion-inspired approach [19] mitigates noise via learned forward-backward dynamics.

## 6 Discussions

We propose a gradient-based framework for jointly learning parameterized quantum circuits (PQCs) and inverse noise operators, enabling noise parameter estimation during training. This approach supports rapid recalibration when the noise model changes, thereby avoiding full retraining and improving adaptability on near-term devices. A parameter-shift–like rule is introduced for scalable, hardware-efficient gradient estimation with respect to inverse noise parameters, ensuring robustness under Pauli noise. While effective, the method faces challenges: parameter growth under low locality (e.g., $c > \Omega(\log n)$) can increase computational cost, and the reliance on simple measurement may limit broader applicability. Future work will focus on reducing the complexity of inverse noise operators and extending applicability to general measurement schemes.

## Acknowledgments

This work is partially supported by the NSF Center for Science of Information (CSoI) Grant CCF-0939370, and also by NSF Grants CCF-2006440 and CCF-2211423.

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

# A Appendix

In the following appendices, we provide additional technical details that complement the findings presented in the main manuscript. In Appendix A.1, we offer a concise overview of key concepts in quantum computing and properties of the Pauli-Lindblad noise model. In Appendix A.2, we present a detailed derivation of the gradient of the loss function. In Appendix A.3, we describe the algorithms used for gradient estimation. In Appendix A.4, we discuss the sample complexity of the quantum algorithm. Finally, in Appendix A.5, we characterize the convergence behavior of the proposed quantum learning approach and provide an analysis of its performance.

## A.1 Preliminary

### A.1.1 Overview

We present a brief overview of some key concepts and definitions in quantum computing.

**Definition A.1** (Space of Linear Operators). If $\mathcal{H}$ be a finite-dimensional Hilbert space, then let $\mathcal{B}(\mathcal{H})$ be the set of all linear operators on $\mathcal{H}$, that is:

$$\mathcal{B}(\mathcal{H}) = \{A : \mathcal{H} \to \mathcal{H}\}.$$

*Remark* A.2. In this paper, we assume $\mathcal{H} = \mathbb{C}^N$, where $N = 2^n$ and $n$ is the number of qubits in the system. In other words, $\mathcal{B}(\mathbb{C}^N)$ is the set of all linear operators from $\mathbb{C}^N$ to $\mathbb{C}^N$.

**Definition A.3** (c-local). If a Hamiltonian $H$ is supported on at most $c$ of the $n \geq c$ qubits (i.e., $H$ acts on qubits in $S \subseteq [n]$, where $|S| = c$), it is called a $c$-**local** Hamiltonian (on $n$ qubits).

*Remark* A.4. In this paper, we say the locality is high when $c = O(\log n)$, where $n$ is the number of qubits.

**Theorem A.5** (Quantum Channel [20]). *A map $\mathcal{E} : \mathcal{B}(\mathcal{H}) \to \mathcal{B}(\mathcal{H})$ is a quantum channel if and only if $\mathcal{E}$ satisfies the following three conditions:*

1. *(**Trace Preserving**) $tr(\mathcal{E}(\rho)) = 1$ for all mixed states $\rho$ on $\mathcal{H}$.*

2. *(**Convex**) $\mathcal{E}(\sum_i p_i \rho_i) = \sum_i p_i \mathcal{E}(\rho_i)$, where $\{p_i\}$ is a probability distribution.*

3. *(**Completely Positive**) For all positive Hermitian operators $A \in \mathcal{B}(\mathcal{H})$, $\mathcal{E}(A)$ is also a positive operator.*

### A.1.2 Pauli Matrices and Pauli Strings

The Pauli matrices $I, X, Y, Z$ are defined as the following 2-by-2 traceless complex matrices:

$$I = \begin{pmatrix} 1 & 0 \\ 0 & 1 \end{pmatrix}, \quad X = \begin{pmatrix} 0 & 1 \\ 1 & 0 \end{pmatrix}, \quad Y = \begin{pmatrix} 0 & -i \\ i & 0 \end{pmatrix}, \quad Z = \begin{pmatrix} 1 & 0 \\ 0 & -1 \end{pmatrix}.$$

A Pauli string $P$ of length $n$ is an element in $\{I, X, Y, Z\}^{\otimes n}$. Since $I = 1 \cdot (X)^0(Z)^0$, $X = 1 \cdot (X)^1(Z)^0$, $Y = i \cdot (X)^1(Z)^1$, and $Z = 1 \cdot (X)^0(Z)^1$, any Pauli string $P \in \{I, X, Y, Z\}^{\otimes n}$ can be uniquely expressed as follows:

$$P = \xi \bigotimes_{j=1}^{n} (X)^{\overrightarrow{x}(j)}(Z)^{\overrightarrow{z}(j)},$$

where $\xi \in \{\pm 1, \pm i\}$, $\overrightarrow{x}, \overrightarrow{z} \in \{0,1\}^n$ are binary vectors, and $\overrightarrow{x}(j)$ ($\overrightarrow{z}(j)$) represents the $j$-th component of the vector $\overrightarrow{x}$ ($\overrightarrow{z}$). Given this observation, we can now define the symplectic inner product between two Pauli strings.

**Definition A.6** (Symplectic Inner Product (Less General)). If $P_1 = \xi_1 \bigotimes_{j=1}^{n} (X)^{\overrightarrow{x}_1(j)}(Z)^{\overrightarrow{z}_1(j)}$ and $P_2 = \xi_2 \bigotimes_{j=1}^{n} (X)^{\overrightarrow{x}_2(j)}(Z)^{\overrightarrow{z}_2(j)}$ are any two Pauli strings, the symplectic inner product between $P_1$ and $P_2$, denoted by $(P_1, P_2)_{sp}$ is defined by:

$$(P_1, P_2)_{sp} = (\overrightarrow{x}_1 \cdot \overrightarrow{z}_2 + \overrightarrow{z}_1 \cdot \overrightarrow{x}_2) \mod 2.$$

The following proposition is a direct application of Definition A.6.

**Proposition A.7.** *Let $P_1 = \xi_1 \bigotimes_{j=1}^{n}(X)^{\vec{x}_1(j)}(Z)^{\vec{z}_1(j)}$ and $P_2 = \xi_2 \bigotimes_{j=1}^{n}(X)^{\vec{x}_2(j)}(Z)^{\vec{z}_2(j)}$ be any two Pauli strings. Then,*

$$P_1 P_2 = (-1)^{(P_1,P_2)_{sp}} P_2 P_1.$$

*Sketch Proof of Proposition A.7.* First, we note that for any $b_1, b_2 \in \{0,1\}$,

$$(X)^{b_1}(Z)^{b_2} = (-1)^{b_1 b_2}(Z)^{b_2}(X)^{b_1}$$

Using the properties of Pauli matrices and Kronecker product, we obtain

$$P_1 P_2$$

$$= \left( \xi_1 \bigotimes_{j=1}^{n}(X)^{\vec{x}_1(j)}(Z)^{\vec{z}_1(j)} \right) \left( \xi_2 \bigotimes_{j=1}^{n}(X)^{\vec{x}_2(j)}(Z)^{\vec{z}_2(j)} \right)$$

$$= \xi_1 \xi_2 \bigotimes_{j=1}^{n}(X)^{\vec{x}_1(j)}(Z)^{\vec{z}_1(j)}(X)^{\vec{x}_2(j)}(Z)^{\vec{z}_2(j)}$$

$$= \xi_1 \xi_2 \bigotimes_{j=1}^{n}(-1)^{\vec{z}_1(j)\cdot\vec{x}_2(j)} \cdot (-1)^{\vec{x}_1(j)\cdot\vec{z}_2(j)}(X)^{\vec{x}_2(j)}(Z)^{\vec{z}_2(j)}(X)^{\vec{x}_1(j)}(Z)^{\vec{z}_1(j)}$$

$$= \xi_1 \xi_2 (-1)^{\sum_{j=1}^{n}\vec{z}_1(j)\cdot\vec{x}_2(j)} \cdot (-1)^{\sum_{j=1}^{n}\vec{x}_1(j)\cdot\vec{z}_2(j)} \bigotimes_{j=1}^{n}(X)^{\vec{x}_2(j)}(Z)^{\vec{z}_2(j)}(X)^{\vec{x}_1(j)}(Z)^{\vec{z}_1(j)}$$

$$= \xi_1 \xi_2 (-1)^{\vec{z}_1\cdot\vec{x}_2+\vec{x}_1\cdot\vec{z}_2} \bigotimes_{j=1}^{n}(X)^{\vec{x}_2(j)}(Z)^{\vec{z}_2(j)}(X)^{\vec{x}_1(j)}(Z)^{\vec{z}_1(j)}$$

$$= \xi_1 \xi_2 (-1)^{(\vec{z}_1\cdot\vec{x}_2+\vec{x}_1\cdot\vec{z}_2) \mod 2} \bigotimes_{j=1}^{n}(X)^{\vec{x}_2(j)}(Z)^{\vec{z}_2(j)}(X)^{\vec{x}_1(j)}(Z)^{\vec{z}_1(j)}$$

$$= (-1)^{(P_1,P_2)_{sp}} P_2 P_1.$$

$$\square$$

The following corollary follows immediately from Proposition A.7.

**Corollary A.8.** *Two Pauli strings commute if and only if their symplectic inner product is 0.*

### A.1.3 Pauli-Lindblad Noise Model

As suggested in [28], the Pauli-Lindblad equation with no internal Hamiltonian dynamics can be written as:

$$\frac{\partial}{\partial t}\rho(t) = \mathcal{L}\rho(t), \quad \text{where } \mathcal{L}(\rho) = \sum_{k\in\mathcal{K}} \lambda_k (P_k \rho P_k - \rho) \tag{13}$$

and the noise model derived from Equation (13) can be expressed as the following quantum channel:

$$\Lambda(\rho) = \bigcirc_{k\in\mathcal{K}} \left( \omega_k \rho + (1-\omega_k)P_k \rho P_k^{\dagger} \right)$$

for some $\mathcal{K} \subseteq [4^n - 1]$, where $\omega_k = (1 + \exp\{-2\lambda_k\})/2$. To construct the inverse channel, [28] used the probabilistic error cancellation (PEC) protocol. The PEC approach allows for the application of an inverse noise channel to correct the errors introduced by the noise channel. The inverse channel is typically constructed by taking a probabilistic approach to undo the effects of the noise. The inverse noise model is by definition is a non physical map and expressed as:

$$\Lambda^{-1}(\rho) = \gamma \bigcirc_{k\in\mathcal{K}} \left( \omega_k \rho - (1-\omega_k)P_k \rho P_k^{\dagger} \right)$$

with $\gamma = \exp\left\{\sum_{k \in \mathcal{K}} 2\lambda_k\right\}$ represents sampling overhead. For each $k \in \mathcal{K}$ the identity is sampled with probability $\omega_k$ or the Pauli $P_k$ is applied. The number of non-identity Paulis are counted and the final Paulis are computed as the product of sampled terms. This process is repeated for all layers and the corresponding number of non-identity Paulis is recorded along with the sampling overhead. For layer $i = 1 \cdots L$, let the respective count be $m_i$ and overhead be $\gamma_i$, then the final measurement outcome is multiplied by $\prod_{i=1}^{L} (-1)^{m_i} \gamma_i$ to get an unbiased expectation value on average.

For our purpose, we reiterate some useful properties of the operator $(aI \bullet I + bP_k \bullet P_k^\dagger)$ for any Pauli string $P_k$ and $a, b \in \mathbb{R}$.

**Proposition A.9.** *Let $P_k$ be any Pauli string. Then, $(aI \bullet I + bP_k \bullet P_k^\dagger)$ is linear.*

*Sketch Proof of Proposition A.9.* Let $A, B \in \mathcal{B}(\mathcal{H})$ be any two linear operators. Let $c, d \in \mathbb{R}$. Then,

$$
\begin{aligned}
&(aI \bullet I + bP_k \bullet P_k^\dagger)(cA + dB) \\
=&aI(cA + dB)I + bP_k(cA + dB)P_k^\dagger \\
=&caIAI + cbP_kAP_k^\dagger + daIBI + dbP_kBP_k^\dagger \\
=&c(aI \bullet I + bP_k \bullet P_k^\dagger)(A) + d(aI \bullet I + bP_k \bullet P_k^\dagger)(B).
\end{aligned}
$$

$\square$

**Proposition A.10.** *Let $P_{k_1}, P_{k_2}$ be any two Pauli strings. Let $a, b, c, d \in \mathbb{R}$ be any real values. Then, $aI \bullet I + bP_{k_1} \bullet P_{k_1}^\dagger$ and $cI \bullet I + dP_{k_2} \bullet P_{k_2}^\dagger$ commute.*

*Sketch Proof of Proposition A.10.* The proof is basically the same as the one used in [28]. First, we write $aI \bullet I + bP_{k_1} \bullet P_{k_1}$ and $cI \bullet I + dP_{k_2} \bullet P_{k_2}$ in their matrix form as the follows:

$$
aI \otimes I + bP_{k_1} \otimes (P_{k_1}^\dagger)^T,
$$

and

$$
cI \otimes I + dP_{k_2} \otimes (P_{k_2}^\dagger)^T.
$$

Then, by applying Proposition A.7, we obtain

$$
\begin{aligned}
&\left(aI \otimes I + bP_{k_1} \otimes (P_{k_1}^\dagger)^T\right)\left(cI \otimes I + dP_{k_2} \otimes (P_{k_2}^\dagger)^T\right) \\
=&ac(I \otimes I)(I \otimes I) + ad(I \otimes I)\left(P_{k_2} \otimes (P_{k_2}^\dagger)^T\right) \\
&+ bc\left(P_{k_1} \otimes (P_{k_1}^\dagger)^T\right)(I \otimes I) + bd\left(P_{k_1} \otimes (P_{k_1}^\dagger)^T\right)\left(P_{k_2} \otimes (P_{k_2}^\dagger)^T\right) \\
=&ac(I \otimes I)(I \otimes I) + ad(I \otimes I)\left(P_{k_2} \otimes (P_{k_2}^\dagger)^T\right) \\
&+ bc\left(P_{k_1} \otimes (P_{k_1}^\dagger)^T\right)(I \otimes I) + bd(-1)^{2\left(P_{k_1}, P_{k_2}\right)_{sp}}\left(P_{k_2} \otimes (P_{k_2}^\dagger)^T\right)\left(P_{k_1} \otimes (P_{k_1}^\dagger)^T\right) \\
=&\left(cI \otimes I + dP_{k_2} \otimes (P_{k_2}^\dagger)^T\right)\left(aI \otimes I + bP_{k_1} \otimes (P_{k_1}^\dagger)^T\right).
\end{aligned}
$$

$\square$

*Remark* A.11. The composite map $\Lambda^{-1} \circ \Lambda$ can be physical under specific conditions. Using the definitions of $\Lambda$ and $\Lambda^{-1}$ (defined in Equation (2) and Equation (3)), together with the commuting property (Proposition A.10), we can rewrite the composite as:

$$
\Lambda^{-1} \circ \Lambda = \bigcirc_{k \in \{1, \cdots, 4^c\}} \left(\frac{1 + e^{2(\sigma_k - \lambda_k)}}{2} I \cdot I + \frac{1 - e^{2(\sigma_k - \lambda_k)}}{2} P_k \cdot P_k\right),
$$

where $\lambda_k$ are noise parameters and $\sigma_k$ are inverse noise parameters. It follows that $\Lambda^{-1} \circ \Lambda$ is implicitly a valid quantum channel if $\sigma_k \leq \lambda_k$ for all $k \in \mathcal{K}$. Unfortunately, verifying this inequality without prior knowledge of $\lambda_k$ may require exponential effort.

## A.2 Gradient

### A.2.1 Derivation of $\frac{\partial \mathrm{tr}(\mathcal{M}\mathcal{U}_R(\rho_t))}{\partial \sigma_{j,q}}$

Using the linearity of $\mathcal{U}_R$ and operator $\frac{\partial}{\partial \sigma_{j,q}}$, we can show that

$$\frac{\partial \mathrm{tr}\left(\mathcal{M}\mathcal{U}_R(\rho_t)\right)}{\partial \sigma_{j,q}} = \mathrm{tr}\left(\mathcal{M}\frac{\partial \mathcal{U}_R(\rho_t)}{\partial \sigma_{j,q}}\right) = \mathrm{tr}\left(\mathcal{M}\frac{\partial \mathcal{U}_{t,>j}\left(\Lambda_j^{-1}(\rho_{t,j})\right)}{\partial \sigma_{j,q}}\right) \tag{14}$$

where $\rho_{t,j} = \Lambda_j \circ \mathrm{Ad}_{U_{j,b_j}} \circ \left(\bigcirc_{l=1}^{j-1}\Lambda_l^{-1}\circ\Lambda_l\circ\mathrm{Ad}_{U_{l,b_l}}\right)(\rho_t)$ and $\mathcal{U}_{t,>j}(\bullet) = \left(\bigcirc_{l=j+1}^{L}\Lambda_l^{-1}\circ\Lambda_l\circ\mathrm{Ad}_{U_{l,b_l}}\right)(\bullet)$.

**Proposition A.12.** *Let $c_1, c_2, c_3, c_4 \in \mathbb{R}$ be any constant real values. Let $P$ be any complex matrix. Suppose $A : \mathbb{R} \to \mathbb{C}^{N\times N}$ is any function depending on the variable $x \in \mathbb{R}$. Then,*

$$\frac{\partial}{\partial x}\left(\left((c_1 + c_2)I \bullet I + (c_3 + c_4)P \bullet P^\dagger\right)(A(x))\right)$$
$$= \left((c_1 + c_2)I \bullet I + (c_3 + c_4)P \bullet P^\dagger\right)\left(\frac{\partial}{\partial x}A(x)\right).$$

*Proof of Proposition A.12.* Plugging in the matrix $A(x)$ into $\left((c_1 + c_2)I \bullet I + (c_3 + c_4)P \bullet P^\dagger\right)$ on the left-hand side, we obtain:

$$\left((c_1 + c_2)I \bullet I + (c_3 + c_4)P \bullet P^\dagger\right)(A(x)) = (c_1 + c_2)A(x) + (c_3 + c_4)PA(x)P^\dagger.$$

Taking the derivative with respect to $x$, we get,

$$\frac{\partial}{\partial x}\left((c_1 + c_2)A(x) + (c_3 + c_4)PA(x)P^\dagger\right) = (c_1 + c_2)\left(\frac{\partial A(x)}{\partial x}\right) + (c_3 + c_4)P\left(\frac{\partial A(x)}{\partial x}\right)P^\dagger,$$

which is exactly the same as the expression on the right-hand side. $\square$

Using Proposition A.12, it is straightforward to verify that operator $\mathcal{U}_{t,>j}$ commutes with $\frac{\partial}{\partial \sigma_{j,q}}$. Therefore, Equation (14) simplifies to

$$\mathrm{tr}\left(\mathcal{M}\frac{\partial \mathcal{U}_{t,>j}\left(\Lambda_j^{-1}(\rho_{t,j})\right)}{\partial \sigma_{j,q}}\right) = \mathrm{tr}\left(\mathcal{M}\mathcal{U}_{t,>j}\left(\frac{\partial \Lambda_j^{-1}(\rho_{t,j})}{\partial \sigma_{j,q}}\right)\right). \tag{15}$$

Next, applying the Pauli decomposition of $\rho_{t,j}$, we obtain

$$\rho_{t,j} = \sum_{g\in\{I,X,Y,Z\}^{\otimes n}} \alpha_{j,g}g. \tag{16}$$

Combining Equation (15), Equation (16), and Proposition A.9, we arrive at

$$\frac{\partial \mathrm{tr}\left(\mathcal{M}\mathcal{U}_R(\rho_t)\right)}{\partial \sigma_{j,q}} = \sum_{g\in\{I,X,Y,Z\}^{\otimes n}} \alpha_{j,g}\mathrm{tr}\left(\mathcal{M}\mathcal{U}_{t,>j}\left(\frac{\partial \Lambda_j^{-1}(g)}{\partial \sigma_{j,q}}\right)\right). \tag{17}$$

To proceed, we need to compute the partial derivative, $\frac{\partial \Lambda_j^{-1}(g)}{\partial \sigma_{j,q}}$. First, let us simplify $\Lambda_j^{-1}(g)$ using the following proposition.

**Proposition A.13.** *If $P, g$ are any two Pauli strings, then*

$$\left(aI \bullet I + bP \bullet P^\dagger\right)(g) = \begin{cases} (a + b)g & [P, g] = 0 \\ (a - b)g & o.w, \end{cases}$$

*for any $a, b \in \mathbb{R}$.*

*Proof of Proposition A.13.* By definition, we have

$$\left( aI \bullet I + bP \bullet P^\dagger \right)(g) = a \cdot g + b \cdot PgP^\dagger$$

Considering two cases:

- **Case 1:** If $[P, g] = 0$, then

$$a \cdot g + b \cdot PgP^\dagger = a \cdot g + b \cdot gPP^\dagger = (a+b)g.$$

- **Case 2:** If $[P, g] \neq 0$, then by Proposition A.7 we know $gP = (-1)Pg$, which implies:

$$a \cdot g + b \cdot PgP = a \cdot g - b \cdot gPP^\dagger = (a-b)g.$$

$\square$

Using Proposition A.13, we deduce, for all $g \in \{I, X, Y, Z\}^{\otimes n}$,

$$\Lambda_j^{-1}(g) = \gamma_j \bigcirc_{k \in \mathcal{K}_j} \left( q_{j,k} \cdot I \bullet I + (q_{j,k} - 1) P_k^{(j)} \bullet (P_k^{(j)})^\dagger \right)(g)$$

$$= \gamma_j \cdot \left( \prod_{\substack{k \in \mathcal{K}_j \\ \left( P_k^{(j)}, g \right)_{sp} = 0}} (2q_{j,k} - 1) \cdot \prod_{\substack{k \in \mathcal{K}_j \\ \left( P_k^{(j)}, g \right)_{sp} = 1}} 1 \right) g$$

$$= \gamma_j \cdot \prod_{\substack{k \in \mathcal{K}_j \\ \left( P_k^{(j)}, g \right)_{sp} = 0}} (2q_{j,k} - 1) \, g.$$

Recall that $\gamma_j = \exp\left\{ 2 \sum_{k \in \mathcal{K}_j} \sigma_{j,k} \right\}$ and $q_{j,k} = \frac{1 + \exp\{-2\sigma_{j,k}\}}{2}$. Substituting the corresponding terms, we obtain

$$\gamma_j \cdot \prod_{\substack{k \in \mathcal{K}_j \\ \left( P_k^{(j)}, g \right)_{sp} = 0}} (2q_{j,k} - 1) \, g$$

$$= \exp\left\{ 2 \sum_{k \in \mathcal{K}_j} \sigma_{j,k} \right\} \cdot \prod_{\substack{k \in \mathcal{K}_j \\ \left( P_k^{(j)}, g \right)_{sp} = 0}} \left( 2 \cdot \frac{1 + \exp\{-2\sigma_{j,k}\}}{2} - 1 \right) g$$

$$= \exp\left\{ 2 \sum_{k \in \mathcal{K}_j} \sigma_{j,k} \right\} \cdot \prod_{\substack{k \in \mathcal{K}_j \\ \left( P_k^{(j)}, g \right)_{sp} = 0}} \exp\{-2\sigma_{j,k}\} \cdot g$$

$$= \exp\left\{ 2 \sum_{k \in \mathcal{K}_j} \sigma_{j,k} \right\} \cdot \prod_{k \in \mathcal{K}_j} \exp\left\{ -2\sigma_{j,k} \cdot \mathbb{I}\left[ \left( P_k^{(j)}, g \right)_{sp} = 0 \right] \right\} g$$

$$= \exp\left\{ 2 \sum_{k \in \mathcal{K}_j} \sigma_{j,k} - \left( 2 \sum_{k \in \mathcal{K}_j} \sigma_{j,k} \cdot \mathbb{I}\left[ \left( P_k^{(j)}, g \right)_{sp} = 0 \right] \right) \right\} g.$$

Here, $\mathbb{I}[\cdot]$ is the indicator function. By splitting $2\sum_{k\in\mathcal{K}_j}\sigma_{j,k}$ into $2\sum_{\substack{k\in\mathcal{K}_j\\\left(P_k^{(j)},g\right)_{sp}=0}}\sigma_{j,k}$ and

$2\sum_{\substack{k\in\mathcal{K}_j\\\left(P_k^{(j)},g\right)_{sp}=1}}\sigma_{j,k}$ and rearranging terms, we obtain

$$
\Lambda_j^{-1}(g) = \exp\left\{2\sum_{\substack{k\in\mathcal{K}_j\\\left(P_k^{(j)},g\right)_{sp}=0}}\sigma_{j,k} - \sigma_{j,k}\right\}\exp\left\{2\sum_{\substack{k\in\mathcal{K}_j\\\left(P_k^{(j)},g\right)_{sp}=1}}\sigma_{j,k}\right\}g
$$

$$
= \exp\left\{2\sum_{k\in\mathcal{K}_j}\sigma_{j,k}\left(P_k^{(j)},g\right)_{sp}\right\}g. \tag{18}
$$

Next, applying chain rule to compute the partial derivative of $\Lambda_j^{-1}(g)$ w.r.t. $\sigma_{j,q}$, we get:

$$
\frac{\partial\Lambda_j^{-1}(g)}{\partial\sigma_{j,q}} = \frac{\partial}{\partial\sigma_{j,q}}\exp\left\{2\sum_{k\in\mathcal{K}_j}\sigma_{j,k}\cdot\left(P_k^{(j)},g\right)_{sp}\right\}g
$$

$$
= \exp\left\{2\sum_{k\in\mathcal{K}_j}\sigma_{j,k}\cdot\left(P_k^{(j)},g\right)_{sp}\right\}\cdot 2\left(P_q^{(j)},g\right)_{sp}g. \tag{19}
$$

Combining Equation (17), Equation (18), and Equation (19), we have

$$
\frac{\partial\operatorname{tr}\left(\mathcal{M}\mathcal{U}_R(\rho_t)\right)}{\partial\sigma_{j,q}} \tag{20}
$$

$$
= \sum_{g\in\{I,X,Y,Z\}^{\otimes n}}\alpha_{j,g}\operatorname{tr}\left(\mathcal{M}\mathcal{U}_{t,>j}\left(\exp\left\{2\sum_{k\in\mathcal{K}_j}\sigma_{j,k}\cdot\left(P_k^{(j)},g\right)_{sp}\right\}\cdot 2\left(P_q^{(j)},g\right)_{sp}\cdot g\right)\right)
$$

$$
= \sum_{g\in\{I,X,Y,Z\}^{\otimes n}}\alpha_{j,g}\cdot 2\left(P_q^{(j)},g\right)_{sp}\operatorname{tr}\left(\mathcal{M}\mathcal{U}_{t,>j}\left(\exp\left\{2\sum_{k\in\mathcal{K}_j}\sigma_{j,k}\cdot\left(P_k^{(j)},g\right)_{sp}\right\}\cdot g\right)\right)
$$

$$
= \sum_{\substack{g\in\{I,X,Y,Z\}^{\otimes n}\\\left(P_q^{(j)},g\right)_{sp}=1}}2\alpha_{j,g}\operatorname{tr}\left(\mathcal{M}\mathcal{U}_{t,>j}\left(\exp\left\{2\sum_{k\in\mathcal{K}_j}\sigma_{j,k}\cdot\left(P_k^{(j)},g\right)_{sp}\right\}\cdot g\right)\right)
$$

$$
= \sum_{\substack{g\in\{I,X,Y,Z\}^{\otimes n}\\\left(P_q^{(j)},g\right)_{sp}=1}}2\alpha_{j,g}\operatorname{tr}\left(\mathcal{M}\mathcal{U}_{t,>j}\left(\Lambda_j^{-1}(g)\right)\right)
$$

$$
= 2\operatorname{tr}\left(\mathcal{M}\mathcal{U}_{t,>j}\left(\Lambda_j^{-1}\left(\sum_{\substack{g\in\{I,X,Y,Z\}^{\otimes n}\\\left(P_q^{(j)},g\right)_{sp}=1}}\alpha_{j,g}g\right)\right)\right). \tag{21}
$$

The expression presented in Equation (21) is impractical for direct use. To address this, we consider the following observation.

**Proposition A.14.** *If $P\in\{I,X,Y,Z\}^{\otimes n}$ is any Pauli string, then*

$$
\sum_{\substack{g\in\{I,X,Y,Z\}^{\otimes n}\\(P,g)_{sp}=1}}\alpha_{j,g}g = \frac{1}{2}\left(\rho_{t,j} - P\rho_{t,j}P^\dagger\right).
$$

*Proof of Proposition A.14.* Again, applying Proposition A.7 the linearity of the adjoint operator, we obtain

$$P\rho_{t,j}P^{\dagger} = \sum_{\substack{g\in\{I,X,Y,Z\}^{\otimes n}\\(P,g)_{sp}=0}} \alpha_{j,g}g - \sum_{\substack{g\in\{I,X,Y,Z\}^{\otimes n}\\(P,g)_{sp}=1}} \alpha_{j,g}g. \tag{22}$$

Combining Equation (16) and Equation (22), we have

$$\frac{1}{2}\left(\rho_{t,j} - P\rho_{t,j}P\right)$$

$$=\frac{1}{2}\left(\left(\sum_{g\in\{I,X,Y,Z\}^{\otimes n}} \alpha_{j,g}g\right) - \left(\sum_{\substack{g\in\{I,X,Y,Z\}^{\otimes n}\\(P,g)_{sp}=0}} \alpha_{j,g}g - \sum_{\substack{g\in\{I,X,Y,Z\}^{\otimes n}\\(P,g)_{sp}=1}} \alpha_{j,g}g\right)\right)$$

$$=\frac{1}{2}\left(\sum_{\substack{g\in\{I,X,Y,Z\}^{\otimes n}\\(P,g)_{sp}=0}} \alpha_{j,g}g + \sum_{\substack{g\in\{I,X,Y,Z\}^{\otimes n}\\(P,g)_{sp}=1}} \alpha_{j,g}g\right)$$

$$-\frac{1}{2}\left(\sum_{\substack{g\in\{I,X,Y,Z\}^{\otimes n}\\(P,g)_{sp}=0}} \alpha_{j,g}g - \sum_{\substack{g\in\{I,X,Y,Z\}^{\otimes n}\\(P,g)_{sp}=1}} \alpha_{j,g}g\right)$$

$$=\frac{1}{2}\cdot 2\cdot\left(\sum_{\substack{g\in\{I,X,Y,Z\}^{\otimes n}\\(P,g)_{sp}=1}} \alpha_{j,g}g\right) = \sum_{\substack{g\in\{I,X,Y,Z\}^{\otimes n}\\(P,g)_{sp}=1}} \alpha_{j,g}g$$

$$\square$$

By applying Proposition A.14, we are able to reformulate Equation (21) into a form analogous to the parameter-shift rule. Specifically, we obtain:

$$\frac{\partial \mathrm{tr}\left(\mathcal{M}\mathcal{U}_R(\rho_t)\right)}{\partial \sigma_{j,q}} = 2\cdot\frac{1}{2}\cdot\left[\mathrm{tr}\left(\mathcal{M}\mathcal{U}_{t,>j}\left(\Lambda_j^{-1}\left(\rho_{t,j}\right)\right)\right) - \mathrm{tr}\left(\mathcal{M}\mathcal{U}_{t,>j}\Lambda_j^{-1}\left(P_q^{(j)}\rho_{t,j}(P_q^{(j)})^{\dagger}\right)\right)\right].$$

### A.2.2  Derivation of $\frac{\partial\mathbf{tr}(\mathcal{M}\mathcal{U}_R(\rho_t))}{\partial\theta_j}$

We include both parameterized gate and constant gates in the PQC using Equation (1), therefore, we have to make sure that gradient calculation is exclusively performed for parameterized gates, i.e., model parameters only. As a result, we derive the expression for $\frac{\partial\mathrm{tr}(\mathcal{M}\mathcal{U}_R(\rho_t))}{\partial\theta_j}$ for all $1\le j\le L$ such that $b_j = 1$ to ensure gradient calculation for appropriate model parameters.

Using the definition and linearity property (Proposition A.12), we express the partial derivative as follows:

$$\frac{\partial\mathrm{tr}\left(\mathcal{M}\mathcal{U}_R(\rho_t)\right)}{\partial\theta_j} =\mathrm{tr}\left(\mathcal{M}\mathcal{U}_{t,>j}\left(\Lambda_j^{-1}\circ\Lambda_j\left(\frac{\partial}{\partial\theta_j}\left(Ad_{U_{j,b_j}}\left(\rho_t^{(j-1)}\right)\right)\right)\right)\right)$$

$$=\mathrm{tr}\left(\mathcal{M}\mathcal{U}_{t,>j}\left(\Lambda_j^{-1}\circ\Lambda_j\left(\frac{\partial}{\partial\theta_j}\left(Ad_{U_j(\theta_j)}\left(\rho_t^{(j-1)}\right)\right)\right)\right)\right)$$

$$=\mathrm{tr}\left(\mathcal{M}\mathcal{U}_{t,>j}\left(\Lambda_j^{-1}\circ\Lambda_j\left(\frac{\partial}{\partial\theta_j}\left(U_j(\theta_j)\rho_t^{(j-1)}U_j^{\dagger}(\theta_j)\right)\right)\right)\right).$$

where $\rho_t^{(j-1)} = \left( \bigcirc_{l=1}^{j-1} \Lambda_l^{-1} \circ \Lambda_l \circ \mathrm{Ad}_{U_{l,b_l}} \right)(\rho_t)$. The partial derivative of $U_j(\theta_j)\rho_t^{(j-1)}U_j^\dagger(\theta_j)$ with respect to $\theta_l$ is given by

$$
\frac{\partial U_j(\theta_j)\rho_t^{(j-1)}U_j^\dagger(\theta_j)}{\partial \theta_j} = \frac{\partial U_j(\theta_j)}{\partial \theta_j}\rho_t^{(j-1)}U_j^\dagger(\theta_j) + U_j(\theta_j)\rho_t^{(j-1)}\frac{\partial U_j^\dagger(\theta_j)}{\partial \theta_j}
$$

$$
= iG_j U_j(\theta_j)\rho_t^{(j-1)}U_j^\dagger(\theta_j) - iU_j(\theta_j)\rho_t^{(j-1)}U_j^\dagger(\theta_j)G_j
$$

$$
= i\left[ G_j, U_j(\theta_j)\rho_t^{(j-1)}U_j^\dagger(\theta_j) \right].
$$

Applying the parameter-shift rule given in [17], we obtain:

$$
\frac{\partial U_j(\theta_j)\rho_t^{(j-1)}U_j^\dagger(\theta_j)}{\partial \theta_j}
$$

$$
= \frac{1}{2}\left( U_j\left(\theta_j + \frac{\pi}{2}\right)\rho_t^{(j-1)}U_j^\dagger\left(\theta_j + \frac{\pi}{2}\right)\right) - \frac{1}{2}\left( U_j\left(\theta_j - \frac{\pi}{2}\right)\rho_t^{(j-1)}U_j^\dagger\left(\theta_j - \frac{\pi}{2}\right)\right).
$$

Altogether, $\frac{\partial \mathrm{tr}(\mathcal{M}\mathcal{U}_R(\rho_t))}{\partial \theta_j}$ is given by:

$$
\frac{\partial \mathrm{tr}\left(\mathcal{M}\mathcal{U}_R(\rho_t)\right)}{\partial \theta_j}
$$

$$
= \frac{1}{2}\mathrm{tr}\left( \mathcal{M}\mathcal{U}_{t,>j}\left( \Lambda_j^{-1} \circ \Lambda_j \circ \mathrm{Ad}_{U_j\left(\theta_j + \frac{\pi}{2}\right)}\left(\rho_t^{(j-1)}\right)\right)\right)
$$

$$
- \frac{1}{2}\mathrm{tr}\left( \mathcal{M}\mathcal{U}_{t,>j}\left( \Lambda_j^{-1} \circ \Lambda_j \circ \mathrm{Ad}_{U_j\left(\theta_j - \frac{\pi}{2}\right)}\left(\rho_t^{(j-1)}\right)\right)\right).
$$

### A.2.3 The Gradient of Loss Function

Given the observations above, applying the chain rule, we obtain:

$$
\frac{\partial \left( y_t - \mathrm{tr}\left(\mathcal{M}\mathcal{U}_R(\rho_t)\right)\right)^2}{\partial \sigma_{j,q}} = -4\left( y_t - \mathrm{tr}\left(\mathcal{M}\mathcal{U}_R(\rho_t)\right)\right) \cdot \mathrm{tr}\left( \mathcal{M}\mathcal{U}_{t,>j}\left( \Lambda_j^{-1} \circ \mathcal{U}_{P_q^{(j)}}(\rho_{t,j})\right)\right), \quad (23)
$$

where $\mathcal{U}_{P_q^{(j)}} = \frac{1}{2}\mathrm{Ad}_I - \frac{1}{2}\mathrm{Ad}_{P_q^{(j)}}$, and:

$$
\frac{\partial \left( y_t - \mathrm{tr}\left(\mathcal{M}\mathcal{U}_R(\rho_t)\right)\right)^2}{\partial \theta_j} = -2\left( y_t - \mathrm{tr}\left(\mathcal{M}\mathcal{U}_R(\rho_t)\right)\right) \cdot \mathrm{tr}\left( \mathcal{M}\mathcal{U}_{t,>j}\left( \Lambda_j^{-1} \circ \Lambda_j \circ \mathcal{U}_{j,\frac{\pi}{2}}\left(\rho_t^{(j-1)}\right)\right)\right)
$$

$$
\tag{24}
$$

, where $\mathcal{U}_{j,a} = \frac{1}{2}\mathrm{Ad}_{U_j(\theta_j+a)} - \frac{1}{2}\mathrm{Ad}_{U_j(\theta_j-a)}$ for $a \in \mathbb{R}$.

## A.3 Estimation of Gradient

### A.3.1 Universal Estimation Algorithm

In this section, we introduce the universal estimation algorithm, a quantum-classical hybrid method designed to estimate the quantity

$$
\mathrm{tr}\left( \mathcal{M} \bigcirc_{l=1}^{L} \Lambda_l^{-1} \circ \mathrm{Ad}_{W_l} \circ \Lambda_l \circ \mathrm{Ad}_{U_l}(\rho)\right). \tag{25}
$$

Here, $\mathcal{M}$ denotes a Hermitian observable, $\rho$ is a density operator, $W_l$ is a Pauli string, and $U_l$ represents an arbitrary (non-Pauli) quantum operator. By construction, each composition $\mathrm{Ad}_{W_l} \circ \Lambda_l \circ \mathrm{Ad}_{U_l}$ defines a valid quantum channel, and can therefore be implemented directly on quantum hardware. The primary challenge lies in the realization of the non-physical inverse channels $\Lambda_l^{-1}$.

Fortunately, a simple classical preprocessing step enables their indirect implementation. Recall that each $\Lambda_l^{-1}$ is defined as a linear combination of identity and conjugation by a Pauli operator, i.e., $q_{k,l} \bullet -(1 - q_{k,l})P_k^{(l)} \bullet P_k^{(l)}$, where $q_{k,l} \in (0,1)$. To realize $\Lambda_l^{-1}$, we sample from the Bernoulli distribution $\mathrm{Bernoulli}(1 - q_{k,l})$. If the outcome is 0, we apply the identity operation; if the outcome is 1, we apply the Pauli string $P_k^{(l)}$ and record a classical multiplier of $-1$. After measuring the quantum

state at the final stage of the algorithm, we multiply the final result by $(-1)^S$, where $S$ is the number of such $-1$ recorded during the run. The detailed implementation is presented in Algorithm 2.

---

**Algorithm 2** Universal Estimation Algorithm
(`universal_estimator`)

---

**Input:** $t \in [|\mathcal{D}|]$, Measurement $\mathcal{M}$, Parameters $\overrightarrow{\sigma} \in \mathcal{X}_{sigma}$, $\overrightarrow{\theta} \in \mathcal{X}_{theta}$.
**Input:** A set of constant Pauli strings $(W_1, \cdots, W_L)$ {By assumption 3.2, all $W_l$ are noiseless}
$S \leftarrow 0$
Prepare the quantum state $\rho_t$
**for** $l = 1$ to $L$ **do**
    {Apply the inverse channel $\Lambda_l^{-1}$}
    $\rho_t \leftarrow \text{Ad}_{W_l} \circ \Lambda_l \circ \text{Ad}_{U_l}(\rho)$ {Evolve the Quantum State}
    {Here, for $\text{Ad}_{U_l}$, we look up the corresponding parameter $\theta$ in $\overrightarrow{\theta}$ when $U_l$ is parameterized.}
    **for** $k \in \mathcal{K}_l$ **do**
        $q_{l,k} \leftarrow \frac{1 + \exp(-2\sigma_{l,k})}{2}$
        $b_{l,k} \sim \text{Bernoulli}(1 - q_{l,k})$
        **if** $b_{l,k} = 0$ **then**
            $\rho_t \leftarrow \rho$ {Apply Identity}
        **else**
            $\rho_t \leftarrow P_k^{(l)} \rho (P_k^{(l)})^\dagger$ {Apply $P_k^{(l)}$}
            $S \leftarrow S + 1$
        **end if**
    **end for**
**end for**
Measure the quantum state $\rho$ on $\mathcal{M}$ and let $\mathbb{O}$ be the outcome of the measurement.
**Return** $\exp\left(2 \sum_{l \in [1,L], k \in \mathcal{K}_l} \sigma_{l,k}^{(t)}\right) \cdot (-1)^S \cdot \mathbb{O}$

---

*Remark* A.15. By construction, the universal estimation algorithm does not impose stringent requirements on the locality of the quantum operations involved (see Remark A.4). However, when the locality is too low—for instance, when $c = n$—the structure of the inverse channel $\Lambda_l^{-1}$ can become significantly more complex. In such cases, the cardinality of the corresponding index set $\mathcal{K}_l$ may grow exponentially with the number of qubits $n$, rendering the algorithm computationally intractable.

We now proceed to show that Algorithm 2 yields an unbiased estimator of the quantity $\text{tr}\left(\mathcal{M} \bigcirc_{l=1}^{L} \Lambda_l^{-1} \circ \text{Ad}_{W_l} \circ \Lambda_l \circ \text{Ad}_{U_l}(\rho_t)\right)$.

**Lemma A.16.** `universal_estimator` *(Algorithm 2) provides an unbiased estimation of* $\text{tr}\left(\mathcal{M} \bigcirc_{l=1}^{L} \Lambda_l^{-1} \circ Ad_{W_l} \circ \Lambda_l \circ Ad_{U_l}(\rho_t)\right)$, *where $\mathcal{M}$ is a Hermitian and $\rho_t$ is a density operator.*

*Proof of Lemma A.16.* Let $(b_{l,k}) \in \{0, 1\}^{|\overrightarrow{\sigma}|}$ be a set of independent Bernoulli random variables with parameter $1 - q_{l,k} = 1 - \frac{1 + \exp(-2\sigma_{l,k})}{2}$. We denote by $\rho_{(b_{l,k})}$, the quantum state right before the measurement $\mathcal{M}$, which depends on the choice of $b_{l,k}$. From the algorithm definition, we note that $(-1)^S$ and $\rho_{(b_{l,k})}$ are random variables depending on $b_{l,k}$, and they can be re-written as follows:

$$(-1)^S = (-1)^{\sum_{l,k} b_{l,k}},$$

and

$$\rho_{(b_{l,k})} = \bigcirc_{l=1}^{L} \left(\bigcirc_{k \in \mathcal{K}_l} \text{Ad}_{(P_k^{(l)})^{b_{l,k}}}\right) \circ \text{Ad}_{W_l} \circ \Lambda_l \circ \text{Ad}_{U_l}(\rho_t). \tag{26}$$

Here, $\text{Ad}_A(\bullet) = A \bullet A^\dagger$ is the adjoint map and the bullet $\bullet$ stands as a placeholder. For clarity, we re-express Equation (26) by vectorizing it using the identity, $\text{vec}[ABC] = (A \otimes C^T)\text{vec}[B]$. We then obtain:

$$\text{vec}\left[\rho_{(b_{l,k})}\right] = \left(\prod_{l=L}^{1} \left(\prod_{k \in \mathcal{K}_l} \left(P_k^{(l)} \otimes \left((P_k^{(l)})^\dagger\right)^T\right)^{b_{l,k}}\right) \mathcal{N}_l\right) \text{vec}[\rho_t],$$

where $\mathcal{N}_l$ is defined by:

$$\left(\prod_{k \in \mathcal{K}_l} \left((\omega_{l,k})I \otimes I + (1 - \omega_{l,k})P_k^{(l)} \otimes \left((P_k^{(l)})^\dagger\right)^T\right)\right)\left(W_l \otimes \left(W_l^\dagger\right)^T\right)\left(U_{l,b_l} \otimes \left(U_{l,b_l}^\dagger\right)^T\right).$$

Taking expectation over $b_{l,k}$, we arrive at:

$$\mathbb{E}_{b_{l,k}}\left[(-1)^{\sum_{l,k} b_{l,k}} \cdot \mathrm{vec}\left[\rho_{(b_{l,k})}\right]\right]$$

$$= \mathbb{E}_{b_{l,k}}\left[(-1)^{\sum_{l,k} b_{l,k}} \cdot \left(\prod_{l=L}^{1}\left(\prod_{k \in \mathcal{K}_l}\left(P_k^{(l)} \otimes \left((P_k^{(l)})^\dagger\right)^T\right)^{b_{l,k}}\right)\mathcal{N}_l\right)\mathrm{vec}\left[\rho_t\right]\right]$$

$$= \mathbb{E}_{b_{l,k}}\left[\left(\prod_{l=L}^{1}\left(\prod_{k \in \mathcal{K}_l}\left(-P_k^{(l)} \otimes \left((P_k^{(l)})^\dagger\right)^T\right)^{b_{l,k}}\right)\mathcal{N}_l\right)\mathrm{vec}\left[\rho_t\right]\right].$$

Since $b_{l,k}$ are independent, it follows that the terms $\left(-P_k^{(l)} \otimes \left((P_k^{(l)})^\dagger\right)^T\right)^{b_{l,k}}$ are also independent. Moreover, $\Lambda_l$ and $\rho_t$ do not depend on $b_{l,k}$. Therefore, applying properties of expected value, we obtain:

$$\mathbb{E}_{b_{l,k}}\left[(-1)^{\sum_{l,k} b_{l,k}} \cdot \mathrm{vec}\left[\rho_{(b_{l,k})}\right]\right]$$

$$= \left(\prod_{l=L}^{1}\left(\prod_{k \in \mathcal{K}_l} \mathbb{E}_{b_{l,k}}\left[\left(-P_k^{(l)} \otimes \left((P_k^{(l)})^\dagger\right)^T\right)^{b_{l,k}}\right]\right)\mathcal{N}_l\right)\mathrm{vec}\left[\rho_t\right]$$

$$= \left(\prod_{l=L}^{1}\left(\prod_{k \in \mathcal{K}_l}\left(q_{l_k} \cdot \left(-P_k^{(l)} \otimes \left((P_k^{(l)})^\dagger\right)^T\right)^{0}\right.\right.\right.$$

$$\left.\left.\left. + (1 - q_{l,k}) \cdot \left(-P_k^{(l)} \otimes \left((P_k^{(l)})^\dagger\right)^T\right)^{1}\right)\right)\mathcal{N}_l\right)\mathrm{vec}\left[\rho_t\right]$$

$$= \left(\prod_{l=L}^{1}\left(\prod_{k \in \mathcal{K}_l}\left(q_{l_k} \cdot I \otimes I - (1 - q_{l,k}) \cdot P_k^{(l)} \otimes \left((P_k^{(l)})^\dagger\right)^T\right)\right)\mathcal{N}_l\right)\mathrm{vec}\left[\rho_t\right]. \qquad (27)$$

Re-writing Equation (27), we express the expectation as:

$$\mathbb{E}_{b_{l,k}}\left[(-1)^{\sum_{l,k} b_{l,k}} \cdot \rho_{(b_{l,k})}\right]$$

$$= \left(\bigcirc_{l=1}^{L}\left(\bigcirc_{k \in \mathcal{K}_l}\left(q_{l_k}\mathrm{Ad}_I - (1 - q_{l,k})\mathrm{Ad}_{P_k^{(l)}}\right)\right) \circ \mathrm{Ad}_{W_l} \circ \Lambda_l \circ \mathrm{Ad}_{U_l}\right)(\rho_t)$$

$$= \left(\bigcirc_{l=1}^{L}\left(\frac{1}{\gamma_l} \cdot \gamma_l \bigcirc_{k \in \mathcal{K}_l}\left(q_{l_k}\mathrm{Ad}_I - (1 - q_{l,k})\mathrm{Ad}_{P_k^{(l)}}\right)\right) \circ \mathrm{Ad}_{W_l} \circ \Lambda_l \circ \mathrm{Ad}_{U_l}\right)(\rho_t)$$

$$= \left(\bigcirc_{l=1}^{L}\left(\frac{1}{\gamma_l}\Lambda_l^{-1}\right) \circ \mathrm{Ad}_{W_l} \circ \Lambda_l \circ \mathrm{Ad}_{U_l}\right)(\rho_t),$$

where $\gamma_l = \exp\left\{2\sum_{k \in \mathcal{K}_l} \sigma_{l,k}\right\}$. For a fixed $b_{l,k}$, one can easily verify that $\rho_{(b_{l,k})}$ is a density operator. Using the property of quantum measurement, we note that the expected measurement outcome $\mathbb{0}$ is given by $\mathrm{tr}\left(\mathcal{M}\rho_{(b_{l,k})}\right)$. Combining these results, the expected value of the output of

the universal estimation algorithm is given by:

$$\mathbb{E}\left[\exp\left(2\sum_{l\in[1,L],k\in\mathcal{K}_l}\sigma_{l,k}^{(t)}\right)\cdot(-1)^S\cdot\mathbb{O}\right]$$

$$=\exp\left(2\sum_{l\in[1,L],k\in\mathcal{K}_l}\sigma_{l,k}^{(t)}\right)\cdot\mathbb{E}\left[(-1)^{\sum_{l,k}b_{l,k}}\cdot\mathbb{O}\right]$$

$$=\exp\left(2\sum_{l\in[1,L],k\in\mathcal{K}_l}\sigma_{l,k}^{(t)}\right)\cdot\mathbb{E}_{b_{l,k}}\left[\mathbb{E}\left[(-1)^{\sum_{l,k}b_{l,k}}\cdot\mathbb{O}\Big|b_{l,k}\right]\right]$$

$$=\exp\left(2\sum_{l\in[1,L],k\in\mathcal{K}_l}\sigma_{l,k}^{(t)}\right)\cdot\mathbb{E}_{b_{l,k}}\left[(-1)^{\sum_{l,k}b_{l,k}}\cdot\mathrm{tr}\left(\mathcal{M}\rho_{(b_{l,k})}\right)\right]$$

$$=\exp\left(2\sum_{l\in[1,L],k\in\mathcal{K}_l}\sigma_{l,k}^{(t)}\right)\cdot\mathrm{tr}\left(\mathcal{M}\cdot\mathbb{E}_{b_{l,k}}\left[(-1)^{\sum_{l,k}b_{l,k}}\cdot\rho_{(b_{l,k})}\right]\right)$$

$$=\left(\prod_{l=L}^1\gamma_l\right)\cdot\mathrm{tr}\left(\mathcal{M}\left(\bigcirc_{l=1}^L\left(\frac{1}{\gamma_l}\Lambda_l^{-1}\right)\circ\mathrm{Ad}_{W_l}\circ\Lambda_l\circ\mathrm{Ad}_{U_l}\right)(\rho_t)\right)$$

$$=\mathrm{tr}\left(\mathcal{M}\left(\bigcirc_{l=1}^L\left(\gamma_l\cdot\frac{1}{\gamma_l}\Lambda_l^{-1}\right)\circ\mathrm{Ad}_{W_l}\circ\Lambda_l\circ\mathrm{Ad}_{U_l}\right)(\rho_t)\right)$$

$$=\mathrm{tr}\left(\mathcal{M}\bigcirc_{l=1}^L\Lambda_l^{-1}\circ\mathrm{Ad}_{W_l}\circ\Lambda_l\circ\mathrm{Ad}_{U_l}(\rho_t)\right).$$

$\square$

This algorithm serves as a foundational component in the estimation of partial derivatives, as discussed in the following section. There, we will show how the partial derivative estimation problem can be reduced to the trace estimation problem stated in Equation (25).

### A.3.2 Estimation of Partial Derivative with the Universal Estimation Algorithm

To estimate the partial derivatives with respect to specific parameters, we first require efficient estimations of the trace terms: $\mathrm{tr}\left(\mathcal{M}\mathcal{U}_R(\rho_t)\right)$, $\mathrm{tr}\left(\mathcal{M}\mathcal{U}_{t,>j}\left(\Lambda_j^{-1}\circ\mathrm{Ad}_{P_q^{(j)}}(\rho_{t,j})\right)\right)$, and $\mathrm{tr}\left(\mathcal{M}\mathcal{U}_{t,>j}\left(\mathrm{Ad}_{U(\theta_j+c)}(\rho_{t,j})\right)\right)$. To facilitate their estimation, we first reformulate them. In particular, the term $\mathrm{tr}\left(\mathcal{M}\mathcal{U}_R(\rho_t)\right)$ can be expressed as

$$\mathrm{tr}\left(\mathcal{M}\bigcirc_{l=1}^L\Lambda_l^{-1}\circ\mathrm{Ad}_I\circ\Lambda_l\circ\mathrm{Ad}_{U_{l,b_l}}(\rho_t)\right)\tag{28}$$

the term $\mathrm{tr}\left(\mathcal{M}\mathcal{U}_{t,>j}\left(\Lambda_j^{-1}\circ\mathrm{Ad}_{P_q^{(j)}}(\rho_{t,j})\right)\right)$ as

$$\mathrm{tr}\left(\mathcal{M}\left(\bigcirc_{l=j+1}^L\Lambda_l^{-1}\circ\mathrm{Ad}_I\circ\Lambda_l\circ\mathrm{Ad}_{U_{l,b_l}}\right)\circ\left(\Lambda_j^{-1}\circ\mathrm{Ad}_{P_q^{(j)}}\circ\Lambda_j\circ\mathrm{Ad}_{U_{j,b_j}}\right)\right.$$
$$\left.\circ\left(\bigcirc_{l=1}^{j-1}\Lambda_l^{-1}\circ\mathrm{Ad}_I\circ\Lambda_l\circ\mathrm{Ad}_{U_{l,b_l}}\right)(\rho_t)\right),\tag{29}$$

and the term $\mathrm{tr}\left(\mathcal{M}\mathcal{U}_{t,>j}\left(\mathrm{Ad}_{U(\theta_j+c)}(\rho_{t,j})\right)\right)$ as

$$\mathrm{tr}\left(\mathcal{M}\left(\bigcirc_{l=j+1}^L\Lambda_l^{-1}\circ\mathrm{Ad}_I\circ\Lambda_l\circ\mathrm{Ad}_{U_{l,b_l}}\right)\circ\left(\Lambda_j^{-1}\circ\mathrm{Ad}_I\circ\Lambda_j\circ\mathrm{Ad}_{U_j(\theta_j+c)}\right)\right.$$
$$\left.\circ\left(\bigcirc_{l=1}^{j-1}\Lambda_l^{-1}\circ\mathrm{Ad}_I\circ\Lambda_l\circ\mathrm{Ad}_{U_{l,b_l}}\right)(\rho_t)\right).\tag{30}$$

Building on the transformations described above, we can efficiently estimate the three trace terms using the universal estimation algorithm presented in Algorithm 2. To estimate the partial derivatives defined in Equation (23) and Equation (24), integrating the observations discussed above, we introduce

two estimators: `sigma_grad_est` and `theta_grad_est`, explicitly detailed in Algorithm 3 and Algorithm 4, respectively.

---

**Algorithm 3** Estimation of $\frac{\partial(y_t - \text{tr}(\mathcal{M}\mathcal{U}_R(\rho_t)))^2}{\partial\sigma_{j,q}}$ (`sigma_grad_est`)

---

**Input:** $t \in [|\mathcal{D}|]$, Measurement $\mathcal{M}$, Parameters $\overrightarrow{\sigma}^{(t)} \in \mathcal{X}_{sigma}$, $\overrightarrow{\theta}^{(t)} \in \mathcal{X}_{theta}$, true label $y_t$, $j \in [L], q \in \mathcal{K}_j$, and positive integers $\widetilde{N}_1^{(j,q)}, \widetilde{N}_2^{(j,q)}$.

$\mathcal{O}_t^{(1)}, \mathcal{O}_t^{(2,\sigma_{j,q})} \leftarrow 0, 0$

**for** $\alpha = 1$ to $\widetilde{N}_1^{(j,q)}$ **do**

   $o_\alpha^{(1)} \leftarrow$ `universal_estimator` $\left(t, \mathcal{M}, \overrightarrow{\sigma}, \overrightarrow{\theta}, (I, \cdots, I)\right)$ {Algorithm 2}

   $\mathcal{O}_t^{(1)} \leftarrow \mathcal{O}_t^{(1)} + o_\alpha^{(1)}$

**end for**

**for** $\alpha = 1$ to $\widetilde{N}_2^{(j,q)}$ **do**

   $b \sim \text{Bernoulli}(\frac{1}{2})$

   **if** $b = 1$ **then**

      $o_\alpha^{(2,\sigma_{j,q})} \leftarrow$ `universal_estimator` $\left(t, \mathcal{M}, \overrightarrow{\sigma}, \overrightarrow{\theta}, (I, \cdots, I)\right)$

   **else**

      $o_\alpha^{(2,\sigma_{j,q})} \leftarrow (-1) \cdot$ `universal_estimator` $\left(\rho_t, \mathcal{M}, \overrightarrow{\sigma}, \overrightarrow{\theta}, \left(I, \cdots, I, P_q^{(j)}, I, \cdots, I\right)\right)$

      {Here, $\left(I, \cdots, I, P_q^{(j)}, I, \cdots, I\right)$ is a $L$-tuple of identity except the one at $(j)$}

   **end if**

   $\mathcal{O}_t^{(2,\sigma_{j,q})} \leftarrow \mathcal{O}_t^{(2,\sigma_{j,q})} + o_\alpha^{(2,\sigma_{j,q})}$

**end for**

**Return** $-4\left(y_t - \frac{1}{\widetilde{N}_1^{(j,q)}}\mathcal{O}_t^{(1)}\right) \cdot \frac{1}{\widetilde{N}_2^{(j,q)}}\mathcal{O}_t^{(2,\sigma_{j,q})}$

---

**Algorithm 4** Estimation of $\frac{\partial(y_t - \text{tr}(\mathcal{M}\mathcal{U}_R(\rho_t)))^2}{\partial\theta_j}$ (`theta_grad_est`)

---

**Input:** $t \in [|\mathcal{D}|]$, Measurement $\mathcal{M}$, Parameters $\overrightarrow{\sigma}^{(t)} \in \mathcal{X}_{sigma}$, $\overrightarrow{\theta}^{(t)} \in \mathcal{X}_{theta}$, true label $y_t$, $j \in \left[\left|\left|\overrightarrow{\theta}\right|\right|\right]$, and a positive integers $\widetilde{N}_1^{(j)}, \widetilde{N}_2^{(j)}$.

$\mathcal{O}_t^{(1)}, \mathcal{O}_t^{(2,\theta_j)} \leftarrow 0, 0$

**for** $\alpha = 1$ to $\widetilde{N}_1^{(j)}$ **do**

   $o_\alpha^{(1)} \leftarrow$ `universal_estimator` $\left(t, \mathcal{M}, \overrightarrow{\sigma}, \overrightarrow{\theta}, (I, \cdots, I)\right)$ {Algorithm 2}

   $\mathcal{O}_t^{(1)} \leftarrow \mathcal{O}_t^{(1)} + o_\alpha^{(1)}$

**end for**

**for** $\alpha = 1$ to $\widetilde{N}_2^{(j)}$ **do**

   $b \sim \text{Bernoulli}(\frac{1}{2})$

   **if** $b = 1$ **then**

      $o_\alpha^{(2,\theta_j)} \leftarrow$ `universal_estimator` $\left(t, \mathcal{M}, \overrightarrow{\sigma}, \overrightarrow{\theta} + \frac{\pi}{2}\overrightarrow{e}_j, (I, \cdots, I)\right)$

      {Here, $\overrightarrow{e}_j \in \mathbb{R}^L$ is the vector that has 1 in the $j^{th}$ component and 0 elsewhere.}

   **else**

      $o_\alpha^{(2,\theta_j)} \leftarrow (-1) \cdot$ `universal_estimator` $\left(\rho_t, \mathcal{M}, \overrightarrow{\sigma}, \overrightarrow{\theta} - \frac{\pi}{2}\overrightarrow{e}_j, (I, \cdots, I)\right)$

   **end if**

   $\mathcal{O}_t^{(2,\theta_j)} \leftarrow O^{(2,\theta_j)} + o_\alpha^{(2,\theta_j)}$

**end for**

**Return** $-2\left(y_t - \frac{1}{\widetilde{N}_1^{(j)}}\mathcal{O}_t^{(1)}\right) \cdot \frac{1}{\widetilde{N}_2^{(j)}}\mathcal{O}_t^{(2,\theta_j)}$

---

We proceed to demonstrate that both algorithms produce unbiased estimators, as stated in the following proposition.

**Proposition A.17.** `sigma_grad_est` *(Algorithm 3) and* `theta_grad_est` *(Algorithm 4) provide unbiased estimators of* $\frac{\partial(y_t - tr(\mathcal{M}\mathcal{U}_R(\rho_t)))^2}{\partial \sigma_{j,q}}$ *and* $\frac{\partial(y_t - tr(\mathcal{M}\mathcal{U}_R(\rho_t)))^2}{\partial \theta_j}$, *respectively.*

*Proof of Proposition A.17.* We will show that Algorithm 3 is an unbiased estimator of $\frac{\partial(y_t - tr(\mathcal{M}\mathcal{U}_R(\rho_t)))^2}{\partial \sigma_{j,q}}$. Then, by applying a similar reasoning, one can demonstrate that the estimation produced by Algorithm 4 is also unbiased.

To start, let us show $\mathbb{E}\left[-4\left(y_t - \frac{1}{\widetilde{N}_1^{(j,q)}}\mathcal{O}_t^{(1)}\right)\right] = -4\left(y_t - tr\left(\mathcal{M}\mathcal{U}_R(\rho_t)\right)\right)$. Let $o_\alpha^{(1)}$ and $o_\alpha^{(2,\sigma_{j,q})}$ be the random variables defined in Algorithm 3. Applying linearity of expectation, we obtain:

$$\mathbb{E}\left[-4\left(y_t - \frac{1}{\widetilde{N}_1^{(j,q)}}\mathcal{O}_t^{(1)}\right)\right] = -4\left(y_t - \frac{1}{\widetilde{N}_1^{(j,q)}}\sum_{\alpha=1}^{\widetilde{N}_1^{(j,q)}}\mathbb{E}\left[o_\alpha^{(1)}\right]\right).$$

Applying Lemma A.16, we know $\mathbb{E}\left[o_\alpha^{(1)}\right] = tr\left(\mathcal{M}\mathcal{U}_R(\rho_t)\right)$. Hence,

$$\mathbb{E}\left[-4\left(y_t - \frac{1}{\widetilde{N}_1^{(j,q)}}\mathcal{O}_t^{(1)}\right)\right] = -4\left(y_t - tr\left(\mathcal{M}\mathcal{U}_R(\rho_t)\right)\right).$$

Continuing, we show that,

$$\mathbb{E}\left[\frac{1}{\widetilde{N}_2^{(j,q)}} \cdot \mathcal{O}_t^{(2,\sigma_{j,q})}\right]$$
$$= \frac{1}{2} \cdot \left(tr\left(\mathcal{M}\mathcal{U}_{t,>j}\left(\Lambda_l^{-1}\left(\rho_t^{(j-1)}\right)\right)\right) - tr\left(\mathcal{M}\mathcal{U}_{t,>j}\left(\Lambda_l^{-1}\left(P_q^{(j)}\rho_{t,j}(P_q^{(j)})^\dagger\right)\right)\right)\right).$$

Applying again linearity of expectation and incorporating the definition of $o_\alpha^{(2,\sigma_{j,q})}$, we arrive at:

$$\mathbb{E}\left[\frac{1}{\widetilde{N}_2^{(j,q)}} \cdot \mathcal{O}_t^{(2,\sigma_{j,q})}\right]$$
$$= \frac{1}{\widetilde{N}_2^{(j,q)}}\sum_{\alpha=1}^{\widetilde{N}_2^{(j,q)}}\mathbb{E}\left[o_\alpha^{(2,\sigma_{j,q})}\right]$$
$$= \frac{1}{\widetilde{N}_2^{(j,q)}}\sum_{\alpha=1}^{\widetilde{N}_2^{(j,q)}}\left(\frac{1}{2}\mathbb{E}\left[\texttt{universal\_estimator}\left(\rho_t, \mathcal{M}, \overrightarrow{\sigma}, \overrightarrow{\theta}, (I, \cdots, I)\right)\right]\right.$$
$$\left. - \frac{1}{2}\mathbb{E}\left[\texttt{universal\_estimator}\left(\rho_t, \mathcal{M}, \overrightarrow{\sigma}, \overrightarrow{\theta}, \left(I, \cdots, I, P_q^{(j)}, I, \cdots, I\right)\right)\right]\right).$$

By Lemma A.16, we know that,

$$\mathbb{E}\left[\frac{1}{\widetilde{N}_2^{(j,q)}} \cdot \mathcal{O}_t^{(2,\sigma_{j,q})}\right]$$

$$=\frac{1}{\widetilde{N}_2^{(j,q)}} \sum_{\alpha=1}^{\widetilde{N}_2^{(j,q)}} \left(\frac{1}{2}\mathrm{tr}\left(\mathcal{M}\mathcal{U}_{t,>j}\left(\Lambda_l^{-1}\left(\rho_t^{(j-1)}\right)\right)\right) - \frac{1}{2}\mathrm{tr}\left(\mathcal{M}\mathcal{U}_{t,>j}\left(\Lambda_l^{-1}\left(P_q^{(j)}\rho_{t,j}(P_q^{(j)})^\dagger\right)\right)\right)\right)$$

$$=\frac{1}{2}\left(\frac{1}{\widetilde{N}_2^{(j,q)}} \sum_{\alpha=1}^{\widetilde{N}_2^{(j,q)}} \mathrm{tr}\left(\mathcal{M}\mathcal{U}_{t,>j}\left(\Lambda_l^{-1}\left(\rho_t^{(j-1)}\right)\right)\right)\right)$$

$$-\frac{1}{2}\left(\frac{1}{\widetilde{N}_2^{(j,q)}} \sum_{\alpha=1}^{\widetilde{N}_2^{(j,q)}} \mathrm{tr}\left(\mathcal{M}\mathcal{U}_{t,>j}\left(\Lambda_l^{-1}\left(P_q^{(j)}\rho_{t,j}(P_q^{(j)})^\dagger\right)\right)\right)\right)$$

$$=\frac{1}{2}\mathrm{tr}\left(\mathcal{M}\mathcal{U}_{t,>j}\left(\Lambda_l^{-1}\left(\rho_t^{(j-1)}\right)\right)\right) - \frac{1}{2}\mathrm{tr}\left(\mathcal{M}\mathcal{U}_{t,>j}\left(\Lambda_l^{-1}\left(P_q^{(j)}\rho_{t,j}(P_q^{(j)})^\dagger\right)\right)\right).$$

Since $-4(y_t - \mathcal{O}_t^{(1)})$ and $\mathcal{O}_t^{(2,\sigma_{j,q})}$ are independent, combining results above, we obtain:

$$\mathbb{E}\left[-4\left(y_t - \frac{1}{\widetilde{N}_1^{(j,q)}} \cdot \mathcal{O}_t^{(1)}\right) \cdot \left(\frac{1}{\widetilde{N}_2^{(j,q)}} \cdot \mathcal{O}_t^{(2,\sigma_{j,q})}\right)\right]$$

$$=\mathbb{E}\left[-4\left(y_t - \frac{1}{\widetilde{N}_1^{(j,q)}} \cdot \mathcal{O}_t^{(1)}\right)\right] \cdot \mathbb{E}\left[\frac{1}{\widetilde{N}_2^{(j,q)}} \cdot \mathcal{O}_t^{(2,\sigma_{j,q})}\right]$$

$$=-4\left(y_t - \mathrm{tr}\left(\mathcal{M}\mathcal{U}_R(\rho_t)\right)\right) \cdot \mathrm{tr}\left(\mathcal{M}\mathcal{U}_{t,>j}\left(\Lambda_j^{-1} \circ \left(\frac{1}{2}\mathrm{Ad}_I - \frac{1}{2}\mathrm{Ad}_{P_q^{(j)}}\right)(\rho_{t,j})\right)\right)$$

$\square$

### A.3.3 Gradient Estimation via Probabilistic Subsampling: Algorithm and Approximation Lemma

By the definition of Algorithm 1, each step involves estimating the gradient of $\mathcal{L}_t$ at $(\overrightarrow{\sigma}, \overrightarrow{\theta})$ for a given $t \in [|\mathcal{D}|]$. To this end, we introduce the following estimation algorithm, which leverages stochastic sampling to estimate gradient w.r.t. both the inverse noise parameters $\overrightarrow{\sigma}$ and PQC parameters $\overrightarrow{\theta}$.

---

**Algorithm 5** Gradient Estimator (`gradient_estimator`)

---

**Input:** Positive Integer $t$, Measurement $\mathcal{M}$, Parameters $\overrightarrow{\sigma}^{(t)} \in \mathcal{X}_{sigma}$, $\overrightarrow{\theta}^{(t)} \in \mathcal{X}_{theta}$, true label $y_t$, an even positive integer $\widetilde{N}$, and a binary vector $(\cdots, p^{(l,k)}, \cdots, p^{(l)}, \cdots)$ of length $\left|\left(\overrightarrow{\sigma}, \overrightarrow{\theta}\right)\right|$.

**Assumption: Assumption 3.2 - Assumption 3.4**

**for** $l \in [L]$ **do**
    **for** $k \in \mathcal{K}_l$ **do**
        **if** $p^{(l,k)} = 1$ **then**
            $\left(\widetilde{\nabla}\mathcal{L}_t\left(\overrightarrow{\sigma}, \overrightarrow{\theta}\right)\right)^{\sigma}_{l,k} \leftarrow$ `sigma_grad_est`$(t, \mathcal{M}, \overrightarrow{\sigma}, \overrightarrow{\theta}, y_t, l, k, \widetilde{N}/2, \widetilde{N}/2)$
        **else**
            $\left(\widetilde{\nabla}\mathcal{L}_t\left(\overrightarrow{\sigma}, \overrightarrow{\theta}\right)\right)^{\sigma}_{l,k} \leftarrow 0$
        **end if**
    **end for**
**end for**
**for** $l \in \left[\left|\left|\overrightarrow{\theta}\right|\right|\right]$ **do**
    **if** $p^{(l)} = 1$ **then**
        $\left(\widetilde{\nabla}\mathcal{L}_t\left(\overrightarrow{\sigma}, \overrightarrow{\theta}\right)\right)^{\theta}_{l} \leftarrow$ `theta_grad_est`$(t, \mathcal{M}, \overrightarrow{\sigma}, \overrightarrow{\theta}, y_t, l, \widetilde{N}/2, \widetilde{N}/2)$
    **else**
        $\left(\widetilde{\nabla}\mathcal{L}_t\left(\overrightarrow{\sigma}, \overrightarrow{\theta}\right)\right)^{\theta}_{l} \leftarrow 0$
    **end if**
**end for**
**Return** $\left(\cdots, \left(\widetilde{\nabla}\mathcal{L}_t\left(\overrightarrow{\sigma}, \overrightarrow{\theta}\right)\right)^{\sigma}_{l,k}, \cdots, \left(\widetilde{\nabla}\mathcal{L}_t\left(\overrightarrow{\sigma}, \overrightarrow{\theta}\right)\right)^{\theta}_{l}, \cdots\right)$

---

In the following lemma, we demonstrate that Algorithm 5 produces an expected $(1/p)$-approximation of the true gradient of $\mathcal{L}_t$ at $(\overrightarrow{\sigma}, \overrightarrow{\theta})$.

**Lemma A.18.** *Let $p \in \mathbb{Z}_{\geq 1}$. `gradient_estimator` (Algorithm 5) is a $(1/p)$-approximation of the true gradient of $\mathcal{L}_t(\overrightarrow{\sigma}, \overrightarrow{\theta})$ in expectation.*

*Proof of Lemma A.18.* For simplicity, define $w = (\overrightarrow{\sigma}, \overrightarrow{\sigma})$ and $|w|$ to be the length of $w$. Let $\overrightarrow{p} \sim (\text{Bernoulli}(1/p))^{|w|}$ be a random binary vector in $\{0, 1\}^{|w|}$. By the definition of Algorithm 5, we note that its output can be represented by a random vector $\overrightarrow{p} \odot \widetilde{\nabla}\mathcal{L}_t(w)$, where $\odot$ is the element-wise vector multiplication and $\widetilde{\nabla}\mathcal{L}_t(w)$ is the estimation of the true full gradient $\nabla\mathcal{L}_t(w)$ generated by Algorithm 3 and Algorithm 4.

Taking the expected value conditioned on $\overrightarrow{p}$, we get:

$$\mathbb{E}\left[\overrightarrow{p} \odot \widetilde{\nabla}\mathcal{L}_t(w) \Big| \overrightarrow{p}\right] = \left(\cdots, (\overrightarrow{p})_\alpha \cdot \mathbb{E}\left[\left(\widetilde{\nabla}\mathcal{L}_t(w)\right)_\alpha\right], \cdots\right).$$

Here, $\left(\widetilde{\nabla}\mathcal{L}_t(w)\right)_\alpha$ is the $\alpha^{th}$ component of the vector $\widetilde{\nabla}\mathcal{L}_t(w)$. From Proposition A.17, we know that $\widetilde{\nabla}\mathcal{L}_t(w)$ is unbiased, meaning that,

$$\mathbb{E}\left[\overrightarrow{p} \odot \widetilde{\nabla}\mathcal{L}_t(w) \Big| \overrightarrow{p}\right] = \left(\cdots, (\overrightarrow{p})_\alpha \cdot (\nabla\mathcal{L}_t(w))_\alpha, \cdots\right) = \overrightarrow{p} \odot \nabla\mathcal{L}_t(w).$$

Taking the full expectation,

$$
\begin{aligned}
\mathbb{E}\left[\overrightarrow{p}\odot\widetilde{\nabla}\mathcal{L}_t\left(w\right)\right] &= \sum_{\overrightarrow{p}\in\{0,1\}^{|w|}} \mathbb{E}\left[\overrightarrow{p}\odot\widetilde{\nabla}\mathcal{L}_t\left(w\right)\Big|\overrightarrow{p}\right]\cdot\Pr\left[\overrightarrow{p}\right] \\
&= \sum_{\overrightarrow{p}\in\{0,1\}^{|w|}} \left(\overrightarrow{p}\odot\nabla\mathcal{L}_t\left(w\right)\right)\cdot\Pr\left[\overrightarrow{p}\right] \\
&= \left(\sum_{\overrightarrow{p}\in\{0,1\}^{|w|}} \Pr\left[\overrightarrow{p}\right]\cdot\overrightarrow{p}\right)\odot\nabla\mathcal{L}_t\left(w\right) \\
&= (\cdots,\mathbb{E}\left[(\overrightarrow{p})_\alpha\right],\cdots,)^T\odot\nabla\mathcal{L}_t\left(w\right) \\
&= (\cdots,\Pr\left[(\overrightarrow{p})_\alpha=0\right]\cdot 0+\Pr\left[(\overrightarrow{p})_\alpha=1\right]\cdot 1,\cdots)^T\odot\nabla\mathcal{L}_t\left(w\right) \\
&= (1/p,\cdots,1/p)^T\odot\nabla\mathcal{L}_t\left(w\right) \\
&= \frac{1}{p}\cdot\nabla\mathcal{L}_t\left(w\right).
\end{aligned}
$$

Here, $(\bullet)_\alpha$ is the $\alpha^{th}$ component of the input vector. $\qquad\square$

*Remark* A.19. The scalar $p\in\mathbb{Z}_{\geq 1}$ must not exceed the total number of parameters (inverse noise parameters and PQC parameters) being optimized in order to guarantee that the expected density of the estimation given by $\mathbb{E}\left[\left\|\overrightarrow{p}_t\odot\widetilde{\nabla}\mathcal{L}_t\left(\overrightarrow{\sigma},\overrightarrow{\theta}\right)\right\|_0\right]$ is not 0. In particular, if $p=1$, $\overrightarrow{p}_t$ is, by definition, an all-ones vector. Consequently, `gradient_estimator` (Algorithm 5) provides an unbiased estimate of the full gradient of $\mathcal{L}_t$ at $(\overrightarrow{\sigma},\overrightarrow{\theta})$.

### A.4  Sample Complexity

A review of Algorithm 5 indicates that it employs a stochastic selection strategy to reduce the overall sample complexity. In the remainder of this section, we show that the total sample complexity decreases by a factor of $1/p$ in expectation. We begin by demonstrating that a mapping of the form $\mathcal{C}\left(y-\frac{1}{n}\sum_{i=1}^{n}\mathbb{X}_i\right)\left(\frac{1}{n}\sum_{i=n+1}^{2n}\mathbb{X}_i\right)$ satisfies the bounded difference property.

**Proposition A.20.** *Let $n\in\mathbb{Z}_{>0}$ be some positive number. Let $\mathbb{X}_1,\cdots,\mathbb{X}_{2n}\in\{-1,1\}$ be $2n$ independent random variables. Define $F:\{-1,1\}^{2n}\to\mathbb{R}$ by $F(\mathbb{X}_1,\cdots,\mathbb{X}_{2n})=\mathcal{C}\left(y-\frac{1}{n}\sum_{i=1}^{n}\mathbb{X}_i\right)\left(\frac{1}{n}\sum_{i=n+1}^{2n}\mathbb{X}_i\right)$ for some constant $\mathcal{C}\in\mathbb{R}$ and $y\in\{-1,1\}$. Then, for all $1\leq j\leq 2n$,*

$$
\max_{\mathbb{X}_j,\mathbb{X}_j'\in\{-1,1\}}\left|F(\mathbb{X}_1,\cdots,\mathbb{X}_j,\cdots,\mathbb{X}_{2n})-F(\mathbb{X}_1,\cdots,\mathbb{X}_j',\cdots,\mathbb{X}_{2n})\right|\leq\frac{|4\mathcal{C}|}{n}.
$$

*Proof of Proposition A.20.* By the definition of $F$, we consider the following two cases:

- **Case 1:** Suppose $1 \leq j \leq n$. Then, for any $\mathbb{X}_j, \mathbb{X}'_j \in \{-1, 1\}$, we obtain:

$$\left| F(\mathbb{X}_1, \cdots, \mathbb{X}_j, \cdots, \mathbb{X}_n, \cdots, \mathbb{X}_{2n}) - F(\mathbb{X}_1, \cdots, \mathbb{X}'_j, \cdots, \mathbb{X}_n, \cdots, \mathbb{X}_{2n}) \right|$$

$$= \left| \mathcal{C} \left( y - \frac{1}{n} \sum_{\substack{i=1 \\ i \neq j}}^{n} \mathbb{X}_i - \frac{\mathbb{X}_j}{n} \right) \left( \frac{1}{n} \sum_{i=n+1}^{2n} \mathbb{X}_i \right) \right.$$

$$\left. - \mathcal{C} \left( y - \frac{1}{n} \sum_{\substack{i=1 \\ i \neq j}}^{n} \mathbb{X}_i - \frac{\mathbb{X}'_j}{n} \right) \left( \frac{1}{n} \sum_{i=n+1}^{2n} \mathbb{X}_i \right) \right|$$

$$= \frac{|\mathcal{C}|}{n} \cdot |\mathbb{X}'_j - \mathbb{X}_j| \cdot \left| \left( \frac{1}{n} \sum_{i=n+1}^{2n} \mathbb{X}_i \right) \right|$$

$$\leq \frac{|\mathcal{C}|}{n} \cdot |\mathbb{X}'_j - \mathbb{X}_j| \cdot \frac{1}{n} \sum_{i=n+1}^{2n} |\mathbb{X}_i| \qquad \text{(triangle inequality)}$$

$$\leq \frac{|\mathcal{C}| \cdot 2}{n^2} \cdot \sum_{i=n+1}^{2n} 1 = \frac{2|\mathcal{C}|}{n}.$$

- **Case 2:** Suppose $n + 1 \leq j \leq 2n$. Again, for any $\mathbb{X}_j, \mathbb{X}'_j \in \{-1, 1\}$, we obtain:

$$\left| F(\mathbb{X}_1, \cdots, \mathbb{X}_n, \cdots, \mathbb{X}_j, \cdots \mathbb{X}_{2n}) - F(\mathbb{X}_1, \cdots, \mathbb{X}_n, \cdots, \mathbb{X}'_j, \cdots, \mathbb{X}_{2n}) \right|$$

$$= \left| \mathcal{C} \left( y - \frac{1}{n} \sum_{i=1}^{n} \mathbb{X}_i \right) \left( \frac{1}{n} \sum_{\substack{i=n+1 \\ i \neq j}}^{2n} \mathbb{X}_i - \frac{\mathbb{X}_j}{n} \right) \right.$$

$$\left. - \mathcal{C} \left( y - \frac{1}{n} \sum_{i=1}^{n} \mathbb{X}_i \right) \left( \frac{1}{n} \sum_{\substack{i=n+1 \\ i \neq j}}^{2n} \mathbb{X}_i - \frac{\mathbb{X}'_j}{n} \right) \right|$$

$$= |\mathcal{C}| \cdot \left| y - \frac{1}{n} \sum_{i=1}^{n} \mathbb{X}_i \right| \cdot \left| \frac{\mathbb{X}_j}{n} - \frac{\mathbb{X}'_j}{n} \right|$$

$$\leq |\mathcal{C}| \cdot \left( |y| + \frac{1}{n} \sum_{i=1}^{n} |\mathbb{X}_i| \right) \cdot \frac{2}{n} \qquad \text{(triangle inequality)}$$

$$\leq |\mathcal{C}| \cdot (1 + 1) \cdot \frac{2}{n} = \frac{4|\mathcal{C}|}{n}.$$

Together, we get

$$\max_{\mathbb{X}_j, \mathbb{X}'_j \in \{-1,1\}} \left| F(\mathbb{X}_1, \cdots, \mathbb{X}_j, \cdots \mathbb{X}_{2n}) - F(\mathbb{X}_1, \cdots, \mathbb{X}'_j, \cdots, \mathbb{X}_{2n}) \right| \leq \max \left\{ \frac{2|\mathcal{C}|}{n}, \frac{4|\mathcal{C}|}{n} \right\} = \frac{4|\mathcal{C}|}{n}.$$

$\square$

### A.4.1 Probabilistic Bound on Estimation Error for Gradient Estimators

In the following lemma, we establish a probabilistic bound for estimating a partial derivative of $\mathcal{L}_t$ with deviation $\varepsilon > 0$.

**Lemma A.21.** *Let* $\widetilde{\partial_{\sigma_{j,q}}} \mathcal{L}_t \left( \overrightarrow{\sigma}, \overrightarrow{\theta} \right)$ *and* $\widetilde{\partial_{\theta_j}} \mathcal{L}_t \left( \overrightarrow{\sigma}, \overrightarrow{\theta} \right)$ *denote the estimated partial derivative of* $\mathcal{L}_t$ *with respect to* $\sigma_{j,q}$ *and* $\theta_j$ *at* $(\overrightarrow{\sigma}, \overrightarrow{\theta})$, *which are generated by Algorithm 3 or Algorithm 4, respectively. Similarly, let* $\partial_{\sigma_{j,q}} \mathcal{L}_t \left( \overrightarrow{\sigma}, \overrightarrow{\theta} \right)$ *and* $\partial_{\theta_j} \mathcal{L}_t \left( \overrightarrow{\sigma}, \overrightarrow{\theta} \right)$ *denote the corresponding true partial*

*derivatives. Then, for all $\varepsilon > 0$, the probabilities* $\Pr\left[\left|\widetilde{\partial_{\sigma_{j,q}}}\mathcal{L}_t\left(\overrightarrow{\sigma},\overrightarrow{\theta}\right) - \partial_{\sigma_{j,q}}\mathcal{L}_t\left(\overrightarrow{\sigma},\overrightarrow{\theta}\right)\right| \geq \varepsilon\Big|t\right]$ *and* $\Pr\left[\left|\widetilde{\partial_{\theta_j}}\mathcal{L}_t\left(\overrightarrow{\sigma},\overrightarrow{\theta}\right) - \partial_{\theta_j}\mathcal{L}_t\left(\overrightarrow{\sigma},\overrightarrow{\theta}\right)\right| \geq \varepsilon\Big|t\right]$ *are both upper-bounded by* $2\exp\left\{\frac{-2\left((\Gamma_{\overrightarrow{\sigma}}^{-1})^2\cdot\varepsilon\right)^2\widetilde{N}}{32^2}\right\}$, *where* $\Gamma_{\overrightarrow{\sigma}}^{-1} = \exp\left\{-2\sum_{l=1}^{L}\sum_{k\in\mathcal{K}_l}\sigma_{l,k}\right\}$ *and* $\widetilde{N}$ *is a positive even integer.*

*Proof of Lemma A.21.* We begin by establishing the bound for $\Pr\left[\left|\widetilde{\partial_{\sigma_{j,q}}}\mathcal{L}_t\left(\overrightarrow{\sigma},\overrightarrow{\theta}\right) - \partial_{\sigma_{j,q}}\mathcal{L}_t\left(\overrightarrow{\sigma},\overrightarrow{\theta}\right)\right| \geq \varepsilon\Big|t\right]$. Since $\Gamma_{\overrightarrow{\sigma}}^{-1} > 0$, we have

$$\left|\widetilde{\partial_{\sigma_{j,q}}}\mathcal{L}_t\left(\overrightarrow{\sigma},\overrightarrow{\theta}\right) - \mathbb{E}\left[\widetilde{\partial_{\sigma_{j,q}}}\mathcal{L}_t\left(\overrightarrow{\sigma},\overrightarrow{\theta}\right)\right]\right| \geq \varepsilon$$
$$\iff (\Gamma_{\overrightarrow{\sigma}}^{-1})^2\cdot\left|\widetilde{\partial_{\sigma_{j,q}}}\mathcal{L}_t\left(\overrightarrow{\sigma},\overrightarrow{\theta}\right) - \mathbb{E}\left[\widetilde{\partial_{\sigma_{j,q}}}\mathcal{L}_t\left(\overrightarrow{\sigma},\overrightarrow{\theta}\right)\right]\right| \geq (\Gamma_{\overrightarrow{\sigma}}^{-1})^2\cdot\varepsilon.$$

Thus, the probability can be written as:

$$\Pr\left[\left|\widetilde{\partial_{\sigma_{j,q}}}\mathcal{L}_t\left(\overrightarrow{\sigma},\overrightarrow{\theta}\right) - \mathbb{E}\left[\widetilde{\partial_{\sigma_{j,q}}}\mathcal{L}_t\left(\overrightarrow{\sigma},\overrightarrow{\theta}\right)\right]\right| \geq \varepsilon\Big|t\right]$$
$$= \Pr\left[(\Gamma_{\overrightarrow{\sigma}}^{-1})^2\cdot\left|\widetilde{\partial_{\sigma_{j,q}}}\mathcal{L}_t\left(\overrightarrow{\sigma},\overrightarrow{\theta}\right) - \mathbb{E}\left[\widetilde{\partial_{\sigma_{j,q}}}\mathcal{L}_t\left(\overrightarrow{\sigma},\overrightarrow{\theta}\right)\right]\right| \geq (\Gamma_{\overrightarrow{\sigma}}^{-1})^2\cdot\varepsilon\Big|t\right]$$
$$= \Pr\left[\left|(\Gamma_{\overrightarrow{\sigma}}^{-1})^2\cdot\widetilde{\partial_{\sigma_{j,q}}}\mathcal{L}_t\left(\overrightarrow{\sigma},\overrightarrow{\theta}\right) - (\Gamma_{\overrightarrow{\sigma}}^{-1})^2\cdot\mathbb{E}\left[\widetilde{\partial_{\sigma_{j,q}}}\mathcal{L}_t\left(\overrightarrow{\sigma},\overrightarrow{\theta}\right)\right]\right| \geq (\Gamma_{\overrightarrow{\sigma}}^{-1})^2\cdot\varepsilon\Big|t\right].$$

Next, we aim to show that the random variable $(\Gamma_{\overrightarrow{\sigma}}^{-1})^2\cdot\widetilde{\partial_{\sigma_{j,q}}}\mathcal{L}_t\left(\overrightarrow{\sigma},\overrightarrow{\theta}\right)$ satisfies the bounded difference property. From the definitions in Algorithm 3, we observe that $\widetilde{\partial_{\sigma_{j,q}}}\mathcal{L}_t\left(\overrightarrow{\sigma},\overrightarrow{\theta}\right)$ can be re-written as a random map from $\{-\Gamma_{\overrightarrow{\sigma}},\Gamma_{\overrightarrow{\sigma}}\}^{\widetilde{N}}$ to $\mathbb{R}$ defined as the following:

$$\left(\widetilde{\partial_{\sigma_{j,q}}}\mathcal{L}_t\left(\overrightarrow{\sigma},\overrightarrow{\theta}\right)\right)\left(o_1^{(1)},\cdots,o_{\widetilde{N}/2}^{(1)},o_1^{(2,\sigma_{j,q})},\cdots,o_{\widetilde{N}/2}^{(2,\sigma_{j,q})}\right)$$
$$= (-4)\left(y_t - \frac{1}{\widetilde{N}/2}\sum_{\beta=1}^{\widetilde{N}/2}o_\beta^{(1)}\right)\left(\frac{1}{\widetilde{N}/2}\sum_{\beta=\widetilde{N}/2+1}^{\widetilde{N}}o_\beta^{(2,\sigma_{j,q})}\right).$$

From the definitions in Algorithm 2, we observe that both $o_\beta^{(1)}$ and $o_\beta^{(2,\sigma_{j,q})}$ are in the set $\{-\Gamma_{\overrightarrow{\sigma}},\Gamma_{\overrightarrow{\sigma}}\}$. This implies that the random variables $\Gamma_{\overrightarrow{\sigma}}^{-1}\cdot o_\beta^{(1)}$ and $\Gamma_{\overrightarrow{\sigma}}^{-1}\cdot o_\beta^{(2,\sigma_{j,q})}$ lie within $\{-1,1\}$. Consequently, $(\Gamma_{\overrightarrow{\sigma}}^{-1})^2\cdot\left(\widetilde{\partial_{\sigma_{j,q}}}\mathcal{L}_t\left(\overrightarrow{\sigma},\overrightarrow{\theta}\right)\right)$ becomes a random map from $\{-1,1\}^{\widetilde{N}}$ to $\mathbb{R}$. For convenience, let us re-express $(\Gamma_{\overrightarrow{\sigma}}^{-1})^2\cdot\left(\widetilde{\partial_{\sigma_{j,q}}}\mathcal{L}_t\left(\overrightarrow{\sigma},\overrightarrow{\theta}\right)\right)$ as a mapping $F:\{-1,1\}^{\widetilde{N}}\to\mathbb{R}$, defined as follows:

$$F\left(\mathbb{Y}_1^{(F)},\cdots,\mathbb{Y}_{\widetilde{N}/2}^{(F)},\cdots,\mathbb{Y}_{\widetilde{N}}^{(F)}\right) = (-4)\left(\Gamma_{\overrightarrow{\sigma}}^{-1}\cdot y_t - \frac{1}{\widetilde{N}/2}\sum_{\beta=1}^{\widetilde{N}/2}\mathbb{Y}_\beta^{(F)}\right)\left(\frac{1}{\widetilde{N}/2}\sum_{\beta=\widetilde{N}/2+1}^{\widetilde{N}}\mathbb{Y}_\beta^{(F)}\right).$$

where $\mathbb{Y}_\beta^{(F)} = \Gamma_{\overrightarrow{\sigma}}^{-1}\cdot o_\beta^{(1)}$ for $1\leq\beta\in\widetilde{N}/2$, and $\mathbb{Y}_\beta^{(F)} = \Gamma_{\overrightarrow{\sigma}}^{-1}\cdot o_\beta^{(2,\sigma_{j,q})}$ for $\widetilde{N}/2+1\leq\beta\in\widetilde{N}$. By applying Proposition A.20, we observe the following bound on the difference when changing a single input:

$$\max_{\mathbb{Y}_\zeta^{(F)},(\mathbb{Y}_\zeta^{(F)})'\in\{-1,1\}}\left|F\left(\mathbb{Y}_1^{(F)},\cdots,\mathbb{Y}_\zeta^{(F)},\cdots,\mathbb{Y}_{\widetilde{N}}^{(F)}\right) - F\left(\mathbb{Y}_1^{(F)},\cdots,(\mathbb{Y}_\zeta^{(F)})',\cdots,\mathbb{Y}_{\widetilde{N}}^{(F)}\right)\right|$$
$$\leq \frac{16}{\widetilde{N}/2} = \frac{32}{\widetilde{N}}.$$

This shows that changing a single input alters the value of $(\Gamma_{\overrightarrow{\sigma}}^{-1})^2\cdot\left(\widetilde{\partial_{\sigma_{j,q}}}\mathcal{L}_t\left(\overrightarrow{\sigma},\overrightarrow{\theta}\right)\right)$ by at most $\frac{32}{\widetilde{N}}$, meaning that $(\Gamma_{\overrightarrow{\sigma}}^{-1})^2\cdot\left(\widetilde{\partial_{\sigma_{j,q}}}\mathcal{L}_t\left(\overrightarrow{\sigma},\overrightarrow{\theta}\right)\right)$ satisfies the bounded differences property. As a result,

by applying McDiarmid's inequality [16], we know that for any $\varepsilon > 0$, we deduce,

$$\Pr\left[\left|(\Gamma_{\vec{\sigma}}^{-1})^2 \cdot \widetilde{\partial_{\sigma_{j,q}}}\mathcal{L}_t\left(\vec{\sigma}, \vec{\theta}\right) - (\Gamma_{\vec{\sigma}}^{-1})^2 \cdot \mathbb{E}\left[\widetilde{\partial_{\sigma_{j,q}}}\mathcal{L}_t\left(\vec{\sigma}, \vec{\theta}\right)\right]\right| \geq (\Gamma_{\vec{\sigma}}^{-1})^2 \cdot \varepsilon\,\Big|\,t\right]$$

$$\leq 2\exp\left\{-2\left((\Gamma_{\vec{\sigma}}^{-1})^2 \cdot \varepsilon\right)^2 \Big/ \left(\sum_{i=1}^{\widetilde{N}}\left(32/\widetilde{N}\right)^2\right)\right\}$$

$$= 2\exp\left\{\frac{-2\left((\Gamma_{\vec{\sigma}}^{-1})^2 \cdot \varepsilon\right)^2 \widetilde{N}}{32^2}\right\}.$$

By Proposition A.17, we note that

$$\mathbb{E}\left[\widetilde{\partial_{\sigma_{j,q}}}\mathcal{L}_t\left(\vec{\sigma}, \vec{\theta}\right)\right] = \partial_{\sigma_{j,q}}\mathcal{L}_t\left(\vec{\sigma}, \vec{\theta}\right).$$

Therefore, we obtain the following probability bound:

$$\Pr\left[\left|\widetilde{\partial_{\sigma_{j,q}}}\mathcal{L}_t\left(\vec{\sigma}, \vec{\theta}\right) - \partial_{\sigma_{j,q}}\mathcal{L}_t\left(\vec{\sigma}, \vec{\theta}\right)\right| \geq \varepsilon\,\Big|\,t\right] \leq 2\exp\left\{\frac{-2\left((\Gamma_{\vec{\sigma}}^{-1})^2 \cdot \varepsilon\right)^2 \widetilde{N}}{32^2}\right\}.$$

Using similar arguments as above, we identify $(\Gamma_{\vec{\sigma}}^{-1})^2 \cdot \widetilde{\partial_{\theta_j}}\mathcal{L}_t\left(\vec{\sigma}, \vec{\theta}\right)$ as a random mapping from $\{-1, 1\}^{\widetilde{N}}$ to $\mathbb{R}$. From Proposition A.20, we observe that changing a single input alters the value of $(\Gamma_{\vec{\sigma}}^{-1})^2 \cdot \widetilde{\partial_{\theta_j}}\mathcal{L}_t\left(\vec{\sigma}, \vec{\theta}\right)$ by at most $\frac{16}{\widetilde{N}} < \frac{32}{\widetilde{N}}$. Again, by applying McDiarmid's inequality and Proposition A.17, we obtain:

$$\Pr\left[\left|\widetilde{\partial_{\theta_j}}\mathcal{L}_t\left(\vec{\sigma}, \vec{\theta}\right) - \partial_{\theta_j}\mathcal{L}_t\left(\vec{\sigma}, \vec{\theta}\right)\right| \geq \varepsilon\right] \leq 2\exp\left\{\frac{-2\left((\Gamma_{\vec{\sigma}}^{-1})^2 \cdot \varepsilon\right)^2 \widetilde{N}}{32^2}\right\}.$$

$\square$

Now returning to the context of Algorithm 1, in each update round, exactly one $\vec{p}$ is sampled and passed into Algorithm 5. For simplicity, we define $\vec{p} \odot \widetilde{\nabla}\mathcal{L}_t\left(\vec{\sigma}, \vec{\theta}\right)$ as the vector of estimated partial derivatives generated by Algorithm 5. Given a particular update round $t$ and $\vec{p}$, the $\alpha^{th}$ entry of $\vec{p} \odot \widetilde{\nabla}\mathcal{L}_t\left(\vec{\sigma}, \vec{\theta}\right) \neq 0$ only when the $\alpha^{th}$ entry in $\vec{p}$ is 1. Thus, to quantify the estimation error, it suffices to measure the deviation between the non-zero entries of $\vec{p} \odot \widetilde{\nabla}\mathcal{L}_t\left(\vec{\sigma}, \vec{\theta}\right)$ and their corresponding true partial derivative.

Let $\vec{p} \odot \nabla\mathcal{L}_t\left(\vec{\sigma}, \vec{\theta}\right)$ denote the vector of the corresponding true partial derivatives. Here, $\nabla\mathcal{L}_t\left(\vec{\sigma}, \vec{\theta}\right)$ is the full true gradient of $\mathcal{L}_t$ at $\left(\vec{\sigma}, \vec{\theta}\right)$, and the value of each entry in $\vec{p} \odot \nabla\mathcal{L}_t\left(\vec{\sigma}, \vec{\theta}\right)$ is determined according to the entries of $\vec{p}$. In the following lemma, we establish a probabilistic bound for $\left\|\vec{p} \odot \widetilde{\nabla}\mathcal{L}_t\left(\vec{\sigma}, \vec{\theta}\right) - \vec{p} \odot \nabla\mathcal{L}_t\left(\vec{\sigma}, \vec{\theta}\right)\right\|_2$ with deviation $\varepsilon > 0$ conditioned on $\vec{p}$ and $t$.

**Lemma A.22.** *Let $p \leq \left|(\vec{\sigma}, \vec{\theta})\right|$ be a positive integer. For all $\varepsilon > 0$ and $\widetilde{N} \in (2\mathbb{Z})_{>0}$,*

$$\Pr\left[\left\|\vec{p} \odot \widetilde{\nabla}\mathcal{L}_t\left(\vec{\sigma}, \vec{\theta}\right) - \vec{p} \odot \nabla\mathcal{L}_t\left(\vec{\sigma}, \vec{\theta}\right)\right\|_2 \geq \varepsilon\,\Big|\,t, \vec{p}\right]$$

$$\leq \|\vec{p}\|_0 \cdot 2\exp\left\{\frac{-2\left((\Gamma_{\vec{\sigma}}^{-1})^2 \cdot \varepsilon\right)^2 \widetilde{N}}{32^2}\right\},$$

*where $\Gamma_{\vec{\sigma}}^{-1} = \exp\left\{-2\sum_{l=1}^{L}\sum_{k\in\mathcal{K}_l}\sigma_{l,k}\right\}$ and $\vec{p} \in \{0,1\}^{\left|(\vec{\sigma}, \vec{\theta})\right|}$.*

*Proof of Lemma A.22.* For simplicity and clarity, we now denote $\left(\overrightarrow{\sigma}, \overrightarrow{\theta}\right)$ by $w$ and the length of $w$ by $|w|$.

First, using the fact $\|x\|_\infty \le \|x\|_2$, we note that $\|x\|_\infty \ge \varepsilon \implies \|x\|_2 \ge \varepsilon$, which implies that:

$$\Pr\left[\left\|\overrightarrow{p} \odot \widetilde{\nabla} \mathcal{L}_t(w) - \overrightarrow{p} \odot \nabla \mathcal{L}_t(w)\right\|_2 \ge \varepsilon \Big| t, \overrightarrow{p}\right]$$

$$\le \Pr\left[\left\|\overrightarrow{p} \odot \widetilde{\nabla} \mathcal{L}_t(w) - \overrightarrow{p} \odot \nabla \mathcal{L}_t(w)\right\|_\infty \ge \varepsilon \Big| t, \overrightarrow{p}\right]$$

$$= \Pr\left[\max_{\substack{\alpha=1,\cdots,|w| \\ (\overrightarrow{p})_\alpha \ne 0}} \left|\left(\widetilde{\nabla}\mathcal{L}_t(w)\right)_\alpha - (\nabla\mathcal{L}_t(w))_\alpha\right| \ge \varepsilon \Big| t, \overrightarrow{p}\right]$$

$$\le \Pr\left[\exists \alpha \in \mathcal{I}_{\overrightarrow{p}} : \left|\left(\widetilde{\nabla}\mathcal{L}_t(w)\right)_\alpha - (\nabla\mathcal{L}_t(w))_\alpha\right| \ge \varepsilon \Big| t, \overrightarrow{p}\right],$$

where $\mathcal{I}_{\overrightarrow{p}} = \{\alpha \in [|w|] | (\overrightarrow{p})_\alpha \ne 0\}$. Applying the union bound, we have:

$$\Pr\left[\exists \alpha \in \mathcal{I}_{\overrightarrow{p}} : \left|\left(\widetilde{\nabla}\mathcal{L}_t(w)\right)_\alpha - (\nabla\mathcal{L}_t(w))_\alpha\right| \ge \varepsilon \Big| t, \overrightarrow{p}\right]$$

$$\le \sum_{\alpha \in \mathcal{I}_{\overrightarrow{p}}} \Pr\left[\left|\left(\widetilde{\nabla}\mathcal{L}_t(w)\right)_\alpha - (\nabla\mathcal{L}_t(w))_\alpha\right| \ge \varepsilon \Big| t\right]$$

Then, since $\left(\widetilde{\nabla}\mathcal{L}_t(w)\right)_\alpha$ are generated by either Algorithm 3 and Algorithm 4, by applying the Lemma A.21, we derive:

$$\Pr\left[\left|\left(\widetilde{\nabla}\mathcal{L}_t(w)\right)_\alpha - (\nabla\mathcal{L}_t(w))_\alpha\right| \ge \varepsilon \Big| t\right] \le 2\exp\left\{\frac{-2\left((\Gamma_{\overrightarrow{\sigma}}^{-1})^2 \cdot \varepsilon\right)^2 \widetilde{N}}{32^2}\right\}.$$

Combining all of the results above, we obtain:

$$\Pr\left[\left\|\overrightarrow{p} \odot \widetilde{\nabla} \mathcal{L}_t(w) - \overrightarrow{p} \odot \nabla \mathcal{L}_t(w)\right\|_2 \ge \varepsilon \Big| t, \overrightarrow{p}\right] \le \sum_{\alpha \in \mathcal{I}_{\overrightarrow{p}}} 2\exp\left\{\frac{-2\left((\Gamma_{\overrightarrow{\sigma}}^{-1})^2 \cdot \varepsilon\right)^2 \widetilde{N}}{32^2}\right\}$$

$$= 2\left|\mathcal{I}_{\overrightarrow{p}}\right| \cdot \exp\left\{\frac{-2\left((\Gamma_{\overrightarrow{\sigma}}^{-1})^2 \cdot \varepsilon\right)^2 \widetilde{N}}{32^2}\right\}.$$

By definition of $\mathcal{I}_{\overrightarrow{p}}$, we note that $\left|\mathcal{I}_{\overrightarrow{p}}\right| = \|\overrightarrow{p}\|_0$. Therefore,

$$\Pr\left[\left\|\overrightarrow{p} \odot \widetilde{\nabla} \mathcal{L}_t(w) - \overrightarrow{p} \odot \nabla \mathcal{L}_t(w)\right\|_2 \ge \varepsilon \Big| t, \overrightarrow{p}\right] \le 2\|\overrightarrow{p}\|_0 \cdot \exp\left\{\frac{-2\left((\Gamma_{\overrightarrow{\sigma}}^{-1})^2 \cdot \varepsilon\right)^2 \widetilde{N}}{32^2}\right\}.$$

$$\tag{31}$$

$\square$

### A.4.2 Expected Bound on the Total Number of Measurements

In the following theorem, we present an expected bound for the total number of measurements required to ensure that the deviation $\left\|\overrightarrow{p} \odot \widetilde{\nabla} \mathcal{L}_t\left(\overrightarrow{\sigma}, \overrightarrow{\theta}\right) - \overrightarrow{p} \odot \nabla \mathcal{L}_t\left(\overrightarrow{\sigma}, \overrightarrow{\theta}\right)\right\|_2$ remains within $\varepsilon$ with probability at least $1 - \delta$ over the distribution of $\overrightarrow{p}$.

**Theorem A.23.** *Given a positive integer $p \le \left|(\overrightarrow{\sigma}, \overrightarrow{\theta})\right|$, $\varepsilon > 0$, and $\delta \in (0, 1)$, the expected total number of measurements required for Algorithm 5 to estimate the gradient of $\mathcal{L}_t$ with deviation at most $\varepsilon$ and probability $1 - \delta$ over the distribution of $\overrightarrow{p}$ is at least*

$$\frac{\left|\left(\overrightarrow{\sigma}, \overrightarrow{\theta}\right)\right|}{p} \cdot \frac{32^2}{2\left((\Gamma_{\overrightarrow{\sigma}}^{-1})^2 \cdot \varepsilon\right)^2} \ln\left(\frac{2\left|\left(\overrightarrow{\sigma}, \overrightarrow{\theta}\right)\right|}{\delta p}\right)$$

*Proof of Theorem A.23.* For simplicity, we again denote $\left(\overrightarrow{\sigma}, \overrightarrow{\theta}\right)$ by $w$, where $|w|$ represents its length. Taking the expected value of Equation (31) over the distribution of $\overrightarrow{p}$, we observe:

$$\Pr\left[\left\|\overrightarrow{p} \odot \widetilde{\nabla}\mathcal{L}_t(w) - \overrightarrow{p} \odot \nabla\mathcal{L}_t(w)\right\|_2 \geq \varepsilon \middle| t\right]$$

$$= \sum_{\overrightarrow{p} \in \{0,1\}^{|w|}} \Pr\left[\left\|\overrightarrow{p} \odot \widetilde{\nabla}\mathcal{L}_t(w) - \overrightarrow{p} \odot \nabla\mathcal{L}_t(w)\right\|_2 \geq \varepsilon \middle| t, \overrightarrow{p}\right] \cdot \Pr\left[\overrightarrow{p}\right]$$

$$\leq 2\exp\left\{\frac{-2\left((\Gamma_{\overrightarrow{\sigma}}^{-1})^2 \cdot \varepsilon\right)^2 \widetilde{N}}{32^2}\right\} \cdot \left(\sum_{\overrightarrow{p} \in \{0,1\}^{|w|}} |\mathcal{I}_{\overrightarrow{p}}| \cdot \left(1 - \frac{1}{p}\right)^{|w| - |\mathcal{I}_{\overrightarrow{p}}|} \cdot \left(\frac{1}{p}\right)^{|\mathcal{I}_{\overrightarrow{p}}|}\right)$$

$$= 2\exp\left\{\frac{-2\left((\Gamma_{\overrightarrow{\sigma}}^{-1})^2 \cdot \varepsilon\right)^2 \widetilde{N}}{32^2}\right\} \cdot \left(\sum_{k=0}^{|w|} k \cdot \binom{|w|}{k} \cdot \left(1 - \frac{1}{p}\right)^{|w| - k} \cdot \left(\frac{1}{p}\right)^{k}\right)$$

Next, Using the identity $\sum_{i=0}^{n} i \cdot \binom{n}{i}(1 - 1/p)^{n-i}(1/p)^i = n/p$ for all $n \in \mathbb{Z}_{\geq 1}$ and $p \geq 1$, we arrive at:

$$\Pr\left[\left\|\overrightarrow{p} \odot \widetilde{\nabla}\mathcal{L}_t(w) - \overrightarrow{p} \odot \nabla\mathcal{L}_t(w)\right\|_2 \geq \varepsilon \middle| t\right] \leq 2\exp\left\{\frac{-2\left((\Gamma_{\overrightarrow{\sigma}}^{-1})^2 \cdot \varepsilon\right)^2 \widetilde{N}}{32^2}\right\} \cdot \frac{|w|}{p} \quad (32)$$

Then, let us determine the expected number of measurements required to estimate each selected partial derivative (i.e., $\widetilde{N}$) over the distribution of $\overrightarrow{p}$. Setting right hand side of Equation (32) to at most $\delta$ and solving for $\widetilde{N}$, we obtain:

$$\widetilde{N} \geq \frac{32^2}{2\left((\Gamma_{\overrightarrow{\sigma}}^{-1})^2 \cdot \varepsilon\right)^2} \ln\left(\frac{2|w|}{\delta p}\right)$$

Moreover, since each entry of $\overrightarrow{p}$ is sampled from a Bernoulli$(1/p)$ distribution, we know that $\mathbb{E}\left[\|\overrightarrow{p}\|_0\right] = \frac{|w|}{p}$. In other words, there are $|w|/p$ number of selected partial derivatives in expectation.

Combining the above results, we conclude that over the distribution of $\overrightarrow{p}$, the expected total number of measurements (i.e., $(|w|/p) \cdot \widetilde{N}$) required is at least:

$$\frac{|w|}{p} \cdot \frac{32^2}{2\left((\Gamma_{\overrightarrow{\sigma}}^{-1})^2 \cdot \varepsilon\right)^2} \ln\left(\frac{2|w|}{\delta p}\right) \quad (33)$$

$\square$

*Remark A.24.* Equation (33) suggests that a small $\varepsilon$ results in an exponential sample size. To mitigate this, one approach is to select $\varepsilon$ large enough to determine the direction of the gradient reliably.

To determine an appropriate value for $\varepsilon$, we consider the value of the partial derivatives at $\left(\overrightarrow{\sigma}, \overrightarrow{\theta}\right)$. From Equation (3), Equation (8), and Equation (10), we observe that the partial derivatives are asymptotically equal to $\Theta(\Gamma_{\overrightarrow{\sigma}}^2 x)$ for some $x \in [-1, 1]$ when $\left(\overrightarrow{\sigma}, \overrightarrow{\theta}\right)$ is not too close to a stationary point. Moreover, their directions are determined by $x$.

Thus, selecting $\varepsilon$ as $\Gamma_{\overrightarrow{\sigma}}^2 \cdot \kappa$ for some $\kappa \in (0, 1)$ suffices to ensure a reliable determination of the gradient direction. Consequently, the expected total number of measurements required becomes at least:

$$\frac{|w|}{p} \cdot \frac{32^2}{2\kappa^2} \ln\left(\frac{2|w|}{\delta p}\right) \quad (34)$$

## A.5 Convergence

### A.5.1 Preliminary

We begin with a review of fundamental results in calculus.

**Proposition A.25** (Extreme Value Theorem [18]). *Let $F : X \to Y$ be continuous, where $Y$ is an ordered set in the order topology. If $X$ is compact, then there exist points $c$ and $d$ in $X$ such that $F(c) \leq F(x) \leq F(d)$ for every $x \in X$.*

**Proposition A.26** (Estimation of Lipschitz constant on $C^1$ functions [13]). *Let $U \subseteq \mathbb{R}^n$ be an open subset, and suppose $F : U \to \mathbb{R}^m$ is of class $C^1$. Then $F$ is Lipschitz continuous on every compact convex subset $K \subset U$. The Lipschitz constant can be taken to be $\sup_{x \in K} \|\nabla F(x)\|_2$, where $\nabla F(x)$ is the Jacobian matrix.*

**Lemma A.27** (Decent Lemma). *If $F : \mathbb{R}^d \to \mathbb{R}$ is of class of $C^1$ and $K$ is any compact and convex subset of $\mathbb{R}^d$, then:*

$$F(y) \leq F(x) + \nabla F(x)(y - x) + \frac{L_K}{2} \|y - x\|_2^2,$$

*for all $x, y \in K$, where $L_K$ is the Lipschitz constant.*

*Sketch Proof of Lemma A.27.* By applying Proposition A.26, we know that $F$ is Lipschitz continuous on $K$. Furthermore, the Lipschitz constant $L_K$ is bounded by $\sup_{x \in K} \|\nabla F(x)\|$. Then, using the inequality established in the proof of Lemma 2.25 in [7], for any $x, y \in K$, we know that:

$$F(y) \leq F(x) + (\nabla F(x))^T (y - x) + \int_0^1 \|\nabla F(x + t(y - x)) - \nabla F(x)\|_2 \cdot \|y - x\|_2 \, dt.$$

Since $K$ is convex, we know that $x + t(y - x) \in K$ for all $t \in [0, 1]$. By the definition of local Lipschitz continuous, we can write:

$$\|\nabla F(x + t(y - x)) - \nabla F(x)\|_2 \leq L_K \|x + t(y - x) - x\|_2 = L_K t \|y - x\|_2,$$

for all $t \in [0, 1]$. Together, we conclude that:

$$\begin{aligned}
F(y) \leq & F(x) + (\nabla F(x))^T (y - x) + \int_0^1 L_K t \|y - x\|_2^2 \, dt \\
= & F(x) + (\nabla F(x))^T (y - x) + L_K \|y - x\|_2^2 \left( \int_0^1 t \, dt \right) \\
= & F(x) + (\nabla F(x))^T (y - x) + \frac{L_K}{2} \|y - x\|_2^2.
\end{aligned}$$

$\square$

Based on the results above, we present the following two properties of our objective function: Lemma A.28 and Lemma A.29.

### A.5.2 Properties of the Objective Function and Estimated Gradient

In this section, we present several important properties of the objective functions $\mathcal{L}$ and $\mathcal{L} + \mathcal{G}$, defined in Section 3, as well as the properties of the estimated gradient obtained from Algorithm 5.

**Lemma A.28.** *Let $\mathcal{L}(\overrightarrow{\sigma}, \overrightarrow{\theta})$ be the function defined in Equation (6). On the compact set $\mathcal{X} = \mathcal{X}_{sigma} \times \mathcal{X}_{theta}$, $\mathcal{L}$ is Lipschitz continuous.*

*Proof of Lemma A.28.* The proof of this lemma directly follows from Proposition A.26. Since maps $\frac{\partial \mathcal{L}}{\partial \sigma_{j,q}}$ given in Equation (10) and $\frac{\partial \mathcal{L}}{\partial \theta_j}$ given in Equation (8) are defined on $\mathbb{R}^{\left|\left(\overrightarrow{\sigma}, \overrightarrow{\theta}\right)\right|}$, we know $\mathcal{L}$ is of class $C^1$. Then, applying Proposition A.26, we know $\mathcal{L}$ is Lipschitz continuous on $\mathcal{X}$. $\square$

**Lemma A.29.** *$\mathcal{L}(\overrightarrow{\sigma}, \overrightarrow{\theta}) + \mathcal{G}(\overrightarrow{\sigma}, \overrightarrow{\theta})$ is lower-bounded on $\mathbb{R}^{\left|(\overrightarrow{\sigma}, \overrightarrow{\theta})\right|}$.*

*Proof of Lemma A.29.* Given the definition of $\mathcal{L}(\overrightarrow{\sigma}, \overrightarrow{\theta})$ and $\mathcal{G}(\overrightarrow{\sigma}, \overrightarrow{\theta})$, we know that $\mathcal{L}(\overrightarrow{\sigma}, \overrightarrow{\theta}) + \mathcal{G}(\overrightarrow{\sigma}, \overrightarrow{\theta})$ is finite and well-defined when $(\overrightarrow{\sigma}, \overrightarrow{\theta}) \in \mathcal{X}$. So, it suffices to check $\mathcal{L}(\overrightarrow{\sigma}, \overrightarrow{\theta}) + \mathcal{G}(\overrightarrow{\sigma}, \overrightarrow{\theta})$ is lower bounded on $\mathcal{X}$. Given the definition of $\mathcal{L}$ and $\mathcal{G}$, we know that $\mathcal{L}(\overrightarrow{\sigma}, \overrightarrow{\theta}) + \mathcal{G}(\overrightarrow{\sigma}, \overrightarrow{\theta}) = \mathcal{L}(\overrightarrow{\sigma}, \overrightarrow{\theta})$ on $\mathcal{X}$. Since $\mathcal{L}(\overrightarrow{\sigma}, \overrightarrow{\theta})$ is continuous and $\mathcal{X}$ is compact, by the extreme value theorem (Proposition A.25), we know $\mathcal{L}(\overrightarrow{\sigma}, \overrightarrow{\theta})$ is lower-bounded, which implies $\mathcal{L}(\overrightarrow{\sigma}, \overrightarrow{\theta}) + \mathcal{G}(\overrightarrow{\sigma}, \overrightarrow{\theta})$ is lower-bounded. $\square$

**Proposition A.30.** Let $\overrightarrow{p}_t \odot \widetilde{\nabla}\mathcal{L}_{i_t}\left(\overrightarrow{\sigma}, \overrightarrow{\theta}\right)$ be an estimation of gradient of $\mathcal{L}_{i_t}$ at some fixed point $\left(\overrightarrow{\sigma}, \overrightarrow{\theta}\right)$ generated by Algorithm 5, where $i_t \sim \text{uniform}\{1, \cdots, |\mathcal{D}|\}$ and $\overrightarrow{p}_t \sim (\text{Bernoulli}(1/p))^{\left|\left(\overrightarrow{\sigma}, \overrightarrow{\theta}\right)\right|}$. Then, $\mathbb{V}\left[\overrightarrow{p}_t \cdot \widetilde{\nabla}\mathcal{L}_{i_t}\left(\overrightarrow{\sigma}, \overrightarrow{\theta}\right)\right] = \mathbb{E}\left[\left\|\overrightarrow{p}_t \cdot \widetilde{\nabla}\mathcal{L}_{i_t}\left(\overrightarrow{\sigma}, \overrightarrow{\theta}\right) - \frac{1}{p} \cdot \nabla\mathcal{L}\left(\overrightarrow{\sigma}, \overrightarrow{\theta}\right)\right\|_2^2\right]$ is bounded.

*Proof of Proposition A.30.* For clarity, let $w$ denote $\left(\overrightarrow{\sigma}, \overrightarrow{\theta}\right)$. First, by taking the full expectation of $\overrightarrow{p}_t \odot \widetilde{\nabla}\mathcal{L}_{i_t}\left(\overrightarrow{\sigma}, \overrightarrow{\theta}\right)$ using Lemma A.18, we obtain:

$$\mathbb{E}\left[\overrightarrow{p}_t \odot \widetilde{\nabla}\mathcal{L}_{i_t}(w)\right] = \frac{1}{|\mathcal{D}|}\sum_{i_t=1}^{|\mathcal{D}|}\mathbb{E}\left[\overrightarrow{p}_t \odot \widetilde{\nabla}\mathcal{L}_{i_t}(w)\Big|i_t\right] = \frac{1}{|\mathcal{D}|}\sum_{i_t=1}^{|\mathcal{D}|}\frac{1}{p}\cdot\nabla\mathcal{L}_{i_t}(w) = \frac{1}{p}\nabla\mathcal{L}(w),$$

(35)

which leads to:

$$\mathbb{E}\left[\left\|\overrightarrow{p}_t \odot \widetilde{\nabla}\mathcal{L}_{i_t}(w) - \mathbb{E}\left[\overrightarrow{p}_t \odot \widetilde{\nabla}\mathcal{L}_{i_t}(w)\right]\right\|_2^2\right]$$

$$=\mathbb{E}\left[\sum_{\alpha=1}^{|w|}\left(\left(\overrightarrow{p}_t \odot \widetilde{\nabla}\mathcal{L}_{i_t}(w)\right)_\alpha - \left(\frac{1}{p}\nabla\mathcal{L}(w)\right)_\alpha\right)^2\right]$$

$$=\sum_{\alpha=1}^{|w|}\mathbb{E}\left[\left(\left(\overrightarrow{p}_t \odot \widetilde{\nabla}\mathcal{L}_{i_t}(w)\right)_\alpha - \left(\frac{1}{p}\nabla\mathcal{L}(w)\right)_\alpha\right)^2\right]$$

$$=\sum_{\alpha=1}^{|w|}\text{var}\left[\left(\overrightarrow{p}_t \odot \widetilde{\nabla}\mathcal{L}_{i_t}(w)\right)_\alpha\right].$$

Here, $(\cdot)_\alpha$ represents the $\alpha^{th}$ component of the input vector. From the observation of Algorithm 3, Algorithm 4, and Algorithm 5, we note that $\left(\overrightarrow{p}_t \odot \widetilde{\nabla}\mathcal{L}_{i_t}(w)\right)_\alpha : \{-\Gamma_{\overrightarrow{\sigma}}, \Gamma_{\overrightarrow{\sigma}}\}^{\widetilde{N}} \to \mathbb{R}$ defined by

$$\left(\overrightarrow{p}_t \odot \widetilde{\nabla}\mathcal{L}_{i_t}(w)\right)_\alpha\left(o_1, \cdots, o_{\widetilde{N}/2}, o_{\widetilde{N}/2+1}, \cdots, o_{\widetilde{N}}\right)$$

$$= -(\overrightarrow{p}_t)_\alpha \cdot \mathcal{C}\left(y_{i_t} - \frac{1}{\widetilde{N}/2}\sum_{\beta=1}^{\widetilde{N}/2}o_\beta\right)\left(\frac{1}{\widetilde{N}/2}\sum_{\beta=\widetilde{N}/2+1}^{\widetilde{N}}o_\beta\right),$$

where $\mathcal{C}$ is 2 or 4 and $\Gamma_{\overrightarrow{\sigma}} = \exp\left\{\sum_{j,k}\sigma_{j,k}\right\}$ and $\widetilde{N}$ is an even number. Since the mapping $\left(\overrightarrow{p}_t \odot \widetilde{\nabla}\mathcal{L}_{i_t}(w)\right)_\alpha$ has a finite domain, we note that:

$$\max_{o_\beta, o'_\beta \in \{-\Gamma_{\overrightarrow{\sigma}}, \Gamma_{\overrightarrow{\sigma}}\}}\left|\left(\overrightarrow{p}_t \odot \widetilde{\nabla}\mathcal{L}_{i_t}(w)\right)_\alpha\left(o_1, \cdots, o_\beta, \cdots, o_{\widetilde{N}}\right)\right.$$

$$\left. - \left(\overrightarrow{p}_t \odot \widetilde{\nabla}\mathcal{L}_{i_t}(w)\right)_\alpha\left(o_1, \cdots, o'_\beta, \cdots, o_{\widetilde{N}}\right)\right|$$

is finite for all $i_t$ and $\overrightarrow{p}_t$. Define

$$\mathfrak{U} = \max_{\substack{o_\beta, o'_\beta \in \{-\Gamma_{\overrightarrow{\sigma}}, \Gamma_{\overrightarrow{\sigma}}\} \\ \overrightarrow{p}_t \in \{0,1\}^{|w|} \\ i_t = 1, \cdots, |\mathcal{D}|}}\left|\left(\overrightarrow{p}_t \odot \widetilde{\nabla}\mathcal{L}_{i_t}(w)\right)_\alpha\left(o_1, \cdots, o_\beta, \cdots, o_{\widetilde{N}}\right)\right. \tag{36}$$

$$\left. - \left(\overrightarrow{p}_t \odot \widetilde{\nabla}\mathcal{L}_{i_t}(w)\right)_\alpha\left(o_1, \cdots, o'_\beta, \cdots, o_{\widetilde{N}}\right)\right|.$$

Using the Efron-Stetin inequality, we obtain

$$\text{var}\left[\left(\overrightarrow{p}_t \odot \widetilde{\nabla}\mathcal{L}_{i_t}(w)\right)_\alpha\right]$$

$$\leq \frac{1}{2}\sum_{\beta=1}^{\widetilde{N}}\mathbb{E}\left[\left(\left(\overrightarrow{p}_t \odot \widetilde{\nabla}\mathcal{L}_{i_t}(w)\right)_\alpha(o_1,\cdots,o_\beta,\cdots,o_{\widetilde{N}})\right.\right.$$

$$\left.\left.-\left(\overrightarrow{p}_t \odot \widetilde{\nabla}\mathcal{L}_{i_t}(w)\right)_\alpha(o_1,\cdots,o'_\beta,\cdots,o_{\widetilde{N}})\right)^2\right]$$

$$\leq \frac{1}{2}\sum_{\beta=1}^{\widetilde{N}}\mathfrak{U}^2 = \frac{\widetilde{N}\cdot\mathfrak{U}^2}{2}.$$

Together, we have

$$\mathbb{V}\left[\overrightarrow{p}_t \odot \widetilde{\nabla}\mathcal{L}_{i_t}(w)\right] \leq |w|\cdot\frac{\widetilde{N}\cdot\mathfrak{U}^2}{2} < \infty.$$

$\square$

### A.5.3 Convergence bound of the Learning Algorithm

We now present the convergence bound in the following theorem.

**Theorem A.31.** *Let $\mathcal{L}(\overrightarrow{\sigma},\overrightarrow{\theta}) + \mathcal{G}(\overrightarrow{\sigma},\overrightarrow{\theta})$ be the objective function, where $\mathcal{L}$ and $\mathcal{G}$ are defined in Equation (6) and Equation (7). Let $\left\{w_t = \left(\overrightarrow{\sigma}^{(t)},\overrightarrow{\theta}^{(t)}\right), t = 1,\cdots,T\right\}$ be a sequence of parameters generated by the proximal-SGD (Algorithm 1). Suppose $w_0$ and $\overline{w}$ are the initial point and a stationary point in $\mathcal{X} = \mathcal{X}_{sigma}\times\mathcal{X}_{theta}$, respectively. Let $\{\eta_t|0 < \eta_t < 1, t = 0,\cdots,T\}$ be a sequence such that $\sum_{t=0}^\infty \eta_t < \infty$. Then,*

$$\left(\frac{1}{\sum_{t=1}^T\eta_t}\sum_{t=1}^T\eta_t\mathbb{E}\left[\mathcal{L}(w_t)\right]\right) - \mathcal{L}(\overline{w}) \leq \frac{p\cdot\|w_0-\overline{w}\|_2^2}{2\sum_{t=1}^T\eta_t} + \frac{3L_\mathcal{X}\sum_{t=1}^T\eta_t\mathfrak{B}_{max}}{2\sum_{t=1}^T\eta_t} + \frac{p\sum_{t=1}^T\eta_t^2\mathcal{V}}{2\sum_{t=1}^T\eta_t},$$

*where $\mathcal{V} = \sup_{w_t}\mathbb{V}\left[\overrightarrow{p}_t\odot\widetilde{\nabla}\mathcal{L}_{i_t}(w_t)\right]$, $p$ is a positive integer, $L_\mathcal{X}$ is the local Lipschitz constant, and $\mathfrak{B}_{max} = \sum_{l=1}^L\sum_{k\in\mathcal{K}_l}\left(\mathfrak{B}^{(l,k)}\right)^2 + \sum_{\substack{l=1\\b_l=1}}\left(\mathfrak{B}_1^{(l)} - \mathfrak{B}_0^{(l)}\right)^2.$*

*Proof of Theorem A.31.* For simplicity and clarity, we define $w_t = \left(\overrightarrow{\sigma}_t,\overrightarrow{\theta}_t\right)$, $\mathcal{X} = \mathcal{X}_{sigma}\times\mathcal{X}_{theta}$, and $\overrightarrow{p}_t\odot\widetilde{\nabla}\mathcal{L}_{i_t}(w_t)$ denotes the gradient estimation for $\mathcal{L}_{i_t}(w_t) = (y_{i_t} - \text{tr}(\mathcal{M}\mathcal{U}_R(\rho_{i_t})))^2$ generated by Algorithm 5. Here, $i_t \sim \text{uniform}\{1,2,\cdots,|\mathcal{D}|\}$ and $\overrightarrow{p}_t \sim (\text{Bernoulli}(1/p))^{|w_t|}$. By definition, we know $w_{t+1} = \text{prox}_\mathcal{G}\left(w_t - \eta_t\overrightarrow{p}_t\odot\widetilde{\nabla}\mathcal{L}_{i_t}(w_t)\right)$, which implies that $w_t\in\mathcal{X}$ for all $t = 0,\cdots,T$. Applying lemma 8.17 from [7], we obtain the following result:

$$\|w_{t+1}-\overline{w}\|_2^2 = \left\|\text{prox}_\mathcal{G}\left(w_t - \eta_t\overrightarrow{p}_t\odot\widetilde{\nabla}\mathcal{L}_{i_t}(w_t)\right) - \text{prox}_\mathcal{G}(\overline{w})\right\|_2^2$$

$$\leq \left\|w_t - \eta_t\overrightarrow{p}_t\odot\widetilde{\nabla}\mathcal{L}_{i_t}(w_t) - \overline{w}\right\|_2^2.$$

Using the fact $\|x-y\|_2^2 = (x-y)^T(x-y) = \|x\|_2^2 - 2x^Ty + \|y\|_2^2$, we can expand the squared norm as below:

$$\left\|w_t - \eta_t\overrightarrow{p}_t\odot\widetilde{\nabla}\mathcal{L}_{i_t}(w_t) - \overline{w}\right\|_2^2$$

$$= \|w_t-\overline{w}\|_2^2 - 2\eta_t\left(\overrightarrow{p}_t\odot\widetilde{\nabla}\mathcal{L}_{i_t}(w_t)\right)^T(w_t-\overline{w}) + \eta_t^2\left\|\overrightarrow{p}_t\odot\widetilde{\nabla}\mathcal{L}_{i_t}(w_t)\right\|_2^2.$$

Together, we have

$$\|w_{t+1}-\overline{w}\|_2^2 - \|w_t-\overline{w}\|_2^2 \leq -2\eta_t\left(\overrightarrow{p}_t\odot\widetilde{\nabla}\mathcal{L}_{i_t}(w_t)\right)^T(w_t-\overline{w}) + \eta_t^2\left\|\overrightarrow{p}_t\odot\widetilde{\nabla}\mathcal{L}_{i_t}(w_t)\right\|_2^2.$$

Now, taking the expectation conditioned on $w_t$, we get

$$\mathbb{E}\left[\left\|w_{t+1} - \overline{w}\right\|_2^2 - \left\|w_t - \overline{w}\right\|_2^2 \middle| w_t\right]$$

$$\leq \mathbb{E}\left[-2\eta_t\left(\overrightarrow{p}_t \odot \widetilde{\nabla}\mathcal{L}_{i_t}(w_t)\right)^T (w_t - \overline{w}) + \eta_t^2 \left\|\overrightarrow{p}_t \odot \widetilde{\nabla}\mathcal{L}_{i_t}(w_t)\right\|_2^2 \middle| w_t\right]$$

$$= -2\eta_t\mathbb{E}\left[\left(\overrightarrow{p}_t \odot \widetilde{\nabla}\mathcal{L}_{i_t}(w_t)\right)^T (w_t - \overline{w}) \middle| w_t\right] + \eta_t^2\mathbb{E}\left[\left\|\overrightarrow{p}_t \odot \widetilde{\nabla}\mathcal{L}_{i_t}(w_t)\right\|_2^2 \middle| w_t\right]. \qquad (37)$$

For the first term in Equation (37), using Proposition A.17 and linearity of gradient, we arrive at:

$$\mathbb{E}\left[\left(\overrightarrow{p}_t \odot \widetilde{\nabla}\mathcal{L}_{i_t}(w_t)\right)^T (w_t - \overline{w}) \middle| w_t\right]$$

$$= \sum_{\overrightarrow{p}_t \in \{0,1\}^{|w_t|}}\left(\frac{1}{|\mathcal{D}|}\sum_{i_t=1}^{|\mathcal{D}|}\mathbb{E}\left[\left(\overrightarrow{p}_t \odot \widetilde{\nabla}\mathcal{L}_{i_t}(w_t)\right)^T (w_t - \overline{w}) \middle| w_t, i_t, \overrightarrow{p}_t\right]\right) \cdot \Pr\left[\overrightarrow{p}_t\right]$$

$$= \sum_{\overrightarrow{p}_t \in \{0,1\}^{|w_t|}}\left(\frac{1}{|\mathcal{D}|}\sum_{i_t=1}^{|\mathcal{D}|}(\overrightarrow{p}_t \odot \nabla\mathcal{L}_{i_t}(w_t))^T (w_t - \overline{w})\right) \cdot \Pr\left[\overrightarrow{p}_t\right]$$

$$= \sum_{\overrightarrow{p}_t \in \{0,1\}^{|w_t|}}(\overrightarrow{p}_t \odot \nabla\mathcal{L}(w_t))^T (w_t - \overline{w}) \cdot \Pr\left[\overrightarrow{p}_t\right]$$

$$= \left(\left(\sum_{\overrightarrow{p}_t \in \{0,1\}^{|w_t|}}\overrightarrow{p}_t \cdot \Pr\left[\overrightarrow{p}_t\right]\right) \odot \nabla\mathcal{L}(w_t)\right)^T (w_t - \overline{w})$$

$$= \frac{1}{p} \cdot (\nabla\mathcal{L}(w_t))^T (w_t - \overline{w}).$$

For the second term in Equation (37), by Proposition A.30, we know that the value $\mathbb{V}\left[\overrightarrow{p}_t \odot \widetilde{\nabla}\mathcal{L}_{i_t}(w_t)\right]$ conditioned on $w_t$ is bounded. This implies that $\mathbb{E}\left[\left\|\overrightarrow{p}_t \odot \widetilde{\nabla}\mathcal{L}_{i_t}(w_t)\right\|_2^2 \middle| w_t\right]$ is also bounded, and specifically we have the inequality:

$$\mathbb{E}\left[\left\|\overrightarrow{p}_t \odot \widetilde{\nabla}\mathcal{L}_{i_t}(w_t)\right\|_2^2 \middle| w_t\right] \leq \mathbb{V}\left[\overrightarrow{p}_t \odot \widetilde{\nabla}\mathcal{L}_{i_t}(w_t)\right].$$

Therefore, we have

$$\mathbb{E}\left[\left\|w_{t+1} - \overline{w}\right\|_2^2 - \left\|w_t - \overline{w}\right\|_2^2 \middle| w_t\right] \leq -\frac{2\eta_t}{p}(\nabla\mathcal{L}(w_t))^T (w_t - \overline{w}) + \eta_t^2\mathbb{V}\left[\overrightarrow{p}_t \odot \widetilde{\nabla}\mathcal{L}_{i_t}(w_t)\right].$$

By combining all the above and taking the full expectation, we get:

$$\mathbb{E}\left[\left\|w_{t+1} - \overline{w}\right\|_2^2 - \left\|w_t - \overline{w}\right\|_2^2\right] = \mathbb{E}\left[\mathbb{E}\left[\left\|w_{t+1} - \overline{w}\right\|_2^2 - \left\|w_t - \overline{w}\right\|_2^2 \middle| w_t\right]\right]$$

$$\leq \mathbb{E}\left[-\frac{2\eta_t}{p}(\nabla\mathcal{L}(w_t))^T (w_t - \overline{w})\right] + \eta_t^2\mathbb{E}\left[\mathbb{V}\left[\overrightarrow{p}_t \odot \widetilde{\nabla}\mathcal{L}_{i_t}(w_t)\right]\right]$$

$$\leq \mathbb{E}\left[-\frac{2\eta_t}{p}(\nabla\mathcal{L}(w_t))^T (w_t - \overline{w})\right] + \eta_t^2\mathcal{V},$$

where $\mathcal{V} = \sup_{w_t}\mathbb{V}\left[\overrightarrow{p}_t \odot \widetilde{\nabla}\mathcal{L}_{i_t}(w_t)\right]$. Telescoping the both sides, we obtain:

$$\sum_{t=1}^{T}\mathbb{E}\left[\left\|w_t - \overline{w}\right\|_2^2 - \left\|w_{t-1} - \overline{w}\right\|_2^2\right] = \mathbb{E}\left[\left\|w_T - \overline{w}\right\|_2^2 - \left\|w_0 - \overline{w}\right\|_2^2\right]$$

$$\leq -\sum_{t=1}^{T}\frac{2\eta_t}{p}\mathbb{E}\left[(\nabla\mathcal{L}(w_t))^T (w_t - \overline{w})\right] + \sum_{t=1}^{T}\eta_t^2\mathcal{V}$$

$$= \sum_{t=1}^{T}\frac{2\eta_t}{p}\mathbb{E}\left[(\nabla\mathcal{L}(w_t))^T (\overline{w} - w_t)\right] + \sum_{t=1}^{T}\eta_t^2\mathcal{V}. \qquad (38)$$

Next, let us bound $(\nabla\mathcal{L}(w_t))^T (\overline{w} - w_t)$. Using the fact that $x^T y = \|x\| \|y\| \cos\theta$ for any $x, y$, we observe that:

$$(\nabla\mathcal{L}(w_t) - \nabla\mathcal{L}(\overline{w}))^T (\overline{w} - w_t) \leq \|\nabla\mathcal{L}(w_t) - \nabla\mathcal{L}(\overline{w})\|_2 \cdot \|\overline{w} - w_t\|_2 .$$

On the left-hand side, we have:

$$(\nabla\mathcal{L}(w_t) - \nabla\mathcal{L}(\overline{w}))^T (\overline{w} - w_t) = (\nabla\mathcal{L}(w_t))^T (\overline{w} - w_t) - (\nabla\mathcal{L}(\overline{w}))^T (\overline{w} - w_t).$$

On the right hand side, since $\mathcal{L}$ is locally Lipschitz continuous on $\mathcal{X}$, by definition, we know that:

$$\|\nabla\mathcal{L}(w_t) - \nabla\mathcal{L}(\overline{w})\|_2 \cdot \|\overline{w} - w_t\|_2 \leq L_\mathcal{X} \|\overline{w} - w_t\|_2^2 .$$

Combining and re-arranging terms, we get:

$$(\nabla\mathcal{L}(w_t))^T (\overline{w} - w_t) \leq (\nabla\mathcal{L}(\overline{w}))^T (\overline{w} - w_t) + L_\mathcal{X} \|\overline{w} - w_t\|_2^2 .$$

Furthermore, applying the decent lemma (Lemma A.27), we have:

$$(\nabla\mathcal{L}(w_t))^T (\overline{w} - w_t) \leq \mathcal{L}(\overline{w}) - \mathcal{L}(w_t) + \frac{3L_\mathcal{X}}{2} \|\overline{w} - w_t\|_2^2 . \tag{39}$$

Now, combining Equation (38) and Equation (39), we obtain:

$$\mathbb{E}\left[ \|w_T - \overline{w}\|_2^2 - \|w_0 - \overline{w}\|_2^2 \right] \leq \sum_{t=1}^{T} \frac{2\eta_t}{p} \mathbb{E}\left[ \mathcal{L}(\overline{w}) - \mathcal{L}(w_t) + \frac{3L_\mathcal{X}}{2} \|\overline{w} - w_t\|_2^2 \right] + \sum_{t=1}^{T} \eta_t^2 \mathcal{V},$$

which is equivalent to

$$\sum_{t=1}^{T} \frac{2\eta_t}{p} \left( \mathbb{E}[\mathcal{L}(w_t)] - \mathcal{L}(\overline{w}) \right)$$

$$\leq \|w_0 - \overline{w}\|_2^2 - \mathbb{E}\left[ \|w_T - \overline{w}\|_2^2 \right] + \frac{3L_\mathcal{X}}{p} \sum_{t=1}^{T} \eta_t \mathbb{E}\left[ \|\overline{w} - w_t\|_2^2 \right] + \sum_{t=1}^{T} \eta_t^2 \mathcal{V}$$

$$\leq \|w_0 - \overline{w}\|_2^2 + \frac{3L_\mathcal{X}}{p} \sum_{t=1}^{T} \eta_t \mathbb{E}\left[ \|\overline{w} - w_t\|_2^2 \right] + \sum_{t=1}^{T} \eta_t^2 \mathcal{V}. \tag{40}$$

Since $w_t$ and $\overline{w}$ are both in $\mathcal{X}$ and $\mathcal{X}$ is a hyper-rectangle, we know that:

$$\|\overline{w} - w_t\|_2^2 \leq \sum_{l=1}^{L} \sum_{k \in \mathcal{K}_l} \left( \mathfrak{B}^{(l,k)} \right)^2 + \sum_{\substack{l=1 \\ b_l=1}} \left( \mathfrak{B}_1^{(l)} - \mathfrak{B}_0^{(l)} \right)^2 = \mathfrak{B}_{max}.$$

Thus, Equation (40) can be written as:

$$\sum_{t=1}^{T} \frac{2\eta_t}{p} \left( \mathbb{E}[\mathcal{L}(w_t)] - \mathcal{L}(\overline{w}) \right) \leq \|w_0 - \overline{w}\|_2^2 + \frac{3L_\mathcal{X}}{p} \cdot \sum_{t=1}^{T} \eta_t \mathfrak{B}_{max} + \sum_{t=1}^{T} \eta_t^2 \mathcal{V}.$$

Finally, dividing both sides of the inequality by $\frac{2\sum_{t=1}^{T} \eta_t}{p}$, we obtain:

$$\left( \frac{1}{\sum_{t=1}^{T} \eta_t} \sum_{t=1}^{T} \eta_t \mathbb{E}[\mathcal{L}(w_t)] \right) - \mathcal{L}(\overline{w}) \leq \frac{p \cdot \|w_0 - \overline{w}\|_2^2}{2\sum_{t=1}^{T} \eta_t} + \frac{3L_\mathcal{X} \sum_{t=1}^{T} \eta_t \mathfrak{B}_{max}}{2\sum_{t=1}^{T} \eta_t} + \frac{p\sum_{t=1}^{T} \eta_t^2 \mathcal{V}}{2\sum_{t=1}^{T} \eta_t}.$$

$\square$

## A.6 Additional Experiments

### A.6.1 Evaluation on Fashion-MNIST Dataset

We evaluate our approach on the Fashion-MNIST dataset [32] (Pullover vs. Shirt; 5000 training samples; standard test split; three seeds). Each $28 \times 28$ image is standardized, reduced by PCA to a compact feature vector (64 dimensions for our 6-qubit scenario), and $\ell_2$-normalized to unit length.

In simulation, we directly initialize the statevector to the corresponding amplitude encoded state $|\psi(x)\rangle$, then apply a 6-qubit, two-layer HEA with parameterized $R_X, R_Y, R_Z$ rotations and a circular CNOT entangler as in Figure 3. The labels are assigned to $-1$ (shirt) and $+1$ (pullover). This direct initialization avoids extra state preparation depth and associated gate noise, so our comparisons isolate training-time noise and mitigation effects. Besides, the initial learning rate ($\eta^{(1)}$) is selected using grid search across a wide range and then refined with binary search for better precision. Subsequently, the learning rate is annealed monotonically.

Under static noise, our inverse learning method initially lags Van den Berg et al.'s pre-characterized approach but surpasses it mid-training, achieving a lower final loss (Figure 5a). The noiseless PQC provides the best-case reference, while the baseline noisy PQC yields the highest error, confirming that both mitigation strategies effectively reduce noise impact. Consistent with these trends, Table 2 summarizes the test accuracies of the compared approaches. Under dynamic noise, we emulate hardware drift by slightly increasing the error rates of three CNOT gates early in training. While Van den Berg et al.'s method retains its fixed calibration, our approach updates the inverse parameters online and achieves lower MSE by tracking the drift (Figure 5b). This highlights the practical advantage of adaptive inverse learning over pre-calibrated approaches in realistic quantum hardware conditions. Finally, probabilistic subsampling analysis using the same static noise profile as Figure 5a, reveals the expected convergence degradation with increased sparsity ($p = 1 > p = 2 > p = 4$), with denser sampling providing more stable optimization, consistent with the MNIST results (Figure 5c).

Table 2: Classification accuracies across Different methods for Fashion-MNIST dataset

|  | ACCURACY |
| --- | --- |
| NOISELESS PQC | $78.87 \pm 1.32$ |
| NOISY PQC (BASELINE) | $69.57 \pm 3.35$ |
| MITIGATED PQC (VAN DEN BERG ET.AL) | $72.37 \pm 2.53$ |
| **OURS** | $74.23 \pm 2.84$ |

### A.6.2 Additional Results—PEC and ZNE under Dynamic Noise

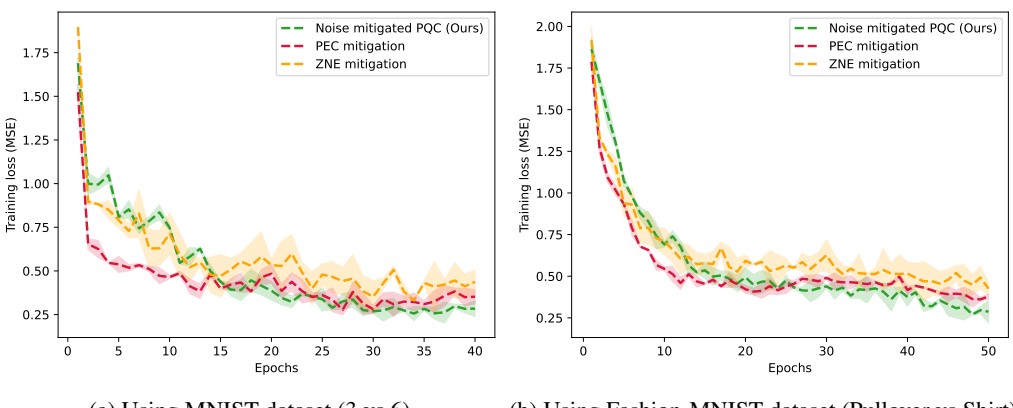

(a) Using MNIST dataset (3 vs 6)  (b) Using Fashion-MNIST dataset (Pullover vs Shirt)

Figure 6: Training loss comparison among PEC, ZNE and our mitigation framework for binary classification task under dynamic noise. Here, each epoch summarizes 50 iterations of the training process.

Under dynamic noise, our adaptive inverse noise learning method updates the inverse noise parameters at every iteration and consistently achieves the lowest final MSE on both MNIST (Figure 6a) and Fashion-MNIST (Figure 6b). PEC often looks competitive because it applies the full inverse from a pre-calibrated noise map, but it has practical limits: (i) it is measurement intensive: the shot demand multiplies across mitigated gates, leading to an effective exponential growth with circuit depth; (ii) it does not track drift without recalibration; and (iii) it loses accuracy as the device moves further away from the calibration point. ZNE performs worst on average with higher variance, because

extrapolation assumes stable noise scaling, which breaks under time-varying noise. The convergence of our method across both datasets supports the conclusion that adaptive inverse noise learning is important for stable optimization on realistic, drifting hardware- capabilities that neither fixed PEC calibrations nor ZNE's static extrapolation provide without manual intervention.

