# OpenReview forum: "Robust Integrated Learning and Pauli Noise Mitigation for Parametrized Quantum Circuits"
_NeurIPS.cc/2025/Conference — NeurIPS 2025 poster_

### Official Review · Reviewer_cSqD · 2025-06-11

**Clarity:** 2
**Significance:** 2
**Originality:** 3
**Rating:** 4
**Confidence:** 3

**Summary:**

This paper presents a gradient-based framework for training parameterized quantum circuits (PQCs) in the presence of Pauli noise, a dominant source of gate errors in noisy intermediate-scale quantum (NISQ) devices. The proposed method integrates noise mitigation directly into the training process by simultaneously learning PQC parameters and the parameters of an inverse noise model. The inverse model is based on a sparse Pauli-Lindblad representation, enabling efficient mitigation of Pauli noise via a dynamically updated, task-specific strategy inspired by the Probabilistic Error Cancellation (PEC) protocol. To estimate gradients of both PQC and inverse noise parameters avoiding the need for full noise tomography, the algorithm leverages stochastic subsampling and proximal stochastic gradient descent (proximal SGD) to ensure stable convergence while maintaining computational efficiency. Convergence and sample complexity bounds are established theoretically. The method is evaluated on a binary classification task using amplitude-encoded MNIST data. Comparisons are made with a noiseless PQC, a noisy PQC without mitigation, and a benchmark derived from Van Den Berg et al.'s pre-characterized noise model.

**Questions:**

I hope the authors to address the major concerns raised in the Strengths And Weaknesses part.

**Ethical Concerns:**

["NO or VERY MINOR ethics concerns only"]

**Final Justification:**

I thank the authors for the follow-up explanations. I think the authors addressed my concerns by pointing out that the absence of barren plateaus is not necessary and showing how to address the SPAM errors. Given that the authors will add the numerical comparison and remove the unnecessary assumptions in the revised manuscript, I believe this work a natural fit for NeurIPS 2025 and will raise my score accordingly.

**Limitations:**

yes

**Quality:**

3

**Strengths And Weaknesses:**

From my perspective, this work investigates the interesting topic of optimizing noisy parametrized quantum circuits. The assumption that the noises in the circuit are Pauli noise is practical in experiments. And the idea to combine error mitigation and parametrized circuit optimization can be interesting. However, I think this paper has a distance from the bar of NeurIPS 2025 due to the following major concerns.

The first weakness lies in the motivation for the work. There has been a series of works on optimizing quantum circuits in the presence of noise in a variational style. Some of these works focused on regarding noise as a part of the circuit and optimizing the parameters, and some others focus on searching for noise-resilient (parametrization of) quantum circuits. The authors fail to include these works in the literature review, and answer why we should focus on quantum error mitigation to correct (mitigate) the noise instead of the previous approaches. If possible, there could also be some numerical comparison between these frameworks.

The second weakness is the practical applicability of the proposed algorithm. In Eq. (5), the authors just directly optimize over all the parameters in Pauli noise and the parametrized gates directly. I am wondering if this simple construction leads to avoidable additional assumptions. As mentioned in the future works part, the framework assumes noiseless single-qubit operations, Hermitian linear observables with only $\pm 1$ eigenvalue, absence of barren plateau, and omits SPAM (state preparation and measurement) noise. These assumptions should significantly restrict the usage of the algorithm on near-term devices that suffer from SPAM noise. In addition, assuming the absence of barren plateau would rule out the applicability of the algorithm to brickwork parametrized circuits unless the depth of the circuit is very shallow, the circuit is assumed to have some stringent structure, or the final observable is local.

---

> ### Author Rebuttal · Authors · 2025-07-30
>
> We thank the reviewer for the thoughtful critique.
>
> ## Responses to specific questions:
>
> ### The first weakness lies in the motivation for the work. There has been a series of works on optimizing quantum circuits in the presence of noise in a variational style. Some of these works focused on regarding noise as a part of the circuit and optimizing the parameters, and some others focus on searching for noise-resilient (parametrization of) quantum circuits. The authors fail to include these works in the literature review, and answer why we should focus on quantum error mitigation to correct (mitigate) the noise instead of the previous approaches. If possible, there could also be some numerical comparison between these frameworks.
>
> While the strategies proposed in prior efforts have demonstrated potential, they suffer from the following key limitations.
>
> First, the ansätze employed in many of these methods are typically shallow, which limits their expressive power. Our work aims to address this limitation directly. As noise propagates through the circuit, wide and deep ansätze often give rise to *noise-induced barren plateaus* in the optimization landscape. This phenomenon significantly increases the computational cost of training, often rendering such efforts practically infeasible. Consequently, effective noise mitigation strategies are essential for scaling to deeper circuit architectures.
>
> Second, prior methods often rely on the assumption of a static noise model to ensure the stability of training and model performance. For instance, noise-aware circuit learning approaches such as [1] optimize quantum circuits under fixed, tomographically characterized noise models. As a result, any drift in the underlying hardware noise necessitates re-tomography and full re-optimization. Similarly, noise-resilient parameterization methods like [2] demonstrate that optimal parameters remain unchanged under stationary Pauli channels, but do not correct for the bias introduced by noise.
>
> Our method addresses this limitation by allowing dynamic recalibration. Specifically, when the structure of the noise model changes, our framework enables efficient adaptation without requiring full retraining. Instead, training can resume from the current parameter state, providing a more robust and practical solution in real-world settings, where noise models often vary over time. This capability is empirically demonstrated in Figure 3b, where our method outperforms static mitigation baseline despite only a small subset of CNOT gates being perturbed under a dynamic noise model—suggesting its potential to handle larger or more frequent noise drifts in realistic scenarios.
>
> ### The second weakness is the practical applicability of the proposed algorithm. In Eq. (5), the authors just directly optimize over all the parameters in Pauli noise and the parametrized gates directly. I am wondering if this simple construction leads to avoidable additional assumptions. As mentioned in the future works part, the framework assumes noiseless single-qubit operations, Hermitian linear observables with only eigenvalues, absence of barren plateau, and omits SPAM (state preparation and measurement) noise. These assumptions should significantly restrict the usage of the algorithm on near-term devices that suffer from SPAM noise. In addition, assuming the absence of barren plateau would rule out the applicability of the algorithm to brickwork parametrized circuits unless the depth of the circuit is very shallow, the circuit is assumed to have some stringent structure, or the final observable is local.
>
> Regarding other noise sources, such as SPAM (State Preparation and Measurement) errors, we acknowledge their impact on overall performance. However, our focus in this work is on gate-level noise. SPAM errors have been extensively studied in the literature, with numerous effective mitigation techniques available---see, for example, [3, 4]. These methods are often implemented in pre-processing or post-processing stages, and therefore, for the purposes of this study, we assume the absence of SPAM errors.
>
> With respect to barren plateaus, all gradient based optimizers suffer from this problem. Specifically, barren plateaus are characterized by the conditions
> $$
>     \mathbb{E}[\partial\_\theta C] = 0 \quad \text{and} \quad \text{Var}[\partial\_\theta C] \leq \mathcal{O}(1/b^n)
> $$
> for some $ b \geq 2 $, $n \geq 1$, where $C$ is the objective function. These conditions imply that an exponential number of measurements is required to estimate gradients unless the initialization is extremely close to a (local or global) minimum---a scenario that is both rare and difficult to identify in practice.
>
> Finally, we would like to emphasize that the challenges mentioned at the end of the critique are inherent to *any* method that attempts to train variational or parameterized quantum circuits. Indeed, training such circuits is generally NP-hard, as shown in [5]. Our work aims to make this problem more tractable under realistic noise assumptions by focusing on scalable gradient estimation and mitigation of noise effects.
>
> ### References:
> [1] Cincio, L., Rudinger, K., Sarovar, M., & Coles, P. J. (2021). Machine learning of noise-resilient quantum circuits. PRX Quantum, 2(1), 010324.
>
> [2] Sharma, K., Khatri, S., Cerezo, M., & Coles, P. J. (2020). Noise resilience of variational quantum compiling. New Journal of Physics, 22(4), 043006.
>
> [3] Bravyi, S., Sheldon, S., Kandala, A., Mckay, D. C., & Gambetta, J. M. (2021). Mitigating measurement errors in multiqubit experiments. Physical Review A, 103(4), 042605.
>
> [4] Van Den Berg, E., Minev, Z. K., & Temme, K. (2022). Model-free readout-error mitigation for quantum expectation values. Physical Review A, 105(3), 032620.
>
> [5] Bittel, L., & Kliesch, M. (2021). Training variational quantum algorithms is NP-hard. Physical review letters, 127(12), 120502.

---

> > ### Comment · Reviewer_cSqD · 2025-08-02
> >
> > I thank the authors for the detailed explanations. Regarding the explanation for the first point, I think the authors have addressed my concern. I agree with the authors that the approach provided in this work is better at dealing with dynamic noise than the previous approaches such as PEC and ZNE. However, I would like to clarify my previous comment on the numerical part. I was actually wondering if the authors can also provide some numerical experiments to show that the algorithm proposed performs better than PEC and ZNE for dynamic noise other than the existing ones that compare between the proposed algorithm and Van Den Berg’s sparse Pauli Lindblad method.
> >
> > Regarding the explanation for the second point, I still believe that the main weakness of this work lies in making too many assumptions. Despite the assumption regarding noiseless Pauli operations and the absence of SPAM errors, it seems to me that the existence of barren plateaus does not necessarily indicate the hardness of training variational quantum circuits, as Ref. [5] only provides a worst-case result. Also, VQAs aim at finding local minima instead of global minima in many scenarios. The authors mentioned that SPAM errors have been extensively studied in the literature. However, it seems unclear how to combine these approaches to mitigate the SPAM errors with the approach proposed in this paper.

---

> > > ### Author Response · Authors · 2025-08-02
> > >
> > > We thank the reviewer for the insightful follow-up.
> > >
> > > We will include an additional experiment in the revised manuscript to compare our approach with PEC and ZNE.
> > >
> > > With regard to barren plateaus, we agree with the reviewer that the assumption of their absence is not needed for our proof or methods. It is important to emphasize that our theoretical results remain valid regardless of whether the objective function exhibits barren plateaus. This is because the derivations do not depend on the absence of barren plateaus and therefore continue to reflect the theoretical performance of our method under general conditions. We include this assumption only to highlight a well-known worst-case consideration: estimating gradients in the presence of barren plateaus may require exponential measurement effort. However, we acknowledge that this assumption is not essential to our contribution and will remove it in the revised manuscript.
> > >
> > > Additionally, we reiterate that our results do not claim convergence to a global minimum. Rather, we establish that the parameters will converge to a (local) stationary point within the bounds of the provided theoretical guarantees.
> > >
> > > Regarding SPAM error mitigation, SPAM (state-preparation and measurement) errors occur strictly before and after the noisy gate layer on which our inverse-noise learning method operates. They can be mitigated independently through a lightweight calibration process that is orthogonal to the learning loop and does not interfere with gradient estimation.
> > >
> > >
> > > To address state-preparation errors, a common strategy is post-selection. Suppose we use a unitary gate $U$ to evolve the initial quantum state $\left|0, \cdots, 0\right\rangle$ into the desired input state $\left|\varphi\right\rangle = U\left|0, \cdots, 0\right\rangle$. To implement post-selection, we can use two $n$-qubit quantum registers, denoted $A$ and $B$, and initialize them in an entangled state:
> > > $$
> > > \rho\_{AB} = \sum\_{i=0}^{2^n - 1} \alpha_i \left|i\right\rangle\langle i|\_A \otimes \left|i\right\rangle\langle i|\_B.
> > > $$
> > > We then apply the unitary $U$ to register $A$, resulting in the state:
> > > $$
> > > (U\otimes I) \rho_{AB} (U \otimes I)^\dagger = \sum_{i=0}^{2^n - 1} \alpha_i U\left|i\right\rangle\langle i|_A U^\dagger \otimes \left|i\right\rangle\langle i|_B.
> > > $$
> > > Finally, we measure register $B$ in the computational basis. If the measurement outcome is $\left|0, \cdots, 0\right\rangle$, we accept the trial and proceed with the computation. Otherwise, the trial is discarded. This post-selection scheme effectively filters out certain types of state-preparation errors by leveraging entanglement and conditional acceptance.
> > >
> > > For measurement errors, we may estimate a read-out confusion (error) matrix $R$ by preparing computational-basis states and measuring them using low-depth calibration circuits, as described in [3] and widely implemented in Qiskit and Mitiq. A regularized inverse $R^{-1}$ is then stored and applied as a classical post-processing step to correct the measurement error. Since $R^{-1}$ is static and linear map is extracted from independent calibration routines, this does not alter the gradient computation or introduce any additional learnable parameters. The corrections can be incorporated as classical post-processing and do not require changes to our algorithm.
> > >
> > > In this manner SPAM error mitigation is completely complementary to our technique.

---

> > > > ### Comment · Reviewer_cSqD · 2025-08-03
> > > >
> > > > I thank the authors for the follow-up explanations. I think the authors addressed my second point by pointing out that the absence of barren plateaus is not necessary and showing how to address the SPAM errors. Given that the authors will add the numerical comparison and remove the unnecessary assumptions in the revised manuscript, I believe this work a natural fit for NeurIPS 2025 and will raise my score accordingly.

---

### Official Review · Reviewer_sqmh · 2025-06-29

**Clarity:** 4
**Significance:** 3
**Originality:** 3
**Rating:** 4
**Confidence:** 2

**Summary:**

This paper presents a novel gradient-based framework for training Parameterized Quantum Circuits on noisy intermediate-scale quantum devices under Pauli noise. The key innovation is the integration of learning both the model parameters and an inverse noise channel within a unified optimization loop. The authors propose an efficient gradient estimation technique to scale training under noisy conditions and validate their method on the MNIST dataset.

**Questions:**

1.	Please see the Weaknesses.
2.	Can the proposed method be applied to multi-class classification tasks?
3.	How sensitive may the method be to the listed assumptions? For example, if SPAM noise exists, how may it influence the derivation process and the performance?

**Ethical Concerns:**

["NO or VERY MINOR ethics concerns only"]

**Final Justification:**

I have carefully considered author responses and also other reviews. I thank the authors for addressing some of my concerns. This is a good paper. I think I will keep my score.

**Limitations:**

yes

**Paper Formatting Concerns:**

no formatting issues

**Quality:**

3

**Strengths And Weaknesses:**

Strengths:
•	Joint optimization of PQC and noise mitigation parameters is novel and well-justified.
•	Theoretical derivations and results are illustrated in a rigorous manner.
•	Experimental results demonstrate the proposed method’s strong performance.
Weaknesses:
•	Strong assumptions are applied, assuming idealized conditions such as noiseless Pauli operations and the absence of SPAM errors.
•	Experiments are limited to the MNIST dataset, potentially limiting the generalibility of the proposed method.
•	Missing ablations on the locality parameter c.

---

> ### Author Rebuttal · Authors · 2025-07-30
>
> Many thanks for the thoughtful and insightful comments.
>
> ## Responses to specific questions:
>
> ### Q1: Addressing Weaknesses
>
> ### Strong assumptions are applied, assuming idealized conditions such as noiseless Pauli operations and the absence of SPAM errors.
>
> Regarding noiseless Pauli operations, we make this assumption based on the fact that Pauli gates generally exhibit lower error rates among one-qubit gates on modern NISQ hardware. Moreover, strategies such as Pauli Check Sandwiching [3] have been developed to further reduce their associated noise. That said, we recognize that Pauli gates are not fault-tolerant in the rigorous sense and that their noise can still influence overall performance. These second order considerations are addressed in subsequent efforts.
>
> Other types of noise, such as SPAM (State Preparation and Measurement) errors, can impact overall performance. However, SPAM errors have been studied in literature, and effective mitigation techniques have been proposed [1, 2]. These strategies are typically applied during pre-processing or post-processing stages, and are orthogonal and complementary to our method.
>
> With regard to barren plateaus, we note that all gradient based optimizers suffer from this problem, which limits the scale and scope of variational methods. However, even within this limited scope, faults pose a major challenge -- which our work addresses and provides important solutions for. We note that our solutions are independent of the barren plateaus problem and will be highly relevant even when the problem is resolved.
>
> ### Experiments are limited to the MNIST dataset, potentially limiting the generalibility of the proposed method.
>
> We acknowledge the limitation of focusing solely on MNIST, and we plan to evaluate our approach on more complex datasets in the final manuscript.
>
> ### Missing ablations on the locality parameter c.
> We note that our theoretical results hold for any positive integer $c \leq n$. In our work, we specifically chose $c = 2$ for the following reasons:
>
>   i) Any $n$-qubit unitary can be constructed from a quantum circuit composed of CNOT and single-qubit gates, which are universal. Thus, $c = 2$ is general enough to represent arbitrary ansätze.
>
>    ii) Two-local gates tend to experience lower noise, which makes the optimization problem more tractable.
>
>    iii) Two-local gates are more practical to implement on current NISQ devices.
>
>
> ### Q2: Can the proposed method be applied to multi-class classification tasks?
>
> One can generalize the measurement scheme to a positive operator-valued measure (POVM). In this case, the objective function has to be modified so that the model outputs the correct label with high probability. However, implementing POVMs is generally more difficult both in terms of design and hardware realization.
>
> ### Q3. How sensitive may the method be to the listed assumptions? For example, if SPAM noise exists, how may it influence the derivation process and the performance?
>
> As mentioned earlier, SPAM errors can be mitigated in pre-processing or post-processing stages. The resulting performance under such errors thus depends on the quality of the chosen mitigation strategy. As for the other assumptions, relaxing the $c$-locality or barren plateau assumptions does not affect the mathematical derivations themselves, since they do not rely on these assumptions. However, the time complexity of the algorithm could increase substantially, and lifting the assumption of a constrained parameter space could result in an objective function that is no longer Lipschitz continuous, in which case the convergence guarantees no longer hold.
>
> We thank the reviewer again for the constructive feedback and the opportunity to clarify these points.
>
> ### References:
> [1] Bravyi, S., Sheldon, S., Kandala, A., Mckay, D. C., & Gambetta, J. M. (2021). Mitigating measurement errors in multiqubit experiments. Physical Review A, 103(4), 042605.
>
> [2] Van Den Berg, E., Minev, Z. K., & Temme, K. (2022). Model-free readout-error mitigation for quantum expectation values. Physical Review A, 105(3), 032620.
>
> [3] Gonzales, A., Shaydulin, R., Saleem, Z. H., & Suchara, M. (2023). Quantum error mitigation by Pauli check sandwiching. Scientific Reports, 13(1), 2122.

---

> > ### Comment · Reviewer_sqmh · 2025-08-08
> >
> > Thanks for addressing my comments and providing your rebuttal. I think this is an interesting paper, and I support it. Evaluation across other datasets will definitely enhance the paper.

---

> > > ### Author Response · Authors · 2025-08-09
> > > **Thanks!**
> > >
> > > We thank the reviewer for the positive assessment and if there is anything else we can do to motivate her/him to increase their score, we are happy to do so!

---

> ### Author Response · Authors · 2025-08-05
>
> Dear Reviewer,
>
> Thank you again for your review and insightful feedback. We hope our rebuttal addressed your main concerns. If any clarification is still needed, or if you'd like to discuss any points further, we would be very happy to engage.
>
> We sincerely appreciate your time during this discussion phase.
>
> Best Regards.

---

### Official Review · Reviewer_qjyC · 2025-06-30

**Clarity:** 3
**Significance:** 3
**Originality:** 3
**Rating:** 4
**Confidence:** 4

**Summary:**

Parameterized quantum circuits (PQCs) are a popular method in both machine learning and more general quantum computing for utilizing near and long term hardware in a way that is flexible and compatible with resources. A key challenge in using these models in the near term is the influence of noise on the resulting quality of the solution or results.  Previous methods for mitigating these errors have relied upon expensive or unstable extrapolations, or fitting of noise models that could later be approximately inverted.  This work addresses this problem in an integrated fashion, but integrating the noise corrections into the variational circuit itself and learning them alongside the parameters of the variational model.  In principle, this is not only conceptually simpler, but also more accurate due to the interplay between the variational flexibility in the parameters and the noise inversion process.  They test this method on a number of model problems and work out the theory in detail.

**Questions:**

1. Can you clarify under what situations its possible to enforce that the inverse operations do not violate variational principles such that they can be used in a general setting without worry?  Is this possible to efficiently do in all cases? What are the limits?  A clear articulation of this in the paper would help boost the significance and clarity of the work

2. It is stated that the operations that are inserted are not physical in the most general case, as they sometimes involve sampling operations etc.  However I do believe there is a subset of these that could be made physical using ancilla qubits and/or classical communication to strengthen the variational ansatz as well as mitigate noise.  Could you elaborate on when this might be possible vs not and what the significance could be more broadly?

3. In a fault tolerant setting where we might assume local, random noise is quite small - how do you envision this framework enhancing the power of your circuits? How might the training approach be different?

**Ethical Concerns:**

["NO or VERY MINOR ethics concerns only"]

**Final Justification:**

Having reviewed all the final points, i believe this approach is still conceptually novel and perhaps useful in the near term.  I believe adding the conditions on the inverse operator to the main text will be useful in improving clarity, but perhaps uses of the method in fault tolerance still remain unclear.   Overall I would lean towards accepting this paper.

**Limitations:**

I feel the paper would be more clear if the authors addressed the tension between variational freedom in the non-physical parameters and general variational principle violations that could occur due to sampling from effectively non-real states.  This would help understand the applicability and potential limitations for more general applications.

**Quality:**

3

**Strengths And Weaknesses:**

Strengths

The concept of integrating the noise mitigation into the overall variational circuit I feel is conceptually novel and interesting.  The integration of all the trainable parameters into a single framework has a way of simplifying how one understands the overall process and could develop further generalizations.  The performance the method gets appears comparable to existing methods while also being new.

I feel this type of framework could be used to also generalize to non-unitary operations more generally being included inside PQCs for purposes other than error mitigation, though that is perhaps the topic of future work.

Weaknesses

While the method is conceptually interesting, perhaps the greatest weakness is the modest performance increases for the relative implementation and theory complexity increases.  Having non-physical operations within a parameterized circuit can run the risk of optimizing towards unphysical or non-realizable solution, running the variational nature of certain calculations.  This is addressed in this work by the objective function component "G" but a more clear discussion of this type of limitation would make the paper easier to read and understand how general the approach actually is.

---

> ### Author Rebuttal · Authors · 2025-07-30
>
> Many thanks for the thoughtful and insightful comments.
>
> With respect to the observed modest performance improvement, this is indeed the case for static noise models. However, the key benefit of our approach is in dynamic noise models. In this case, the improvements from our method are substantial. In particular, in Figure 3b, only three CNOT gates were perturbed under a dynamic noise model, yet our method outperformed the static mitigation baseline—suggesting that larger or more frequent dynamic noise drifts could yield even greater improvements.
> Notably, one of the key take-aways from our experiments is that our approach, while targeted to dynamic settings, yields slightly better performance even in the static setting.
>
> There are two main reasons for introducing the constraint component $\mathcal{G}$. First, it restricts the search space to a convex feasible set and ensures that the parameters used in the inverse or recovery operation do not diverge. Second, it guarantees that the objective function is Lipschitz continuous over the domain. Without this constraint (i.e., removing $\mathcal{G}$), the objective function becomes non-globally Lipschitz continuous due to the presence of an exponential normalizing factor. This lack of global Lipschitz continuity implies that theoretical guarantees for convergence may no longer hold. Nevertheless, we observe that the objective function remains *locally* Lipschitz continuous. It is possible to generalize our convergence results to any convex and compact (i.e., closed and bounded) parameter space.
>
> ## Responses to specific questions:
>
> ### Q1:  Can you clarify under what situations its possible to enforce that the inverse operations do not violate variational principles such that they can be used in a general setting without worry? Is this possible to efficiently do in all cases? What are the limits? A clear articulation of this in the paper would help boost the significance and clarity of the work.
>
> In order for the inverse channel $\Lambda^{-1}$ not to violate variational principles for **all** density operators $\rho$, $\Lambda^{-1}(\Lambda(\rho))$ must also be a valid density operator. In our work, we define:
> $$
> \Lambda^{-1}(\rho) = q \rho - (1 - q) P \rho P,
> $$
> and then
> $$
> \mathrm{tr}(\Lambda^{-1}(\rho)) = q \, \mathrm{tr}(\rho) - (1 - q) \, \mathrm{tr}(P \rho P) = 2q - 1,
> $$
> which equals 1 only if $q = 1$. Therefore, in general, $\Lambda^{-1}$ is not trace-preserving unless it is the identity map, which implies that no non-trivial $\Lambda^{-1}$ will satisfy variational principles for all $\rho$.
>
> However, the composite map $\Lambda_l^{-1} \circ \Lambda_l$ can be physical under specific conditions. Using the definitions of $\Lambda_l$ and $\Lambda_l^{-1}$ (Equations 2 and 3 in our paper), together with the commuting property (Proposition A.10), we can rewrite the composite as:
>
> $$
> \Lambda_l^{-1} \circ \Lambda_l = \bigcirc\_{k \in \mathcal{K}\_l} \left( \frac{1 + e^{2(\sigma\_{l,k} - \lambda\_{l,k})}}{2} I \cdot I + \frac{1 - e^{2(\sigma\_{l,k} - \lambda\_{l,k})}}{2} P\_k \cdot P\_k \right),
> $$
> where $\lambda_{l,k}$ are noise parameters and $\sigma_{l,k}$ are inverse noise parameters. It follows that $\Lambda_l^{-1} \circ \Lambda_l$ is implicitly a valid quantum channel if $\sigma_{l,k} \leq \lambda_{l,k}$ for all $l \in [L]$, $k \in \mathcal{K}\_l$. Unfortunately, verifying this inequality without prior knowledge of $\lambda_{l,k}$ may require exponential effort.
>
> We are happy to add this discussion to the manuscript.
>
>
> ### Q2: It is stated that the operations that are inserted are not physical in the most general case, as they sometimes involve sampling operations etc. However I do believe there is a subset of these that could be made physical using ancilla qubits and/or classical communication to strengthen the variational ansatz as well as mitigate noise. Could you elaborate on when this might be possible vs not and what the significance could be more broadly?
>
> The major difficulty in implementing $\Lambda^{-1}$ on quantum hardware lies in designing a quantum circuit $U$ that simulates the behavior of the non-physical term $-P \cdot P$, where $P$ is a Pauli operator. Specifically, such a circuit would need to satisfy the condition $U \equiv iP$ and simultaneously $U^\dagger \equiv iP$. However, this leads to a contradiction: since $iP$ is skew-Hermitian, we have $(iP)^\dagger = -iP$. Therefore, $U$ cannot be equivalent to $iP$ and Hermitian/unitary at the same time. As a result, we believe that there is no standard quantum circuit capable of directly implementing such an operation.
>
> We specifically chose $\Lambda^{-1}$ to be non-physical because it is the only general inverse/recovery map for Pauli-Lindbladian noise. Suppose we attempt to construct a quantum channel $\mathcal{E} = q I \cdot I + (1 - q) P \cdot P$ that inverts $\Lambda = w I \cdot I + (1 - w) P \cdot P$. Then their composition is given by:
> $$
> \mathcal{E} \circ \Lambda = (1 - q - w + 2 w q) I \cdot I + (q + w - 2 w q) P \cdot P.
> $$
> For this composition to equal the identity channel, we must have:
> $$
> q + w - 2 w q = 0.
> $$
> This equation has only two solutions: $q = w = 0$ or $q = w = 1$. Thus, $\Lambda$ must be either $I \cdot I$ or $P \cdot P$---not a convex combination---to admit an inverse of the form $\mathcal{E}$. Therefore, we conclude that no physical (CPTP) inverse $\Lambda^{-1}$ exists in general for Pauli noise.
>
>
> ### Q3: In a fault tolerant setting where we might assume local, random noise is quite small - how do you envision this framework enhancing the power of your circuits? How might the training approach be different?
>
> While fault-tolerant settings reduce error rates significantly, noise propagation still occurs. As noise accumulates across wide and deep circuits, the resulting optimization landscape can exhibit *noise-induced barren plateaus*, characterized by vanishing gradients and exponential sample complexity. This makes the training of such ansätz computationally expensive or infeasible. Consequently, effective noise mitigation remains a critical requirement for scaling variational models to deeper architectures, even in near fault-tolerant regimes.

---

> > ### Comment · Reviewer_qjyC · 2025-08-04
> >
> > I believe the added exposition on when the inverse operation and physical and the effect on variational approaches will help to improve the clarity, and will thus improve my score along that axis.  I'm not sure I agree with the comment that in fault tolerant settings this type of noise will be prevalent given the ability to exponentially reduce errors.  However, overall I still lean towards this being an interesting paper that would be interesting to the audience of NeurIPS.

---

### Official Review · Reviewer_BRsB · 2025-07-06

**Clarity:** 3
**Significance:** 4
**Originality:** 3
**Rating:** 4
**Confidence:** 3

**Summary:**

This paper introduces a method to train parameterized quantum circuits (PQCs) in a way that accounts for Pauli noise that occurs during gate operations. The authors propose jointly learning an inverse noise model(constructed using a
sparse Lindblad-based formulation) alongside the usual PQC parameters so that training can adapt to the noise in a more effective way as compared to existing noise mitigation strategies like ZNE and PEC.
- While gradients for the PQC parameters can be computed efficiently using the parameter shift rule, estimating gradients for the inverse noise parameters is more difficult because it scales exponentially with the number of qubits. To tackle this, the paper presents a new gradient estimation technique designed specifically for these noise parameters, which helps make the overall optimization more scalable and robust.

- The work provides comprehensive derivations of the gradient estimators, analysis of sample complexity and convergence, and numerical experiments showing how the method performs in practice.

**Questions:**

- Of the assumptions you list, which do you think will pose the biggest challenge for applying your method to real hardware?
- Do you see a path to testing this framework on current NISQ hardware, beyond classical simulation? What specific obstacles would you expect to encounter in those experiments?

**Ethical Concerns:**

["NO or VERY MINOR ethics concerns only"]

**Final Justification:**

This paper presents a scalable joint optimization approach to mitigate Pauli noise in PQC training, with a novel gradient estimation method, supporting theory, and empirical results showing advantages over baselines. I would like to recommend acceptance for this work.

**Limitations:**

yes

**Quality:**

3

**Strengths And Weaknesses:**

Strengths:

- The paper addresses an important practical challenge in quantum machine learning: mitigating Pauli noise during PQC training. It proposes a joint optimization framework that integrates learning an inverse noise model with PQC parameter tuning, which is a thoughtful and promising approach. This approach mitigates the need to apply predetermined noise models, making it scalable.

- It introduces a novel gradient estimation method for the inverse noise parameters, tackling the scalability issues that arise as qubit counts grow. Clear derivations of gradients and explainations of how they can be efficiently estimated are provided.

- Includes theoretical analyses of sample complexity and convergence.

- Demonstrates the approach empirically with numerical experiments that illustrate its potential advantages over the relevant baseline.

- The paper is generally well-organized and clearly structured.


Weaknesses:
- A major weakness is the making a large number of assumptions(see Section 3.1.2) to make the problem tractable

---

> ### Author Rebuttal · Authors · 2025-07-30
>
> We sincerely thank the reviewer for the thoughtful and insightful comments.
>
> The reviewer makes two critical observations. With respect to other sources of noise, in particular, SPAM noise, we note that effective mitigation techniques have been proposed in prior work [1, 2]. These strategies can often be applied during pre-processing or post-processing stages, and are orthogonal and complementary to our core contribution.
>
> With regard to barren plateaus, we note that all gradient based optimizers suffer from this problem, which limits the scale and scope of variational methods. However, even within this limited scope, gate-level noise pose a major challenge -- which our work addresses and provides important solutions for. We note that our solutions are independent of the barren plateaus problem and will be highly relevant even when the problem is resolved.
>
> ## Responses to specific questions:
>
>
> ### Q1: Of the assumptions you list, which do you think will pose the biggest challenge for applying your method to real hardware?
>
> The assumption on single-qubit noiseless Pauli operators ( X, Y, Z) poses one of the most significant challenges for real quantum hardware. While Pauli gates in NISQ devices generally have low noise, the inverses or recovery operations are also susceptible to noise. These small errors can propagate, particularly in deeper and wider circuits, compromising the effectiveness of noise mitigation. Techniques to reduce this impact, such as Pauli Check Sandwiching [3], play an important role and are being investigated in our current work.
>
>
> ### Q2: Do you see a path to testing this framework on current NISQ hardware, beyond classical simulation? What specific obstacles would you expect to encounter in those experiments?
>
> We are actively exploring adaptations of our framework to NISQ systems. In this context, a key challenge is the communication overhead between the classical and quantum hardware of the variational framework. Due to the frequent interactions between classical optimization procedures and quantum evaluations, communication bottlenecks affect runtime performance. Addressing this bottleneck is crucial to realizing a practical implementation.
>
>
>
> ### References:
> [1] Bravyi, S., Sheldon, S., Kandala, A., Mckay, D. C., & Gambetta, J. M. (2021). Mitigating measurement errors in multiqubit experiments. Physical Review A, 103(4), 042605.
>
> [2] Van Den Berg, E., Minev, Z. K., & Temme, K. (2022). Model-free readout-error mitigation for quantum expectation values. Physical Review A, 105(3), 032620.
>
> [3] Gonzales, A., Shaydulin, R., Saleem, Z. H., & Suchara, M. (2023). Quantum error mitigation by Pauli check sandwiching. Scientific Reports, 13(1), 2122.

---

> > ### Comment · Reviewer_BRsB · 2025-08-08
> > **Response to Rebuttal by Authors**
> >
> > I would like to thank the authors for their response. I have no further concerns and will maintain my score, recommending acceptance of this work.

---

> ### Author Response · Authors · 2025-08-05
>
> Dear Reviewer,
>
> Thank you again for your review and insightful feedback. We hope our rebuttal addressed your main concerns. If any clarification is still needed, or if you'd like to discuss any points further, we would be very happy to engage.
>
> We sincerely appreciate your time during this discussion phase.
>
> Best Regards.

---

### Note · Authors · 2025-08-15

We present a scalable *joint* optimization framework that simultaneously learns the PQC parameters while dynamically mitigating Pauli noise. For efficient noise mitigation, we introduce a novel quantum algorithm that unbiasedly estimates partial derivatives w.r.t. inverse noise parameters on noisy quantum hardware.

We thank the AC and reviewers for their constructive feedback, which was met with strong support after our clarifications. **BRsB** expressed “no further concerns…recommending acceptance,” **sqmh** stated “I support it,” **qjyC** noted they “lean towards this being an interesting paper… NeurIPS,” and will raise the clarity score, and **cSqD** called it “better at dealing with dynamic noise than... PEC and ZNE” and “a natural fit for NeurIPS 2025 and will raise my score accordingly.”

Reviewers recognized the novelty and relevance of our approach: “a thoughtful and promising approach” with “clear derivations of gradients” and “strong empirical results” (**BRsB**); “conceptually novel and interesting” with potential to “generalize beyond error mitigation” and offering “a way of simplifying how one understands the overall process” (**qjyC**); “novel and well-justified” with “rigorous” theory and “strong performance” (**sqmh**); and that it “investigates the interesting topic of optimizing noisy PQCs” (**cSqD**).

Concerns are focused on: “a large number of assumptions” (**BRsB**); “non-physical operations… can run the risk of optimizing towards unphysical… solutions,” urging clarity on physicality (**qjyC**); “idealized conditions such as noiseless Pauli operations and the absence of SPAM” and MNIST-only evaluation, suggesting more datasets (**sqmh**); and “too many assumptions” with a call for PEC/ZNE comparisons (**cSqD**).

In rebuttal, we clarified that noiseless Pauli assumption is achievable since errors of Pauli operations are relatively low and can be mitigated via calibration/suppression; that SPAM lies outside the targeted gate-noise layer and can be addressed independently in the pre-/post-processing steps; that the barren plateau assumption will be removed; and that we will clarify physicality of the inverse operation, add PEC/ZNE baselines, and expand experiments.

With the planned additions, the final version will address all critiques while preserving the originality and scalability of our framework, making a timely contribution to robust near-term quantum computation and filling a gap not covered by existing mitigation approaches.

---

### Decision · Program_Chairs · 2025-09-17

**Decision:**

Accept (poster)

**Comment:**

This paper proposed a gradient-based framework for learning parameterized quantum circuits (PQCs) assuming noise exists in quantum gates. In theory, this paper proved a sample complexity bound on estimation of gradients, as well as convergence guarantee of optimization the loss function by gradient descents. Numerical results demonstrate that the proposed method is better than previous quantum error mitigation methods such as Van Den Berg et al.'s Sparse Pauli Lindblad Method.

Reviewers found the studied problem and model of practical relevance. For the initial version of reviews, there are concerns about the assumptions on theory results (Assumptions 3.1-3.5) and settings of the experiments. During the rebuttal, the authors made detailed explanations and promised various changes, including that the c-locality and barren plateau assumptions can be removed because they do not affect the mathematical derivations themselves, and more experiments will be added for more complex datasets as well as for PEC/ZNE baselines. The reviewers then converge to accept this paper unanimously.

In the final version of the paper, the authors should consider to add:
- Polish the theory parts to remove unnecessary assumptions especially those on barren plateau, and make a clearer discussion on which assumption are definitely necessary to prove the main theorems (Theorem 3.1 and Theorem 3.2);
- Add more experiments for more complex datasets as well as for PEC/ZNE baselines;
- This was not mentioned in reviews, but when the AC checked the paper, it is observed that Theorem 3.2 is a bound that didn't specify the learning rate. It would be helpful for the authors to explain 1) how in theory we shall choose the learning rates to optimize the convergence rate; and 2) how are learning rates chosen in the experiments in Section 4.
- A typo to fix: Line 154, a inverse -> an inverse